# Thompson Sampling Efficiently Learns to Control Diffusion Processes

**Mohamad Kazem Shirani Faradonbeh**
Department of Statistics
University of Georgia
Athens, GA 30602
mohamadksf@uga.edu

**Mohamad Sadegh Shirani Faradonbeh**
Graduate School of Business
Stanford University
Stanford, CA, 94305
sshirani@stanford.edu

**Mohsen Bayati**
Graduate School of Business
Stanford University
Stanford, CA, 94305
bayati@stanford.edu

## Abstract

Diffusion processes that evolve according to linear stochastic differential equations are an important family of continuous-time dynamic decision-making models. Optimal policies are well-studied for them, under full certainty about the drift matrices. However, little is known about data-driven control of diffusion processes with uncertain drift matrices as conventional discrete-time analysis techniques are not applicable. In addition, while the task can be viewed as a reinforcement learning problem involving exploration and exploitation trade-off, ensuring system stability is a fundamental component of designing optimal policies. We establish that the popular Thompson sampling algorithm learns optimal actions fast, incurring only a square-root of time regret, and also stabilizes the system in a short time period. To the best of our knowledge, this is the first such result for Thompson sampling in a diffusion process control problem. We validate our theoretical results through empirical simulations with real matrices. Moreover, we observe that Thompson sampling significantly improves (worst-case) regret, compared to the state-of-the-art algorithms, suggesting Thompson sampling explores in a more guarded fashion. Our theoretical analysis involves characterization of a certain *optimality manifold* that ties the local geometry of the drift parameters to the optimal control of the diffusion process. We expect this technique to be of broader interest.

## 1  Introduction

One of the most natural reinforcement learning (RL) algorithms for controlling a diffusion process with unknown parameters is based on Thompson sampling (TS) [1]: a Bayesian posterior for the model is calculated based on its time evolution, and a control policy is then designed by treating a sampled model from the posterior as the truth. Despite its simplicity, guaranteeing efficiency and whether sampling the actions from the posterior could lead to unbounded future trajectories is unknown. In fact, the only known such theoretical result for control of a diffusion process is for an epsilon-greedy type policy that randomize the control actions at a certain rate [2].

In this work, we consider a $p$ dimensional state signal $\{\boldsymbol{x}_t\}_{t \geq 0}$ that obeys the (Ito) stochastic differential equation (SDE)

$$\boldsymbol{dx}_t = (A_0 \boldsymbol{x}_t + B_0 \boldsymbol{u}_t)\,\boldsymbol{dt} + \boldsymbol{d}\mathbb{W}_t\,, \tag{1}$$

36th Conference on Neural Information Processing Systems (NeurIPS 2022).

where the *drift matrices* $A_0$ and $B_0$ are unknown, $\boldsymbol{u}_t \in \mathbb{R}^q$ is the control action at any time $t \geq 0$, and it is designed based on values of $\boldsymbol{x}_s$ for $s \in [0, t]$. The matrix $B_0 \in \mathbb{R}^{p \times q}$ models the influence of the control action on the state evolution over time, while $A_0 \in \mathbb{R}^{p \times p}$ is the (open-loop) transition matrix reflecting interactions between the coordinates of the state vector $\boldsymbol{x}_t$. The diffusion term in (1) consists of a non-standard Wiener process $\mathbb{W}_t$ that will be defined in the next section. The goal is to study efficient RL policies that can design $\boldsymbol{u}_t$ to minimize a quadratic cost function, defined in the next section, subject to uncertainties around $A_0$ and $B_0$.

At a first glance, this problem is similar to most RL problems since the optimal policy must balance between the two objectives of learning the unknown matrices $A_0$ and $B_0$ (exploration) and optimally selecting the control signals $\boldsymbol{u}_t$ to minimize the cost (exploitation). However, unlike most RL problems that have finite or bounded-support state space, ensuring *stability*, that $\boldsymbol{x}_t$ stays bounded, is a crucial part of designing optimal policies. For example, in the discrete-time version of the problem, robust exploration is used to protect against unpredictably unstable trajectories [3–6].

**Related literature.** The existing literature studies efficiency of TS for learning optimal decisions in finite action spaces [7–12]. In this stream of research, it is shown that, over time, the posterior distribution concentrates around low-cost actions [13–15]. TS is also studied in further discrete-time settings with the environment represented by parameters that belong to a continuum, and Bayesian and frequentist regret bounds are shown for linear-quadratic regulators [16–31]. However, effectiveness of TS in highly noisy environments that are modeled by diffusion processes remains unexplored to date, due to technical challenges that will be described below.

For continuous-time linear time invariant dynamical systems, infinite-time consistency results are shown under a variety of technical assumptions, followed by alternating policies that cause (small) linear regrets [32–36]. From a computational viewpoint, pure exploration algorithms for computing optimal policies based on multiple trajectories of the state and action data are studied as well [37–39], for which a useful survey is available [40]. However, papers that study exploration versus exploitation and provide non-asymptotic estimation rates or regret bounds are limited to a few recent work about offline RL [41–43], with the exception of Randomized-Estimates policies [2] that TS outperforms them, as will be illustrated at the end of this paper.

**Contributions.** This work, first establishes that TS learns to stabilize the diffusion process (1). Specifically, in Theorem 1 of Section 3, we provide the first theoretical stabilization guarantee for diffusion processes, showing that the probability of preventing the state process from growing unbounded grows to 1, at an exponential rate that depends on square-root of the time length devoted to stabilization. As mentioned above, for RL problems with finite state spaces, the process is by definition stabilized, regardless of the policy. However, for the Euclidean state space of $\boldsymbol{x}_t$ in (1), stabilization is necessary to ensure that the state and the cost do not grow unbounded.

Then, efficiency of TS in balancing exploration versus exploitation for minimizing a cost function that has a quadratic form of both the state and the control action is shown. Indeed, we establish in Theorem 2 of Section 4 that the regret TS incurs, grows as the *square-root of time*, while the squared estimation error decays with the same rate. It is also shown that both the above quantities grow quadratically with the dimension. To the authors' knowledge, the presented results are the first theoretical analyses of TS for learning to control diffusion processes.

Additionally, through extensive simulations we illustrate that TS enjoys smaller average regret and substantially lower worst-case regret than the existing policies [2], thanks to its informed exploration.

It is important to highlight that theoretical analysis of RL policies for diffusion processes is highly non-trivial. Specifically, the conventional discrete-time RL technical tools are not applicable, due to uncountable cardinality of the random variables involved in a diffusion process, the unavoidable dependence between them, and the high level of processing and estimation noise. To address these, we make four main contributions. First, non-asymptotic and uniform upper bounds for continuous-time martingales and for Ito integrals are required to quantify the estimation accuracy. For that purpose, we establish concentration inequalities and show sub-exponential tail bounds for *double stochastic integrals*. Second, one needs sharp bounds for the impact of estimation errors on eigenvalues of certain non-linear matrices of the drift parameters that determine actions taken by TS policy. To tackle that, we perform a novel and tight *eigenvalue perturbation-analysis* based on the approximation error, dimension, and spectrum of the matrices. We also establish *Lipschitz continuity* of the control policy

with respect to the drift matrices, by developing new techniques based on matrix-valued curves. Third, to capture evaluation of both immediate and long-term effects of sub-optimal actions, we employ *Ito calculus* to bound the stochastic regret and specify effects of all problem parameters. Finally, to study learning from data trajectories that the condition number of their information matrix grows unbounded, we develop stochastic inequalities for *self-normalized continuous-time martingales*, and *spectral analysis* of non-linear functions of random matrices.

**Organization.** The organization of the subsequent sections is as follows. We formulate the problem in Section 2, while Algorithm 1 that utilizes TS for learning to stabilize the process and its high-probability performance guarantee are presented in Section 3. Then, in Section 4, TS is considered for learning to minimize a quadratic cost function, and the rates of estimation and regret are established. Next, theoretical analysis are provided in Section 5, followed by real-world numerical results of Section 6. Detailed proofs, auxiliary lemmas, and additional simulations are provided in a longer version of the paper [54].

**Notation.** The smallest (the largest) eigenvalue of matrix $M$, in magnitude, is denoted by $\underline{\lambda}(M)$ ($\overline{\lambda}(M)$). For a vector $a$, $\|a\|$ is the $\ell_2$ norm, and for a matrix $M$, $\|M\|$ is the operator norm that is the supremum of $\|Ma\|$ for $a$ on the unit sphere. $\boldsymbol{N}(\mu, \Sigma)$ is Gaussian distribution with mean $\mu$ and covariance $\Sigma$. If $\mu$ is a matrix (instead of vector), then $\boldsymbol{N}(\mu, \Sigma)$ denotes a distribution on matrices of the same dimension as $\mu$, such that all columns are independent and share the covariance matrix $\Sigma$. In this paper, transition matrices $A \in \mathbb{R}^{p \times p}$ together with input matrices $B \in \mathbb{R}^{p \times q}$ are jointly denoted by the $(p+q) \times p$ parameter matrix $\boldsymbol{\theta} = [A, B]^\top$. We employ $\vee$ ($\wedge$) for maximum (minimum). Finally, $a \lesssim b$ expresses that $a \leq \alpha_0 b$, for some fixed constant $\alpha_0$.

## 2 Problem Statement

We study the problem of designing provably efficient reinforcement learning policies for minimizing a quadratic cost function in an uncertain linear diffusion process. To proceed, fix the complete probability space $(\Omega, \{\mathcal{F}_t\}_{t \geq 0}, \mathbb{P})$, where $\Omega$ is the sample space, $\{\mathcal{F}_t\}_{t \geq 0}$ is a continuous-time filtration (i.e., increasing sigma-fields), and $\mathbb{P}$ is the probability measure defined on $\mathcal{F}_\infty$.

The state comprises the diffusion process $\boldsymbol{x}_t$ in (1), where $\boldsymbol{\theta_0} = [A_0, B_0]^\top \in \mathbb{R}^{(p+q) \times p}$ is the unknown drift parameter. The diffusion term in (1) follows infinitesimal variations of the $p$ dimensional Wiener process $\{\mathbb{W}_t\}_{t \geq 0}$. That is, $\{\mathbb{W}_t\}_{t \geq 0}$ is a multivariate Gaussian process with independent increments and with the stationary covariance matrix $\Sigma_\mathbb{W}$, such that for all $0 \leq s_1 \leq s_2 \leq t_1 \leq t_2$,

$$\begin{bmatrix} \mathbb{W}_{t_2} - \mathbb{W}_{t_1} \\ \mathbb{W}_{s_2} - \mathbb{W}_{s_1} \end{bmatrix} \sim \boldsymbol{N}\left( \begin{bmatrix} 0_p \\ 0_p \end{bmatrix}, \begin{bmatrix} (t_2 - t_1)\Sigma_\mathbb{W} & 0_{p \times p} \\ 0_{p \times p} & (s_2 - s_1)\Sigma_\mathbb{W} \end{bmatrix} \right). \tag{2}$$

Existence, construction, continuity, and non-differentiability of Wiener processes are well-known [44]. It is standard to assume that $\Sigma_\mathbb{W}$ is positive definite, which is a common condition in learning-based control [40, 41, 2, 42] to ensure accurate estimation over time.

The RL policy designs the action $\{\boldsymbol{u}_t\}_{t \geq 0}$, based on the observed system state by the time, as well as the previously applied actions, to minimize the long-run average cost

$$\limsup_{T \to \infty} \frac{1}{T} \int_0^T [\boldsymbol{x}_t^\top, \boldsymbol{u}_t^\top] Q \begin{bmatrix} \boldsymbol{x}_t \\ \boldsymbol{u}_t \end{bmatrix} dt, \quad \text{for} \quad Q = \begin{bmatrix} Q_x & Q_{xu} \\ Q_{xu}^\top & Q_u \end{bmatrix}. \tag{3}$$

Above, the cost is determined by the positive definite matrix $Q$, where $Q_x \in \mathbb{R}^{p \times p}$, $Q_u \in \mathbb{R}^{q \times q}$, $Q_{xu} \in \mathbb{R}^{p \times q}$. In fact, $Q$ determines the weights of different coordinates of $\boldsymbol{x}_t$, $\boldsymbol{u}_t$ in the cost function, so that the policy aims to make the states small, by deploying small actions. The cost matrix $Q$ is assumed known to the policy. Formally, the problem is to minimize (3) by the policy

$$\boldsymbol{u}_t = \widehat{\boldsymbol{\pi}}\left( Q, \{\boldsymbol{x}_s\}_{0 \leq s \leq t}, \{\boldsymbol{u}_s\}_{0 \leq s < t} \right). \tag{4}$$

Without loss of generality, and for the ease of presentation, we follow the canonical formulation that sets $Q_{xu} = 0$; one can simply convert the case $Q_{xu} \neq 0$ to the canonical form, by employing a rotation to $\boldsymbol{x}_t$, $\boldsymbol{u}_t$ [45–48]. It is well-known that if, hypothetically, the truth $\boldsymbol{\theta_0}$ was known, an optimal

policy $\boldsymbol{\pi_{opt}}$ could be explicitly found by solving the continuous-time algebraic Riccati equation. That is, for a generic drift matrix $\boldsymbol{\theta} = [A, B]^\top$, finding the symmetric $p \times p$ matrix $P(\boldsymbol{\theta})$ that satisfies

$$A^\top P(\boldsymbol{\theta}) + P(\boldsymbol{\theta}) A - P(\boldsymbol{\theta}) B Q_u^{-1} B^\top P(\boldsymbol{\theta}) + Q_x = 0. \tag{5}$$

This means, for the true parameter $\boldsymbol{\theta_0} = [A_0, B_0]^\top$, we can let $P(\boldsymbol{\theta_0})$ solve the above equation, and define the policy

$$\boldsymbol{\pi_{opt}} : \quad \boldsymbol{u}_t = -Q_u^{-1} B_0^\top P(\boldsymbol{\theta_0}) \boldsymbol{x}_t, \quad \forall t \geq 0. \tag{6}$$

It is known that the linear time-invariant policy $\boldsymbol{\pi_{opt}}$ minimizes the average cost in (3) [45–48]. This optimal policy also stabilizes the system such that under $\boldsymbol{\pi_{opt}}$, the diffusion process $\boldsymbol{x}_t$ does not grow unbounded with time. Below, we define stabilizability and elaborate these properties.

**Definition 1** *The process in* (1) *is stabilizable, if all eigenvalues of* $\overline{A} = A_0 + B_0 K$ *have negative real-parts, for a matrix $K$. Such $K, \overline{A}$ are called a stabilizer and the stable closed-loop matrix.*

We assume that the process (1) with the drift parameter $\boldsymbol{\theta_0}$ is stabilizable. Therefore, $P(\boldsymbol{\theta_0})$ exists, is unique, and can be computed according to continuous-time Riccati differential equations [45–48]. Furthermore, it is known that real-parts of all eigenvalues of $\overline{A}_0 = A_0 - B_0 Q_u^{-1} B_0^\top P(\boldsymbol{\theta_0})$ are negative, i.e., the matrix $\exp(\overline{A}_0 t)$ decays exponentially fast as $t$ grows [45–48]. So, under the linear feedback in (6), the closed-loop transition matrix is $\overline{A}_0$ and the solution of (1) is the Ornstein–Uhlenbeck process $\boldsymbol{x}_t = e^{\overline{A}_0 t} \boldsymbol{x}_0 + \int_0^t e^{\overline{A}_0(t-s)} d\mathbb{W}_s$ [44], which evolves in a stable manner because of $\left| \overline{\lambda}\left( \exp\left(\overline{A}_0 t\right)\right) \right| < 1$. In the sequel, we use (5) and refer to the solution $P(\boldsymbol{\theta})$ for different stabilizable $\boldsymbol{\theta}$. More details about the above optimal feedback policy can be found in the aforementioned references.

In absence of exact knowledge of $\boldsymbol{\theta_0}$, a policy $\widehat{\boldsymbol{\pi}}$ collects data and leverages it to approximate $\boldsymbol{\pi_{opt}}$ in (6). Therefore, at all (finite) times, there is a gap between the cost of $\widehat{\boldsymbol{\pi}}$, compared to that of $\boldsymbol{\pi_{opt}}$. The cumulative performance degradation due to this gap is the *regret* of the policy $\widehat{\boldsymbol{\pi}}$, that we aim to minimize. Technically, whenever the control action $\boldsymbol{u}_t$ is designed by the policy $\widehat{\boldsymbol{\pi}}$ according to (4), concatenate the resulting state and input signals to get the observation $\boldsymbol{z}_t(\widehat{\boldsymbol{\pi}}) = \left[\boldsymbol{x}_t^\top, \boldsymbol{u}_t^\top\right]^\top$. If it is clear from the context, we drop $\widehat{\boldsymbol{\pi}}$. Similarly, $\boldsymbol{z}_t(\boldsymbol{\pi_{opt}})$ denotes the observation signal of $\boldsymbol{\pi_{opt}}$. Now, the regret at time $T$ is defined by:

$$\text{Reg}_{\widehat{\boldsymbol{\pi}}}(T) = \int_0^T \left( \left\| Q^{1/2} \boldsymbol{z}_t(\widehat{\boldsymbol{\pi}}) \right\|^2 - \left\| Q^{1/2} \boldsymbol{z}_t(\boldsymbol{\pi_{opt}}) \right\|^2 \right) dt.$$

A secondary objective is the learning accuracy of $\boldsymbol{\theta_0}$ from the single trajectory of the data generated by $\widehat{\boldsymbol{\pi}}$. Letting $\widehat{\boldsymbol{\theta}_t}$ be the parameter estimate at time $t$, we are interested in scaling of $\left\| \widehat{\boldsymbol{\theta}_t} - \boldsymbol{\theta_0} \right\|$ with respect to $t$, $p$, and $q$.

## 3 Stabilizing the Diffusion Process

This section focuses on establishing that Thompson sampling (TS) learns to stabilize the diffusion process (1). First, let us intuitively discuss the problem of stabilizing unknown diffusion processes. Given that the optimal policy in (6) stabilizes the process in (1), a natural candidate to obtain a stable process under uncertainty of the drift matrices $A_0, B_0$, is a linear feedback of the form $\boldsymbol{u}_t = K\boldsymbol{x}_t$. So, letting $\overline{A} = A_0 + B_0 K$, we have $\boldsymbol{x}_t = e^{\overline{A} t} \boldsymbol{x}_0 + \int_0^t e^{\overline{A}(t-s)} d\mathbb{W}_s$ [44]. Thus, if real-part of an eigenvalue of $\overline{A}$ is non-negative, then the magnitude of $\boldsymbol{x}_t$ grows unbounded with $t$ [44]. Therefore, addressing instabilities of this form is important, *prior* to minimizing the cost. Otherwise, the regret grows (super) linearly with time. In particular, if $A_0$ has some eigenvalue(s) with non-negative real-part(s), then it is necessary to employ feedback to preclude instabilities.

In addition to minimizing the cost, the algebraic Riccati equation in (5) provides a reliable and widely-used framework for stabilization, as discussed after Definition 1. Accordingly, due to

uncertainty about $\boldsymbol{\theta_0}$, one can solve (5) and find $P\left(\widehat{\boldsymbol{\theta}}\right)$, only for an approximation $\widehat{\boldsymbol{\theta}}$ of $\boldsymbol{\theta_0}$. Then, we expect to stabilize the system in (1) by applying a linear feedback that is designed for the approximate drift matrix $\widehat{\boldsymbol{\theta}}$. Technically, we need to ensure that all eigenvalues of $A_0 - B_0 Q_u^{-1} \widehat{B}^\top P\left(\widehat{\boldsymbol{\theta}}\right)$ lie in the open left half-plane. To ensure that these requirements are met in a sustainable manner, the main challenges are

(i) fast and accurate learning of $\boldsymbol{\theta_0}$ so that after a short time period, a small error $\widehat{\boldsymbol{\theta}} - \boldsymbol{\theta_0}$ is guaranteed,

(ii) specifying the effect of the error $\widehat{\boldsymbol{\theta}} - \boldsymbol{\theta_0}$, on stability of $A_0 - B_0 Q_u^{-1}\widehat{B}^\top P\left(\widehat{\boldsymbol{\theta}}\right)$, and

(iii) devising a remedy for the case that the stabilization procedure fails.

Note that the last challenge is unavoidable, since learning from finite data can never be perfectly accurate, and so any finite-time stabilization procedure has a (possibly small) positive failure probability.

Algorithm 1 addresses the above challenges by applying additionally randomized control actions, and using them to provide a posterior belief $\mathcal{D}$ about $\boldsymbol{\theta_0}$. Note that the posterior is *not* concentrated at $\boldsymbol{\theta_0}$, and a sample $\widehat{\boldsymbol{\theta}}$ from $\mathcal{D}$ approximates $\boldsymbol{\theta_0}$, crudely. Still, the theoretical analysis of Theorem 1 indicates that the failure probability of Algorithm 1 decays exponentially fast with the length of the time interval it is executed. Importantly, this small failure probability can shrink further by repeating the procedure of sampling from $\mathcal{D}$. So, stabilization under uncertainty is guaranteed, after a limited time of interacting with the environment.

To proceed, let $\{w_n\}_{n=0}^{\kappa}$ be a sequence of independent Gaussian vectors with the distribution $w_n \sim \boldsymbol{N}\left(0, \sigma_w^2 I_q\right)$, for some fixed constant $\sigma_w$. Suppose that we aim to devote the time length $\boldsymbol{\tau}$ to collect observations for learning to stabilize. Note that since stabilization is performed before moving forward to the main objective of minimizing the cost functions, the stabilization time length $\boldsymbol{\tau}$ is desired to be as short as possible. We divide this time interval of length $\boldsymbol{\tau}$ to $\boldsymbol{\kappa}$ sub-intervals of equal length, and randomize an initial linear feedback policy by adding $\{w_n\}_{n=0}^{\kappa}$. That is, for $n = 0, 1, \cdots, \boldsymbol{\kappa} - 1$, Algorithm 1 employs the control action

$$\boldsymbol{u}_t = K\boldsymbol{x}_t + w_n, \qquad \text{for} \qquad \frac{n\boldsymbol{\tau}}{\boldsymbol{\kappa}} \le t < \frac{(n+1)\boldsymbol{\tau}}{\boldsymbol{\kappa}}, \tag{7}$$

where $K$ is an initial stabilizing feedback so that all eigenvalues of $A_0 + B_0 K$ lie in the open left half-plane. In practice, such $K$ is easily found using physical knowledge of the model, e.g., via conservative control sequence for an airplane [49, 50]. However, note that such actions are sub-optimal involving large regrets. Therefore, they are only temporarily applied, for the sake of data collection. Then, the data collected during the time interval $0 \le t \le \boldsymbol{\tau}$ will be utilized by the algorithm to determine the posterior belief $\mathcal{D}_{\boldsymbol{\tau}}$, as follows. Recalling the notation $\boldsymbol{z}_t^\top = \left[\boldsymbol{x}_t^\top, \boldsymbol{u}_t^\top\right]$, let $\widehat{\mu}_0, \widehat{\Sigma}_0$ be the mean and the precision matrix of a prior normal distribution on $\boldsymbol{\theta_0}$ (using the notation defined in Section 1 for random matrices). Such a prior belief can be objectively calculated according to a previously available data or formed by subjective believes about the diffusion process understudy, and in both cases can be used for speeding up the learning-based control of the system. Nonetheless, if there is no such prior, we simply let $\widehat{\mu}_0 = 0_{(p+q)\times p}$ and $\widehat{\Sigma}_0 = I_{p+q}$. Then, define

$$\widehat{\Sigma}_{\boldsymbol{\tau}} = \widehat{\Sigma}_0 + \int_0^{\boldsymbol{\tau}} \boldsymbol{z}_s \boldsymbol{z}_s^\top \, ds, \qquad \widehat{\mu}_{\boldsymbol{\tau}} = \widehat{\Sigma}_{\boldsymbol{\tau}}^{-1}\left(\widehat{\Sigma}_0 \widehat{\mu}_0 + \int_0^{\boldsymbol{\tau}} \boldsymbol{z}_s \, d\boldsymbol{x}_s^\top\right). \tag{8}$$

Using $\widehat{\Sigma}_{\boldsymbol{\tau}} \in \mathbb{R}^{(p+q)\times(p+q)}$ together with the mean matrix $\widehat{\mu}_{\boldsymbol{\tau}}$, Algorithm 1 forms the posterior belief

$$\mathcal{D}_{\boldsymbol{\tau}} = \boldsymbol{N}\left(\widehat{\mu}_{\boldsymbol{\tau}}, \widehat{\Sigma}_{\boldsymbol{\tau}}^{-1}\right), \tag{9}$$

about the drift parameter $\boldsymbol{\theta_0}$. So, as defined in the notation, the posterior distribution of every column $i = 1, \cdots, p$ of $\boldsymbol{\theta_0}$, is an independent multivariate normal with the covariance matrix $\widehat{\Sigma}_{\boldsymbol{\tau}}^{-1}$, while the mean is the column $i$ of $\widehat{\mu}_{\boldsymbol{\tau}}$. The final step of Algorithm 1 is to output a sample $\widehat{\boldsymbol{\theta}}$ from $\mathcal{D}_{\boldsymbol{\tau}}$.

Next, to establish performance guarantees for Algorithm 1, let us quantify the *ideal* stability by

$$\boldsymbol{\zeta_0} = -\log \overline{\lambda}\left(\exp\left[A_0 - B_0 Q_u^{-1} B_0^\top P\left(\boldsymbol{\theta_0}\right)\right]\right). \tag{10}$$

**Algorithm 1 : Stabilization under Uncertainty**

Inputs: initial feedback $K$, stabilization time length $\tau$
**for** $n = 0, 1, \cdots, \kappa - 1$ **do**
    **while** $n\tau\kappa^{-1} \leq t < (n+1)\tau\kappa^{-1}$ **do**
        Apply control action $\boldsymbol{u}_t$ in (7)
    **end while**
**end for**
Calculate $\widehat{\Sigma}_{\boldsymbol{\tau}}, \widehat{\mu}_{\boldsymbol{\tau}}$ according to (8)
Return sample $\widehat{\boldsymbol{\theta}}$ from the distribution $\mathcal{D}_{\boldsymbol{\tau}}$ in (9)

By definition, $\boldsymbol{\zeta_0}$ is positive. In fact, it is the smallest distance between the imaginary axis in the complex-plane, and the eigenvalues of the transition matrix $\overline{A}_0 = A_0 - B_0 Q_u^{-1} B_0^\top P(\boldsymbol{\theta_0})$, under the optimal policy in (6). Since $\boldsymbol{\theta_0}$ is unavailable, it is *not* realistic to expect that after applying a policy based on $\widehat{\boldsymbol{\theta}}$ given by Algorithm 1, real-parts of all eigenvalues of the resulting matrix $A_0 - B_0 Q_u^{-1} \widehat{B}^\top P\left(\widehat{\boldsymbol{\theta}}\right)$ are at most $-\boldsymbol{\zeta_0}$. However, $\boldsymbol{\zeta_0}$ is crucial in studying stabilization, such that stabilizing controllers for systems with larger $\boldsymbol{\zeta_0}$ can be learned faster. The exact effect of this quantity, as well as those of other properties of the diffusion process, are formally established in the following result. Informally, the failure probability of Algorithm 1 decays exponentially with $\boldsymbol{\tau}^{1/2}$.

**Theorem 1 (Stabilization Guarantee)** *For the sample $\widehat{\boldsymbol{\theta}}$ given by Algorithm 1, let $\mathcal{E}_{\boldsymbol{\tau}}$ be the failure event that $A_0 - B_0 Q_u^{-1} \widehat{B}^\top P\left(\widehat{\boldsymbol{\theta}}\right)$ has an eigenvalue in the closed right half-plane. Then, if $\boldsymbol{\kappa} \gtrsim \boldsymbol{\tau}^2$ and $\log\left(pq\boldsymbol{\kappa}\right) \lesssim \boldsymbol{\tau}^{1/2}$, we have*

$$\log \mathbb{P}(\mathcal{E}_{\boldsymbol{\tau}}) \lesssim - \frac{\underline{\lambda}(\Sigma_{\mathbb{W}}) \wedge \sigma_w^2}{\overline{\lambda}(\Sigma_{\mathbb{W}}) \vee \sigma_w^2} \frac{1 \wedge \boldsymbol{\zeta_0}^p}{1 \vee \|K\|^3} \sqrt{\frac{\tau}{p^3 q}}. \tag{11}$$

The above result indicates that more heterogeneity in coordinates of the Wiener noise renders stabilization harder. Moreover, using (10), the term $1 \wedge \boldsymbol{\zeta_0}^p$ reflects that less stable diffusion processes with smaller $\boldsymbol{\zeta_0}$, are significantly harder to stabilize under uncertainty. Also as one can expect, larger dimensions make learning to stabilize harder. This is contributed by higher number of parameters to learn, as well as higher sensitivity of eigenvalues for processes of larger dimensions. Finally, the failure probability decays as $\boldsymbol{\tau}^{1/2}$, mainly because continuous-time martingales have sub-exponential distributions, unlike sub-Gaussianity of discrete-time counterparts [51–53].

## 4 Thompson Sampling for Efficient Control: Algorithm and Theory

In this section, we proceed towards analysis of Thompson sampling (TS) for minimizing the quadratic cost in (3), and show that it efficiently learns the optimal control actions. That is, TS balances the exploration versus exploitation, such that its regret grows with (nearly) the square-root rate, as time grows. In the sequel, we introduce Algorithm 2 and discuss the conceptual and technical frameworks it relies on. Then, we establish efficiency by showing regret bounds in terms of different problem parameters and provide the rates of estimating the unknown drift matrices.

In Algorithm 2, first the learning-based stabilization Algorithm 1 is run during the time period $0 \leq t < \boldsymbol{\tau}_0$. So, according to Theorem 1, the optimal feedback of $\widehat{\boldsymbol{\theta}_0}$ stabilizes the system with a high probability, as long as $\boldsymbol{\tau}_0$ is sufficiently large. Note that if growth of the state vector indicates that Algorithm 1 failed to stabilize, one can repeat sampling from $\mathcal{D}_{\boldsymbol{\tau}_0}$. So, we assume that the evolution of the controlled diffusion process remains stable when Algorithm 2 is being executed. On the other hand, the other benefit of running Algorithm 1 at the beginning is that it performs an initial exploration phase that will be utilized by Algorithm 2 to minimize the regret.

Then, in order to learn the optimal policy $\boldsymbol{\pi_{opt}}$ with minimal sub-optimality, RL algorithms need to cope with a fundamental challenge, commonly known as the exploration-exploitation dilemma. To see that, first note that an acceptable policy that aims to have sub-linear regret, needs to take near-optimal control actions in a long run; $\boldsymbol{u}_t \approx -Q_u^{-1} B_0^\top P(\boldsymbol{\theta_0}) \boldsymbol{x}_t$. Although such policies exploit well and their control actions are close to that of $\boldsymbol{\pi_{opt}}$, their regret grows large since they fail to

explore. Technically, the trajectory of observations $\{z_t\}_{t \geq 0}$ is not rich enough to provide accurate estimations, since in $z_t^\top = \left[ x_t^\top, u_t^\top \right]$, the signal $u_t$ is (almost) a linear function of the state signal $x_t$, and so does not contribute towards gathering information about the unknown parameter $\theta_0$. Conversely, for sufficient explorations, RL policies need to take actions that deviate from those of $\pi_{\mathrm{opt}}$, which imposes large regret. Accordingly, the above trade-off needs to be delicately balanced; what we show that TS does.

Algorithm 2 is episodic; the parameter estimates $\widehat{\theta}_n$ are updated only at the end of the episodes at times $\{\tau_n\}_{n=0}^\infty$, while during every episode, actions are taken *as if* $\widehat{\theta}_n = \left[ \widehat{A}_n, \widehat{B}_n \right]^\top$ is the unknown truth $\theta_0$. That is, for $\tau_{n-1} \leq t < \tau_n$, using $P\left(\widehat{\theta}_n\right)$ in (5), we let $u_t = -Q_u^{-1} \widehat{B}_n^\top P\left(\widehat{\theta}_n\right) x_t$. Then, for each $n = 1, 2, \cdots$, at time $\tau_n$, we use all the observations collected so far, to find $\widehat{\Sigma}_{\tau_n}, \widehat{\mu}_{\tau_n}$ according to (8). Next, we use them to sample $\widehat{\theta}_n$ from the posterior $\mathcal{D}_{\tau_n}$ in (9).

The episodes in Algorithm 2 are chosen such that their end points satisfy

$$0 < \underline{\alpha} \leq \inf_{n \geq 0} \frac{\tau_{n+1} - \tau_n}{\tau_n} \leq \sup_{n \geq 0} \frac{\tau_{n+1} - \tau_n}{\tau_n} \leq \overline{\alpha} < \infty, \tag{12}$$

for some fixed constants $\underline{\alpha}, \overline{\alpha}$. Broadly speaking, (12) lets the episode lengths of Algorithm 2 scale properly to avoid unnecessary updates of parameter estimates, while at the same time performing sufficient exploration. To see that, first note that since $\widehat{\Sigma}_\tau$ grows with $\tau$, the estimation error $\widehat{\theta}_n - \theta_0$ decays (at best polynomially fast) with $\tau_n$. So, until ensuring that updating the posterior yields to significantly better approximations, it will not be beneficial to update it, sample from it, and solve (5). So, the period $\tau_{n+1} - \tau_n$ that the data up to time $\tau_n$ is utilized, is set to be as long as $\underline{\alpha} \tau_n$. On the other hand, the above period cannot be too long, since we aim to improve the parameter estimates after collecting enough new observations; $\tau_{n+1} \leq (1 + \overline{\alpha}) \tau_n$. A simple setting is to let $\underline{\alpha} = \overline{\alpha}$, which yields to exponential episodes $\tau_n = \tau_0 (1 + \overline{\alpha})^n$. Note that for TS in continuous time, posterior updates should be limited to sufficiently-apart time points. Otherwise, repetitive updates are computationally impractical, and also can degrade the performance by preventing control actions from having enough time to effectively influence.

---

**Algorithm 2 : Thompson Sampling for Efficient Control of Diffusion Processes**

Inputs: stabilization time $\tau_0$
Calculate sample $\widehat{\theta}_0$ by running Algorithm 1 for time $\tau_0$
**for** $n = 1, 2, \cdots$ **do**
    **while** $\tau_{n-1} \leq t < \tau_n$ **do**
        Apply control action $u_t = -Q_u^{-1} \widehat{B}_{n-1}^\top P\left(\widehat{\theta}_{n-1}\right) x_t$
    **end while**
    Letting $\widehat{\Sigma}_{\tau_n}, \widehat{\mu}_{\tau_n}$ be as (8), sample $\widehat{\theta}_n$ from $\mathcal{D}_{\tau_n}$ given in (9)
**end for**

---

We show next that Algorithm 2 addresses the exploration-exploitation trade-off efficiently. To see the intuition, consider the sequence of posteriors $\mathcal{D}_{\tau_n}$. The explorations Algorithm 2 performs by sampling $\widehat{\theta}_n$ from $\mathcal{D}_{\tau_n}$, depends on $\widehat{\Sigma}_{\tau_n}$. Now, if hypothetically $\underline{\lambda}\left(\widehat{\Sigma}_{\tau_n}\right)$ is not large enough, then $\mathcal{D}_{\tau_n}$ does not sufficiently concentrate around $\widehat{\mu}_{\tau_n}$ and so $\widehat{\theta}_n$ will probably deviate from the previous samples $\left\{ \widehat{\theta}_i \right\}_{i=1}^{n-1}$. So, the algorithm explores more and obtains richer data $z_t$ by diversifying the control signal $u_t$. This renders the next mean $\widehat{\mu}_{\tau_{n+1}}$ a more accurate approximation of $\theta_0$, and also makes $\underline{\lambda}\left(\widehat{\Sigma}_{\tau_{n+1}}\right)$ grow faster than before. Thus, the next posterior $\mathcal{D}_{\tau_{n+1}}$ provides a better sample with smaller estimation error $\widehat{\theta}_{n+1} - \theta_0$. Similarly, if a posterior is excessively concentrated, in a few episodes the posteriors adjust accordingly to the proper level of exploration. Hence, TS eventually balances the exploration versus the exploitation. This is formalized below.

**Theorem 2 (Regret and Estimation Rates)** *Parameter estimates and regret of Algorithm 2, satisfy*

$$\left\| \widehat{\boldsymbol{\theta}}_{\boldsymbol{n}} - \boldsymbol{\theta_0} \right\|^2 \; \lesssim \; \frac{\overline{\lambda}\left(\Sigma_{\mathbb{W}}\right)}{\underline{\lambda}\left(\Sigma_{\mathbb{W}}\right)} \; (p+q)\,p \; \; \boldsymbol{\tau}_n^{-1/2} \log \boldsymbol{\tau}_n \,,$$

$$\mathrm{Reg}\left(T\right) \; \lesssim \; \left(\overline{\lambda}\left(\Sigma_{\mathbb{W}}\right) + \sigma_w^2\right) \boldsymbol{\tau}_0 + \frac{\overline{\lambda}\left(\Sigma_{\mathbb{W}}\right)^2}{\underline{\lambda}\left(\Sigma_{\mathbb{W}}\right)} \frac{\left\| P\left(\boldsymbol{\theta_0}\right) \right\|^6}{\underline{\lambda}\left(Q\right)^6} \; \; (p+q)\,p \; \; T^{1/2} \log T \,.$$

In the above regret and estimation rates, and similar to Theorem 1, $\overline{\lambda}\left(\Sigma_{\mathbb{W}}\right)/\underline{\lambda}\left(\Sigma_{\mathbb{W}}\right)$ reflects the impact of heterogeneity in coordinates of $\mathbb{W}_t$ on the quality of learning. Also, larger $\log(1 + \overline{\alpha})$ corresponds to longer episodes which compromises the estimation. Further, $p(p+q)$ shows that larger number of parameters linearly worsens the learning accuracy. In the regret bound, $\|P\left(\boldsymbol{\theta_0}\right)\|/\underline{\lambda}\left(Q\right)$ indicates effect of the true problem parameters $\boldsymbol{\theta_0}, Q$. Finally, $\left(\overline{\lambda}\left(\Sigma_{\mathbb{W}}\right) + \sigma_w^2\right)\boldsymbol{\tau}_0$ captures the initial phase that Algorithm 1 is run for stabilization, which takes sub-optimal control actions as in (7).

## 5    Intuition and Summary of the Analysis

The goal of this section is to provide a high-level roadmap of the proofs of Theorems 1 and 2, and convey the main intuition behind the analysis. Complete proofs and the technical lemmas can be found in the longer version of the paper [54].

**Summary of the Proof of Theorem 1.**    The main steps involve analyzing the estimation, studying its effect on the solutions of (5), and characterizing impact of errors in entries of parameter matrices on their eigenvalues. Next, we elaborate on these steps.

We show that the error satisfies $\left\| \widehat{\boldsymbol{\theta}} - \boldsymbol{\theta_0} \right\| \lesssim p(p+q)^{1/2}\boldsymbol{\tau}^{-1/2}$. More precisely, the error depends mainly on total strength of the observation signals $\boldsymbol{z}_t$, which are captured in the precision matrix $\widehat{\Sigma}_{\boldsymbol{\tau}}$, as well as total interactions between the signal $\boldsymbol{z}_t$ and the noise $\mathbb{W}_t$ in the form of the stochastic integral matrix $\int_0^{\boldsymbol{\tau}} \boldsymbol{z}_t d\mathbb{W}_t^{\top}$. However, we establish an upper bound $\overline{\lambda}\left(\widehat{\Sigma}_{\boldsymbol{\tau}}^{-1}\right) \lesssim \boldsymbol{\tau}^{-1}$, that indicates the concentration rate of the posterior $\mathcal{D}_{\boldsymbol{\tau}}$. Similarly, thanks to the randomization signal $w_n$, the signals $\boldsymbol{z}_t$ are diverse enough to effectively explore the set of matrices $\boldsymbol{\theta} = [A, B]^{\top}$, leading to accurate approximation of $\boldsymbol{\theta_0}$ by the posterior mean matrix $\widehat{\mu}_{\boldsymbol{\tau}}$. Then, to bound the error terms caused by the Wiener noise $\mathbb{W}_t$, we establish the rate $p(p+q)^{1/2}\boldsymbol{\tau}^{1/2}$. Indeed, we show that the entries of this error matrix are continuous-time martingales, and use exponential inequalities for quadratic forms and double stochastic integrals [52, 51] to establish that they have a sub-exponential distribution.

Moreover, the error rate of the feedback satisfies a similar property; $\left\| \widehat{B}^{\top} P\left(\widehat{\boldsymbol{\theta}}\right) - B_0^{\top} P\left(\boldsymbol{\theta_0}\right) \right\| \lesssim p(p+q)^{1/2}\boldsymbol{\tau}^{-1/2}$. So, letting $\overline{A} = A_0 - B_0 Q_u^{-1}\widehat{B}^{\top} P\left(\widehat{\boldsymbol{\theta}}\right)$ and $\overline{A}_0 = A_0 - B_0 Q_u^{-1} B_0^{\top} P\left(\boldsymbol{\theta_0}\right)$, it holds that $\left\| \overline{A} - \overline{A}_0 \right\| \lesssim p(p+q)^{1/2}\boldsymbol{\tau}^{-1/2}$. Next, to consider the effect of the errors on the eigenvalues of $\overline{A}$, we compare them to the eigenvalues of $\overline{A}_0$, which are bounded by $-\boldsymbol{\zeta_0}$ in (10). To that end, we establish a novel and tight perturbation analysis for eigenvalues of matrices, with respect to their entries and spectral properties. Using that, we show that the difference between the eigenvalues of $\overline{A}$ and $\overline{A}_0$ scales as $\left(1 \vee r^{1/2}\left\|\overline{A} - \overline{A}_0\right\|\right)^{1/r}$, where $r$ is the size of the largest block in the Jordan block-diagonalization of $\overline{A}_0$. Therefore, for stability of $\overline{A}$, we need $\left\|\overline{A} - \overline{A}_0\right\| \lesssim p^{-1/2}\left(1 \wedge \boldsymbol{\zeta_0}^p\right)$, since $r \leq p$. Note that if $\overline{A}_0$ is diagonalizable, $r = 1$ implies that we can replace the above upper bound by $1 \wedge \boldsymbol{\zeta_0}$. Putting this stability result together with the estimation error in the previous paragraph, we obtain (11).

**Summary of the Proof of Theorem 2.**    To establish the estimation rates, we develop multiple intermediate lemmas quantifying the exact amount of exploration Algorithm 2 performs. First, we utilize the fact that the bias of the posterior distribution $\mathcal{D}_{\boldsymbol{\tau}_n}$ depends on its covariance matrix $\widehat{\Sigma}_{\boldsymbol{\tau}_n}$, as well as a self-normalized continuous-time matrix-valued martingale. For the effect of the former, i.e., $\overline{\lambda}\left(\widehat{\Sigma}_{\boldsymbol{\tau}_n}^{-1/2}\right)$, we show an upper-bound of the order $\boldsymbol{\tau}_n^{-1/4}$. To that end, the local geometry of the optimality manifolds that contain drift parameters $\boldsymbol{\theta}$ that has the same optimal feedback as that of the

unknown truth $\boldsymbol{\theta_0}$ in (6) are fully specified, and spectral properties of non-linear functions of random matrices are studied. Then, we establish a stochastic inequality for the self-normalized martingale, indicating that its scaling is of the order $p(p+q)\log \boldsymbol{\tau}_n$. Therefore, utilizing the fact that $\widehat{\boldsymbol{\theta}}_n - \widehat{\mu}_{\boldsymbol{\tau}_n}$ has the same scaling as the bias matrix $\widehat{\mu}_{\boldsymbol{\tau}_n} - \boldsymbol{\theta_0}$, we obtain the estimation rates of Theorem 2.

Next, to prove the presented regret bound, we establish a delicate and tight analysis for the dominant effect of the control signal $\boldsymbol{u}_t$ on the regret Algorithm 2 incurs. Technically, by carefully examining the infinitesimal influences of the control actions at every time on the cost, we show that it suffices to integrate the squared deviations $\left\| \boldsymbol{u}_t + Q_u^{-1}\widehat{B}_n^\top P\left(\widehat{\boldsymbol{\theta}}_n\right)\boldsymbol{x}_t \right\|^2$ to obtain $\mathrm{Reg}\,(T)$. We proceed toward specifying the effect of the exploration Algorithm 2 performs on its exploitation performance by proving the Lipschitz continuity of the solutions of the Riccati equation (5) with respect to the drift parameters: $\left\| P\left(\widehat{\boldsymbol{\theta}}_n\right) - P\left(\boldsymbol{\theta_0}\right) \right\| \lesssim \left\| \widehat{\boldsymbol{\theta}}_n - \boldsymbol{\theta_0} \right\|$. This result is a very important property of (5) that lets the rates of deviations from the optimal action scale the same as the estimation error, and is proven by careful analysis of integration along matrix-valued curves in the space of drift matrices, as well as spectral analysis for approximate solutions of a Lyapunov equation. Thus, the regret bound is achieved, using the estimation error result in Theorem 2.

## 6    Numerical Analysis

We empirically evaluate the theoretical results of Theorems 1 and 2 for the flight control of X-29A airplane at 2000 ft [49]. Further numerical results for Boeing 747 airplanes [50] and for blood glucose control [55], can be found in the longer version of the paper [54]. The true drift matrices of the X-29A airplane are

$$A_0 = \begin{bmatrix} -0.16 & 0.07 & -1.00 & 0.04 \\ -15.20 & -2.60 & 1.11 & 0.00 \\ 6.84 & -0.10 & -0.06 & 0.00 \\ 0.00 & 1.00 & 0.07 & 0.00 \end{bmatrix}, \quad B_0 = \begin{bmatrix} -0.0006 & 0.0007 \\ 1.3430 & 0.2345 \\ 0.0897 & -0.0710 \\ 0.0000 & 0.0000 \end{bmatrix}.$$

Further, we let $\Sigma_{\mathbb{W}} = 0.25\,I_p$, $Q_x = I_p$, and $Q_u = 0.1\,I_q$ where $I_n$ is the $n$ by $n$ identity matrix. To update the diffusion process $\boldsymbol{x}_t$ in (1), time-steps of length $10^{-3}$ are employed. Then, in Algorithm 1, we let $\sigma_w = 5, \boldsymbol{\kappa} = \lfloor \boldsymbol{\tau}^{3/2} \rfloor$, while $\boldsymbol{\tau}$ varies from 4 to 20 seconds. The initial feedback $K$ is generated randomly. The results for 1000 repetitions are depicted on the left plot of Figure 1, confirming Theorem 1 that the failure probability of stabilization, decreases exponentially in $\boldsymbol{\tau}$.

On the right hand side of Figure 1, Algorithm 2 is executed for 600 second, for $\boldsymbol{\tau}_n = 20 \times 1.1^n$. We compare TS with the *Randomized Estimate* algorithm [2] for 100 different repetitions. Average- and worst-case values of the estimation error and the regret are reported, both normalized by their scaling with time and dimension, as in Theorem 2. The graphs show that (especially the worst-case) regret of TS substantially outperforms, suggesting that TS explores in a more robust fashion.

## 7    Concluding Remarks and Future Work

We studied Thompson sampling (TS) reinforcement learning policies to control a diffusion process with unknown drift matrices. First, we proposed a stabilization algorithm for linear diffusion processes, and established that its failure probability decays exponentially with time. Furthermore, efficiency of TS in balancing exploration versus exploitation for minimizing a quadratic cost function is shown. More precisely, regret bounds growing as square-root of time and square of dimensions are established for Algorithm 2. Empirical studies showcasing superiority of TS over state-of-the-art, are provided as well.

As the first theoretical analysis of TS for control of a continuous-time model, this work implies multiple important future directions. Establishing minimax regret lower-bounds for diffusion process control problem is yet unanswered. Moreover, studying the performance of TS for robust control of the diffusion processes aiming to simultaneously minimize the cost function for a family of drift matrices, is also an interesting direction for further investigation. Another problem of interest is efficiency of TS for learning to control under partial observation where the state is not observed and instead a noisy linear function of the state is available as the output signal.

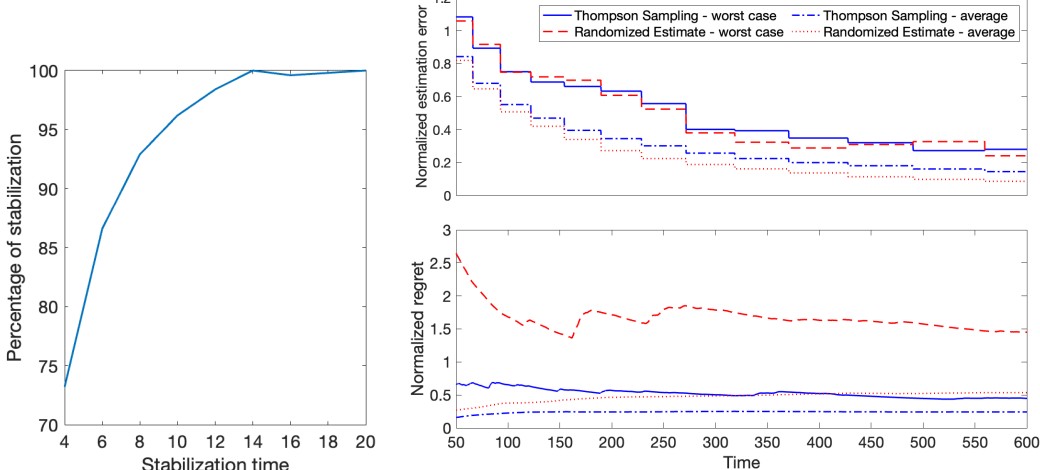

Figure 1: For the X-29A flight control problem, percentage of stabilization for 1000 runs of Algorithm 1 is plotted on the left. The graphs on the right depict the performance of Algorithm 2 (blue) compared to Randomized Estimate policy (red) [2]. The top graph plots the normalized squared estimation error, $\left\|\widehat{\boldsymbol{\theta}}_{\boldsymbol{n}} - \boldsymbol{\theta}_{\boldsymbol{0}}\right\|^2$ divided by $p(p+q)\boldsymbol{\tau}_n^{-1/2}\log\boldsymbol{\tau}_n$, versus time, while the lower one showcases the regret $\text{Reg}\,(T)$, normalized by $p(p+q)T^{1/2}\log T$. Curves for the worst-case among 100 replications are provided for both quantities, as well as for the averages over all replicates.

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
