**for** $n = 0, 1, \cdots, \boldsymbol{\kappa} - 1$ **do**
    **while** $n\boldsymbol{\tau}\boldsymbol{\kappa}^{-1} \leq t < (n+1)\boldsymbol{\tau}\boldsymbol{\kappa}^{-1}$ **do**
        Apply control action $\boldsymbol{u}_t$ in (7)
    **end while**
**end for**
Calculate $\widehat{\Sigma}_{\boldsymbol{\tau}}, \widehat{\mu}_{\boldsymbol{\tau}}$ according to (8)
Return sample $\widehat{\boldsymbol{\theta}}$ from the distribution $\mathcal{D}_{\boldsymbol{\tau}}$ in (9)

---

Next, to establish performance guarantees for Algorithm 1, let us quantify the *ideal* stability by

$$\boldsymbol{\zeta_0} = -\log \overline{\lambda}\left(\exp\left[A_0 - B_0 Q_u^{-1} B_0^{\top} P\left(\boldsymbol{\theta_0}\right)\right]\right). \tag{10}$$

By definition, $\boldsymbol{\zeta_0}$ is positive. In fact, it is the smallest distance between the imaginary axis in the complex-plane, and the eigenvalues of the transition matrix $\overline{A}_0 = A_0 - B_0 Q_u^{-1} B_0^{\top} P\left(\boldsymbol{\theta_0}\right)$, under the optimal policy in (6). Since $\boldsymbol{\theta_0}$ is unavailable, it is *not* realistic to expect that after applying a policy based on $\widehat{\boldsymbol{\theta}}$ given by Algorithm 1, real-parts of all eigenvalues of the resulting matrix $A_0 - B_0 Q_u^{-1} \widehat{B}^{\top} P\left(\widehat{\boldsymbol{\theta}}\right)$ are at most $-\boldsymbol{\zeta_0}$. However, $\boldsymbol{\zeta_0}$ is crucial in studying stabilization, such that stabilizing controllers for systems with larger $\boldsymbol{\zeta_0}$ can be learned faster. The exact effect of this quantity, as well as those of other properties of the diffusion process, are formally established in the following result. Informally, the failure probability of Algorithm 1 decays exponentially with $\boldsymbol{\tau}^{1/2}$.

**Theorem 1 (Stabilization Guarantee)** *For the sample $\widehat{\theta}$ given by Algorithm 1, let $\mathcal{E}_{\tau}$ be the failure event that $A_0 - B_0 Q_u^{-1} \widehat{B}^{\top} P\left(\widehat{\theta}\right)$ has an eigenvalue in the closed right half-plane. Then, if $\kappa \gtrsim \tau^2$, we have*

$$\log \mathbb{P}(\mathcal{E}_{\tau}) \lesssim - \frac{\underline{\lambda}\left(\Sigma_{\mathbb{W}}\right) \wedge \sigma_w^2}{\overline{\lambda}\left(\Sigma_{\mathbb{W}}\right) \vee \sigma_w^2} \frac{1 \wedge \

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

Calculate sample $\widehat{\boldsymbol{\theta}_0}$ by running Algorithm 1 for time $\boldsymbol{\tau}_0$
**for** $n = 1, 2, \cdots$ **do**
    **while** $\boldsymbol{\tau}_{n-1} \leq t < \boldsymbol{\tau}_n$ **do**
        Apply control action $\boldsymbol{u}_t = -Q_u^{-1}\widehat{B}_{n-1}^\top P\left(\widehat{\boldsymbol{\theta}_{n-1}}\right)\boldsymbol{x}_t$
    **end while**
    Letting $\widehat{\Sigma}_{\boldsymbol{\tau}_n}, \widehat{\mu}_{\boldsymbol{\tau}_n}$ be as (8), sample $\widehat{\boldsymbol{\theta}_n}$ from $\mathcal{D}_{\boldsymbol{\tau}_n}$ given in (9)
**end for**

---

We show next that Algorithm 2 addresses the exploration-exploitation trade-off efficiently. To see the intuition, consider the sequence of posteriors $\mathcal{D}_{\boldsymbol{\tau}_n}$. The explorations Algorithm 2 performs by sampling $\widehat{\boldsymbol{\theta}_n}$ from $\mathcal{D}_{\boldsymbol{\tau}_n}$, depends on $\widehat{\Sigma}_{\boldsymbol{\tau}_n}$. Now, if hypothetically $\underline{\lambda}\left(\widehat{\Sigma}_{\boldsymbol{\tau}_n}\right)$ is not large enough, then $\mathcal{D}_{\boldsymbol{\tau}_n}$ does not sufficiently concentrate around $\widehat{\mu}_{\boldsymbol{\tau}_n}$ and so $\widehat{\boldsymbol{\theta}_n}$ will probably deviate from the previous samples $\left\{\widehat{\boldsymbol{\theta}_i}\right\}_{i=1}^{n-1}$. So, the algorithm explores more and obtains richer data $\boldsymbol{z}_t$ by diversifying the control signal $\boldsymbol{u}_t$. This renders the next mean $\widehat{\mu}_{\boldsymbol{\tau}_{n+1}}$ a more accurate approximation of $\boldsymbol{\theta}_0$, and also makes $\underline{\lambda}\left(\widehat{\Sigma}_{\boldsymbol{\tau}_{n+1}}\right)$ grow faster than before. Thus, the next posterior $\mathcal{D}_{\boldsymbol{\tau}_{n+1}}$ provides a better sample with smaller estimation error $\widehat{\boldsymbol{\theta}_{n+1}} - \boldsymbol{\theta}_0$. Similarly, if a posterior is excessively concentrated, in a few episodes the posteriors adjust accordingly to the proper level of exploration. Hence, TS eventually balances the exploration versus the exploitation. This is formalized below.

**Theorem 2 (Regret and Estimation Rates)** *Parameter estimates and regret of Algorithm 2, satisfy*

$$\left\|\widehat{\boldsymbol{\theta}_n} - \boldsymbol{\theta}_0\right\|^2 \;\lesssim\; \frac{\overline{\lambda}(\Sigma_{\mathbb{W}})}{\underline{\lambda}(\Sigma_{\mathbb{W}})}\log(1 + \overline{\alpha}) \quad (p+q)\,p \quad \boldsymbol{\tau}_n^{-1/2}\log\boldsymbol{\tau}_n\,,$$

$$\mathrm{Reg}\,(T) \;\lesssim\; \left(\overline{\lambda}(\Sigma_{\mathbb{W}}) + \sigma_w^2\right)\boldsymbol{\tau}_0 + \frac{\overline{\lambda}(\Sigma_{\mathbb{W}})^2}{\underline{\lambda}(\Sigma_{\mathbb{W}})}\frac{\overline{\alpha}\|P(\boldsymbol{\theta}_0)\|^6}{\log(\underline{\alpha}+1)\underline{\lambda}(Q)^6} \quad (p+q)\,p \quad

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

simulation is for blood glucose control [41]. We present the results for X-29A airplane in this section, and defer the other two examples to the appendix. The true drift matrices of the X-29A airplane are $A_0 = \begin{bmatrix} -0.16 & 0.07 & -1.00 & 0.04 \\ -15.20 & -2.60 & 1.11 & 0.00 \\ 6.84 & -0.10 & -0.06 & 0.00 \\ 0.00 & 1.00 & 0.07 & 0.00 \end{bmatrix}$, $B_0 = \begin{bmatrix} -0.0006 & 0.0007 \\ 1.3430 & 0.2345 \\ 0.0897 & -0.0710 \\ 0.0000 & 0.0000 \end{bmatrix}$. Further, we let $\Sigma_{\mathbb{W}} = 0.5\, I_p$, $Q_x = I_p$, and $Q_u = 0.1\, I_q$ where $I_n$ is the $n$ by $n$ identity matrix. To update the diffusion process $\boldsymbol{x}_t$ in (1), time-steps of length $10^{-3}$ are employed. Then, in Algorithm 1, we let $\sigma_w = 5$, $\boldsymbol{\kappa} = \lfloor \boldsymbol{\tau}^{3/2} \rfloor$, while $\boldsymbol{\tau}$ varies from $4$ to $20$ seconds. The initial feedback $K$ is generated randomly. The results for 1000 repetitions are depicted on the left plot of Figure 1, confirming Theorem 1 that the failure probability of stabilization, decreases exponentially in $\boldsymbol{\tau}$.

On the right hand side of Figure 1, Algorithm 2 is executed for 600 second, for $\boldsymbol{\tau}_n = 20 \times 1.1^n$. We compare TS with the *Randomized Estimate* algorithm [2] for 100 different repetitions. Average- and worst-case values of the estimation error and the regret are reported, both normalized by their scaling with time and dimension, as in Theorem 2. The graphs show that (especially the worst-case) regret of TS substantially outperforms, suggesting that TS explores in a more robust fashion. Simulations for Boeing 747 and for the blood glucose control, in the appendix, corroborate the above findings.

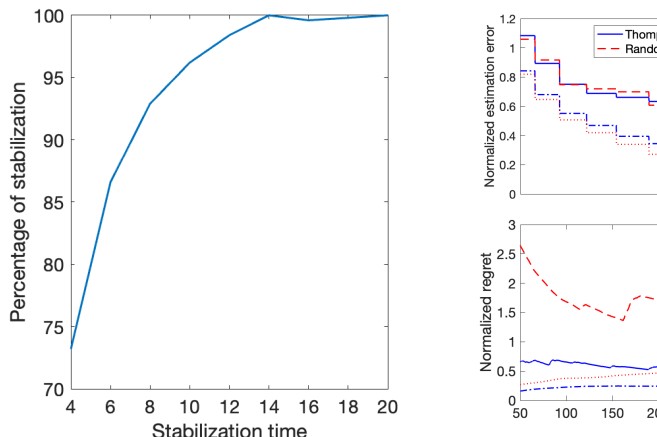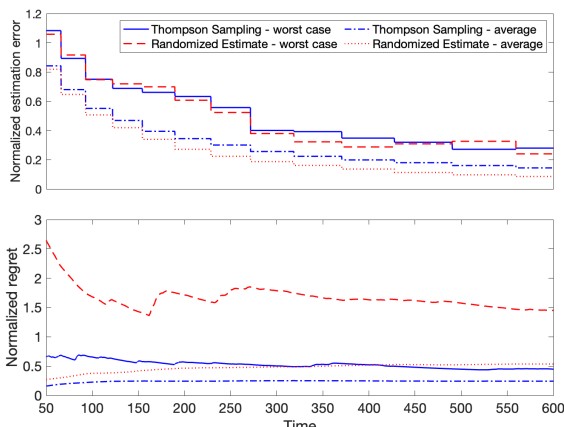

Figure 1: For the X-29A flight control problem, percentage of stabilization for 1000 runs of Algorithm 1 is plotted on the left. The graphs on the right depict the performance of Algorithm 2 (blue) compared to Randomized Estimate policy (red) [2]. The top graph plots the normalized squared estimation error, $\left\| \widehat{\boldsymbol{\theta}_n} - \boldsymbol{\theta_0} \right\|^2$ divided by $p(p+q)\boldsymbol{\tau}_n^{-1/2} \log \boldsymbol{\tau}_n$, versus time, while the lower one showcases the regret $\mathrm{Reg}\,(T)$, normalized by $p(p+q)T^{1/2}\log T$. Curves for the worst-case among 100 replications are provided for both quantities, as well as for the averages over all replicates.

## 7 Concluding Remarks and Future Work

We studied Thompson sampling (TS) RL policies to control a diffusion process with unknown drift matrices. First, we proposed a stabilization algorithm for linear diffusion processes, and established that its failure probability decays exponentially with time. Further, efficiency of TS in balancing exploration versus exploitation for minimizing a quadratic cost function is shown. More precisely, regret bounds growing as square-root of time and square of dimensions are established for Algorithm 2. Empirical studies showcasing superiority of TS over state-of-the-art are provided as well.

As the first theoretical analysis of TS for control of a continuous-time model, this work implies multiple important future directions. Establishing minimax regret lower-bounds for diffusion process control problem is yet unanswered. Moreover, studying the performance of TS for robust control of the diffusion processes aiming to simultaneously minimize the cost function for a family of drift matrices, is also an interesting direction for further investigation. Another problem of interest is efficiency of TS for learning to control under partial observation where the state is not observed and instead a noisy linear function of the state is available as the output signal.

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

# Checklist

1. For all authors...

   (a) Do the main claims made in the abstract and introduction accurately reflect the paper's contributions and scope? [Yes]

   (b) Did you describe the limitations of your work? [Yes] ; See Sections 3 and 4.

   (c) Did you discuss any potential negative societal impacts of your work? [N/A]

   (d) Have you read the ethics review guidelines and ensured that your paper conforms to them? [Yes]

2. If you are including theoretical results...

   (a) Did you state the full set of assumptions of all theoretical results? [Yes] ; See Theorems 1 and 2, as well as their discussions.

   (b) Did you include complete proofs of all theoretical results? [Yes] ; Proofs are provided as appendices.

3. If you ran experiments...

   (a) Did you include the code, data, and instructions needed to reproduce the main experimental results (either in the supplemental material or as a URL)? [Yes] ; See Section 6

   (b) Did you specify all the training details (e.g., data splits, hyperparameters, how they were chosen)? [N/A]

   (c) Did you report error bars (e.g., with respect to the random seed after running experiments multiple times)? [N/A] ; The reported curves reflect the worst-case analysis and so no error bars are needed.

   (d) Did you include the total amount of compute and the type of resources used (e.g., type of GPUs, internal cluster, or cloud provider)? [No] .

4. If you are using existing assets (e.g., code, data, models) or curating/releasing new assets...

   (a) If your work uses existing assets, did you cite the creators? [N/A]

   (b) Did you mention the license of the assets? [N/A]

   (c) Did you include any new assets either in the supplemental material or as a URL? [N/A]

   (d) Did you discuss whether and how consent was obtained from people whose data you're using/curating? [N/A]

   (e) Did you discuss whether the data you are using/curating contains personally identifiable information or offensive content? [N/A]

5. If you used crowdsourcing or conducted research with human subjects...

   (a) Did you include the full text of instructions given to participants and screenshots, if applicable? [N/A]

   (b) Did you describe any potential participant risks, with links to Institutional Review Board (IRB) approvals, if applicable? [N/A]

   (c) Did you include the estimated hourly wage paid to participants and the total amount spent on participant compensation? [N/A]

 **Organization of Appendices**

 This paper has four appendices. First, we prove Theorem 1 in Appendix A, together with multiple
 lemmas that the proof of the theorem relies on, and their statements and proofs are provided in
 Appendix A as well. Similarly, the proof for Theorem 2 together with intermediate steps, all are
 presented in Appendix B. Then, Appendix C consists of statements and proofs of other results that
 are used for establishing both theorems. Finally, empirical simulations beyond those presented in
 Section 6 are presented in Appendix D.

## A    Proof of Theorem 1

For analyzing the estimation error, we establish Lemma 4, which under the condition $\kappa \gtrsim \tau^2$
provides that with probability at least $1 - \delta$,

$$\left\| \widehat{\boldsymbol{\theta}} - \boldsymbol{\theta_0} \right\| \lesssim \frac{p \left(p + q\right)^{1/2}}{\tau^{1/2}} \frac{\overline{\lambda}\left(\Sigma_{\mathbb{W}}\right) \vee \sigma_w^2}{\underline{\lambda}\left(\Sigma_{\mathbb{W}}\right) \wedge \sigma_w^2} \left(1 + \|K\|\right)^3 \log\left(\frac{pq\kappa}{\delta}\right).$$

Note that in the proof of Lemma 4, results of Lemmas 1, 2, and 3 are used.

Therefore, Lemma 12 implies that for solutions of (5), with probability at least $1 - \delta$, it holds that

$$\left\| P\left(\widehat{\boldsymbol{\theta}}\right) - P\left(\boldsymbol{\theta_0}\right) \right\| \lesssim \frac{p \left(p + q\right)^{1/2}}{\tau^{1/2}} \frac{\overline{\lambda}\left(\Sigma_{\mathbb{W}}\right) \vee \sigma_w^2}{\underline{\lambda}\left(\Sigma_{\mathbb{W}}\right) \wedge \sigma_w^2} \left(1 + \|K\|\right)^3 \log\left(\frac{pq\kappa}{\delta}\right).$$

Note that we get the same expression as the right-hand-side above, as an upper bound for
$\left\| \widehat{B}^\top P\left(\widehat{\boldsymbol{\theta}}\right) - B_0^\top P\left(\boldsymbol{\theta_0}\right) \right\|$. So, letting

$$\overline{A} = A_0 - B_0 Q_u^{-1} \widehat{B}^\top P\left(\widehat{\boldsymbol{\theta}}\right), \qquad \overline{A}_0 = A_0 - B_0 Q_u^{-1} B_0^\top P\left(\boldsymbol{\theta_0}\right),$$

we obtain

$$\left\| \overline{A} - \overline{A}_0 \right\| \lesssim \sqrt{\frac{p^2 q}{\tau}} \frac{\overline{\lambda}\left(\Sigma_{\mathbb{W}}\right) \vee \sigma_w^2}{\underline{\lambda}\left(\Sigma_{\mathbb{W}}\right) \wedge \sigma_w^2} \left(1 + \|K\|\right)^3 \log\left(\frac{pq\kappa}{\delta}\right), \tag{13}$$

with probability at least $1 - \delta$.

Next, to consider the effect of the above errors on the eigenvalues of $\overline{A}$, we compare them to that
of $\overline{A}_0$. Note that real-parts of all eigenvalues of $\overline{A}_0$ are at most $-\boldsymbol{\zeta_0}$, as defined in (10). So, using
the result and the notation of Lemma 5, for all eigenvalues of $\overline{A}$ being on the open left half-plane it
suffices to have $\Delta_{\overline{A}_0}\left(\overline{A} - \overline{A}_0\right) \leq \boldsymbol{\zeta_0}$. Also, in lights of Lemma 5, suppose that $\boldsymbol{r}$ is the size of the
largest block in the Jordan block-diagonalization of $\overline{A}_0$. So, (36) implies that if

$$\left\| \overline{A} - \overline{A}_0 \right\| \lesssim \boldsymbol{r}^{-1/2} \left(1 \wedge \boldsymbol{\zeta_0}^{\boldsymbol{r}}\right),$$

then Algorithm 1 successfully stabilizes the diffusion process in (1). Thus, (13) shows that the failure
probability of Algorithm 1; $\mathbb{P}(\mathcal{E}_{\boldsymbol{\tau}})$, satisfies

$$\sqrt{\frac{p^2 q}{\tau}} \frac{\overline{\lambda}\left(\Sigma_{\mathbb{W}}\right) \vee \sigma_w^2}{\underline{\lambda}\left(\Sigma_{\mathbb{W}}\right) \wedge \sigma_w^2} \left(1 + \|K\|\right)^3 \log\left(\frac{pq\kappa}{\mathbb{P}(\mathcal{E}_{\boldsymbol{\tau}})}\right) \gtrsim \boldsymbol{r}^{-1/2} \left(1 \wedge \boldsymbol{\zeta_0}^{\boldsymbol{r}}\right).$$

Finally, $\boldsymbol{r} \leq p$ together with $\log\left(pq\kappa\right) \lesssim \tau^{1/2}$, lead to the desired result.

In the remainder of this section, technical lemmas above are stated and their proofs will be provided.

### A.1    Bounding cross products of state and randomization

**Definition 2** *For a set $\mathcal{S}$, let $\mathbb{1}\{\mathcal{S}\}$ be the indicator function that is $1$ on $\mathcal{S}$, and vanishes outside of $\mathcal{S}$.*

**Lemma 1** *In Algorithm 1, for $t \geq 0$, define the piecewise-constant signal $v(t)$ below according to
the randomization sequence $w_n$:*

$$v(t) = \sum_{n=0}^{\kappa-1} \mathbb{1}\left\{\frac{n\tau}{\kappa} \leq t < \frac{(n+1)\tau}{\kappa}\right\} w_n. \tag{14}$$

*Then, with probability at least $1 - \delta$, we have*

$$\left\| \int_0^\tau \boldsymbol{x}_s v(s)^\top \boldsymbol{ds} - \frac{\boldsymbol{\tau}^2}{2\boldsymbol{\kappa}^2} B_0 \sum_{n=0}^{\boldsymbol{\kappa}-1} w_n w_n^\top \right\|$$

$$\lesssim \left( \sigma_w^2 + \overline{\lambda} \left( \Sigma_{\mathbb{W}} \right) \right) \left( 1 + \int_0^\tau \left\| e^{\overline{A}t} \right\| \boldsymbol{dt} \right) \left( pq^{1/2} \boldsymbol{\tau}^{1/2} \log \frac{pq}{\delta} + q \left[ 1 + \frac{\boldsymbol{\tau}^2}{\boldsymbol{\kappa}^2} \right] \frac{\boldsymbol{\tau}}{\boldsymbol{\kappa}^{1/2}} \log^{3/2} \frac{\boldsymbol{\kappa}q}{\delta} \right).$$

Proof. First, after plugging the control signal $\boldsymbol{u}_t$ in (1) and solving the resulting stochastic differential equation, we obtain

$$\boldsymbol{x}_t = e^{\overline{A}t}\boldsymbol{x}_0 + \int_0^t e^{\overline{A}(t-s)} \boldsymbol{d\mathbb{W}}_s + \int_0^t e^{\overline{A}(t-s)} B_0 v(s) \boldsymbol{ds}. \tag{15}$$

This implies that

$$\int_0^\tau \boldsymbol{x}_t v(t)^\top \boldsymbol{dt} = \Phi_1 + \Phi_2 + \Phi_3,$$

where

$$\Phi_1 = \int_0^\tau e^{\overline{A}t}\boldsymbol{x}_0 v(t)^\top \boldsymbol{dt} = \sum_{n=0}^{\boldsymbol{\kappa}-1} \left( \int_{n\boldsymbol{\tau}\boldsymbol{\kappa}^{-1}}^{(n+1)\boldsymbol{\tau}\boldsymbol{\kappa}^{-1}} e^{\overline{A}t} \boldsymbol{dt} \right) \boldsymbol{x}_0 w_n^\top,$$

$$\Phi_2 = \int_0^\tau \int_0^t e^{\overline{A}(t-s)} \boldsymbol{d\mathbb{W}}_s v(t)^\top \boldsymbol{dt} = \sum_{n=0}^{\boldsymbol{\kappa}-1} \left( \int_{n\boldsymbol{\tau}\boldsymbol{\kappa}^{-1}}^{(n+1)\boldsymbol{\tau}\boldsymbol{\kappa}^{-1}} \int_0^t e^{\overline{A}(t-s)} \boldsymbol{d\mathbb{W}}_s \boldsymbol{dt} \right) w_n^\top,$$

$$\Phi_3 = \int_0^\tau \int_0^t e^{\overline{A}(t-s)} B_0 v(s) \boldsymbol{ds} v(t)^\top \boldsymbol{dt}.$$

To analyze $\Phi_1$, we use the fact that every entry of $\Phi_1$ is a normal random variable with mean zero and variance at most

$$\sigma_w^2 \sum_{n=0}^{\boldsymbol{\kappa}-1} \left\| \int_{n\boldsymbol{\tau}\boldsymbol{\kappa}^{-1}}^{(n+1)\boldsymbol{\tau}\boldsymbol{\kappa}^{-1}} e^{\overline{A}t} \boldsymbol{dt} \right\|^2 \|\boldsymbol{x}_0\|^2 \leq \sigma_w^2 \left( \int_0^\tau \left\| e^{\overline{A}t} \right\| \boldsymbol{dt} \right)^2 \|\boldsymbol{x}_0\|^2.$$

Therefore, with probability at least $1 - \delta$, it holds that

$$\|\Phi_1\| \lesssim \sigma_w \left( \int_0^\tau \left\| e^{\overline{A}t} \right\| \boldsymbol{dt} \right) \|\boldsymbol{x}_0\| \sqrt{pq \log\left(\frac{pq}{\delta}\right)}. \tag{16}$$

544 Furthermore, to study $\Phi_2$, Fubini Theorem [31] gives

$$
\int\limits_{n\boldsymbol{\tau}\boldsymbol{\kappa}^{-1}}^{(n+1)\boldsymbol{\tau}\boldsymbol{\kappa}^{-1}} \int\limits_0^t e^{\overline{A}(t-s)}\boldsymbol{d}\mathbb{W}_s \boldsymbol{dt} \;=\; \int\limits_0^{n\boldsymbol{\tau}\boldsymbol{\kappa}^{-1}} \left( \int\limits_{n\boldsymbol{\tau}\boldsymbol{\kappa}^{-1}}^{(n+1)\boldsymbol{\tau}\boldsymbol{\kappa}^{-1}} e^{\overline{A}(t-n\boldsymbol{\tau}\boldsymbol{\kappa}^{-1})}\boldsymbol{dt} \right) e^{\overline{A}(n\boldsymbol{\tau}\boldsymbol{\kappa}^{-1}-s)}\boldsymbol{d}\mathbb{W}_s
$$

$$
+ \int\limits_{n\boldsymbol{\tau}\boldsymbol{\kappa}^{-1}}^{(n+1)\boldsymbol{\tau}\boldsymbol{\kappa}^{-1}} \left( \int\limits_s^{(n+1)\boldsymbol{\tau}\boldsymbol{\kappa}^{-1}} e^{\overline{A}(t-s)}\boldsymbol{dt} \right)\boldsymbol{d}\mathbb{W}_s
$$

$$
= \; F \sum_{m=1}^n e^{\overline{A}(n-m)\boldsymbol{\tau}\boldsymbol{\kappa}^{-1}} \int\limits_{(m-1)\boldsymbol{\tau}\boldsymbol{\kappa}^{-1}}^{m\boldsymbol{\tau}\boldsymbol{\kappa}^{-1}} e^{\overline{A}(m\boldsymbol{\tau}\boldsymbol{\kappa}^{-1}-s)}\boldsymbol{d}\mathbb{W}_s
$$

$$
+ \int\limits_{n\boldsymbol{\tau}\boldsymbol{\kappa}^{-1}}^{(n+1)\boldsymbol{\tau}\boldsymbol{\kappa}^{-1}} G_s \boldsymbol{d}\mathbb{W}_s
$$

545 where the matrix $F = \int\limits_{n\boldsymbol{\tau}\boldsymbol{\kappa}^{-1}}^{(n+1)\boldsymbol{\tau}\boldsymbol{\kappa}^{-1}} e^{\overline{A}(t-n\boldsymbol{\tau}\boldsymbol{\kappa}^{-1})}\boldsymbol{dt}$, does not depend on $s$ or $n$, and the matrix

546 $G_s = \int\limits_s^{(n+1)\boldsymbol{\tau}\boldsymbol{\kappa}^{-1}} e^{\overline{A}(t-s)}\boldsymbol{dt}$, does not depend on $n$, since $n\boldsymbol{\tau}\boldsymbol{\kappa}^{-1} \leq s \leq (n+1)\boldsymbol{\tau}\boldsymbol{\kappa}^{-1}$.

547 So, letting $e_i, i = 1, \cdots, p$, be the standard basis of the Euclidean space, conditioned on the Wiener
548 process $\{\mathbb{W}_s\}_{s\geq 0}$, for every $j = 1, \cdots, q$, the coordinate $j$ of $e_i^\top \Phi_2$ is a mean zero normal random
549 variable. Thus, given $\{\mathbb{W}_s\}_{s\geq 0}$, with probability at least $1 - \delta$, it holds that

$$
\left(e_i^\top \Phi_2 e_j\right)^2 \lesssim \mathrm{var}\left(e_i^\top \Phi_2 e_j \Big| \mathcal{F}\left(\mathbb{W}_{0:\boldsymbol{\tau}}\right)\right) \log \frac{1}{\delta}.
$$

550 Now, to calculate the conditional variance, we can write

$$
\frac{\mathrm{var}\left(e_i^\top \Phi_2 e_j \Big| \mathcal{F}\left(\mathbb{W}_{0:\boldsymbol{\tau}}\right)\right)}{\sigma_w^2} = \sum_{n=0}^{\boldsymbol{\kappa}-1} \left[ e_i^\top \int\limits_{n\boldsymbol{\tau}\boldsymbol{\kappa}^{-1}}^{(n+1)\boldsymbol{\tau}\boldsymbol{\kappa}^{-1}} \int\limits_0^t e^{\overline{A}(t-s)}\boldsymbol{d}\mathbb{W}_s \boldsymbol{dt} \right]^2 \lesssim \sum_{n=1}^{\boldsymbol{\kappa}-1} \left[ \left(\sum_{m=1}^n \beta_{m,n}\right)^2 + \alpha_n^2 \right],
$$

551 where

$$
\beta_{m,n} = e_i^\top F e^{\overline{A}(n-m)\boldsymbol{\tau}\boldsymbol{\kappa}^{-1}} \int\limits_{(m-1)\boldsymbol{\tau}\boldsymbol{\kappa}^{-1}}^{m\boldsymbol{\tau}\boldsymbol{\kappa}^{-1}} e^{\overline{A}(m\boldsymbol{\tau}\boldsymbol{\kappa}^{-1}-s)}\boldsymbol{d}\mathbb{W}_s,
$$

$$
\alpha_n = e_i^\top \int\limits_{n\boldsymbol{\tau}\boldsymbol{\kappa}^{-1}}^{(n+1)\boldsymbol{\tau}\boldsymbol{\kappa}^{-1}} G_s \boldsymbol{d}\mathbb{W}_s.
$$

552 To proceed, define the matrix $H = [H_{n,m}]$, where for $1 \leq m, n \leq \boldsymbol{\kappa} - 1$, every block $H_{n,m} \in \mathbb{R}^{1\times p}$
553 is

$$
H_{n,m} = e_i^\top F e^{\overline{A}(n-m)\boldsymbol{\tau}\boldsymbol{\kappa}^{-1}},
$$

554 for $m \leq n$, and is 0 for $m > n$. Then, denote

$$
\Gamma = \begin{bmatrix} \int\limits_0^{\boldsymbol{\tau}\boldsymbol{\kappa}^{-1}} e^{\overline{A}(\boldsymbol{\tau}\boldsymbol{\kappa}^{-1}-s)}\boldsymbol{d}\mathbb{W}_s \\ \int\limits_{\boldsymbol{\tau}\boldsymbol{\kappa}^{-1}}^{2\boldsymbol{\tau}\boldsymbol{\kappa}^{-1}} e^{\overline{A}(2\boldsymbol{\tau}\boldsymbol{\kappa}^{-1}-s)}\boldsymbol{d}\mathbb{W}_s \\ \vdots \\ \int\limits_{(\boldsymbol{\kappa}-1)\boldsymbol{\tau}\boldsymbol{\kappa}^{-1}}^{\boldsymbol{\tau}} e^{\overline{A}(\boldsymbol{\tau}-s)}\boldsymbol{d}\mathbb{W}_s \end{bmatrix} \in \mathbb{R}^{p(\boldsymbol{\kappa}-1)\times 1},
$$

to get

$$\sum_{n=0}^{\kappa-1}\left(\sum_{m=1}^{n}\beta_{m,n}^2\right) = \|H\Gamma\|^2 \le \overline{\lambda}\left(H^\top H\right)\|\Gamma\|^2.$$

Now, for the matrix $H$, we have [42, 43]:

$$\overline{\lambda}\left(H^\top H\right) \lesssim \left(\sum_{n=1}^{\kappa-1}\|H_{n,1}\|\right)^2 \lesssim \left(\tau\kappa^{-1}\sum_{n=1}^{\kappa}e^{\overline{A}n\tau\kappa^{-1}}\right)^2 \lesssim \left(\int_0^\tau\left\|e^{\overline{A}t}\right\|dt\right)^2.$$

Note that thanks to the independent increments of the Wiener process, the blocks of $\Gamma$ are statistically independent. Further, by Ito Isometry [31], every block of $\Gamma$ is a mean-zero normally distributed vector with the covariance matrix

$$\int_0^{\tau\kappa^{-1}} e^{\overline{A}(\tau\kappa^{-1}-s)}\Sigma_{\mathbb{W}}e^{\overline{A}^\top(\tau\kappa^{-1}-s)}ds.$$

So, according to the exponential inequalities for quadratic forms of normally distributed random variables [39], it holds with probability at least $1-\delta$, that

$$\|\Gamma\|^2 \lesssim p\kappa\overline{\lambda}\left(\Sigma_{\mathbb{W}}\right)\left(\tau\kappa^{-1}\right)\log\frac{1}{\delta}.$$

Thus, with probability at least $1-\delta$, we have

$$\sum_{n=0}^{\kappa-1}\left(\sum_{m=1}^{n}\beta_{m,n}^2\right) \lesssim \left(\int_0^\tau\left\|e^{\overline{A}t}\right\|dt\right)^2 p\overline{\lambda}\left(\Sigma_{\mathbb{W}}\right)\tau\log\frac{1}{\delta}.$$

Similarly, the bound above can be shown for $\sum_{n=1}^{\kappa-1}\alpha_n^2$. Hence, we obtain the corresponding high probability bound for a single entry $\mathrm{e}_i^\top\Phi_2\mathrm{e}_j$ of $\Phi_2$, which together with a union bound, implies that

$$\|\Phi_2\| \lesssim \sigma_w pq^{1/2}\left(\int_0^\tau\left\|e^{\overline{A}t}\right\|dt\right)\overline{\lambda}\left(\Sigma_{\mathbb{W}}\right)^{1/2}\tau^{1/2}\log\left(\frac{pq}{\delta}\right), \qquad (17)$$

with probability at least $1-\delta$.

Next, according to Fubini Theorem, $\Phi_3$ can also be written as

$$\Phi_3 = \int_0^\tau\int_0^s e^{\overline{A}(s-t)}B_0 v(t)v(s)^\top dtds = \int_0^\tau\int_t^\tau e^{\overline{A}(s-t)}B_0 v(t)v(s)^\top dsdt.$$

Thus, we have

$$2\Phi_3 = \int_0^\tau\int_0^\tau e^{\overline{A}|t-s|}B_0 v(t\wedge s)v(s\vee t)^\top dtds.$$

Recall that the signal $v(t)$ in (14) is piecewise-constant, with values determined by the randomization sequence $w_n$. So, the above double integral can be written as a double sum

$$2\Phi_3 = \sum_{n=0}^{\kappa-1}\sum_{m=0}^{\kappa-1}\left(\int_{n\tau\kappa^{-1}}^{(n+1)\tau\kappa^{-1}}\int_{m\tau\kappa^{-1}}^{(m+1)\tau\kappa^{-1}} e^{\overline{A}|t-s|}dsdt\right)B_0 w_{m\wedge n}w_{m\vee n}^\top$$

$$= \sum_{n=0}^{\kappa-1}\sum_{m=0}^{\kappa-1}\left(e^{\overline{A}|m-n|\tau\kappa^{-1}}\int_0^{\tau\kappa^{-1}}\int_0^{\tau\kappa^{-1}} e^{\overline{A}|t-s|}dsdt\right)B_0 w_{m\wedge n}w_{m\vee n}^\top.$$

Thus, we have

$$2\Phi_3 - \frac{\boldsymbol{\tau}^2}{\boldsymbol{\kappa}^2} B_0 \sum_{n=0}^{\boldsymbol{\kappa}-1} w_n w_n^\top = \Phi_4 + \Phi_5, \tag{18}$$

for

$$\Phi_4 = \left( \int_0^{\boldsymbol{\tau\kappa}^{-1}} \int_0^{\boldsymbol{\tau\kappa}^{-1}} e^{\overline{A}|t-s|} \boldsymbol{ds dt} - \boldsymbol{\tau}^2 \boldsymbol{\kappa}^{-2} I_q \right) B_0 \sum_{n=0}^{\boldsymbol{\kappa}-1} w_n w_n^\top,$$

$$\Phi_5 = 2 \left( \int_0^{\boldsymbol{\tau\kappa}^{-1}} \int_0^{\boldsymbol{\tau\kappa}^{-1}} e^{\overline{A}|t-s|} \boldsymbol{ds dt} \right) \sum_{n=0}^{\boldsymbol{\kappa}-1} \sum_{m=n+1}^{\boldsymbol{\kappa}-1} \left( e^{\overline{A}(m-n)\boldsymbol{\tau\kappa}^{-1}} B_0 w_n w_m^\top \right).$$

To proceed, we use the following concentration inequality for random matrices with martingale difference structures, titled as Matrix Azuma inequality [40].

**Theorem 3** *Let* $\{\Psi_n\}_{n=1}^k$ *be a* $d_1 \times d_2$ *martingale difference sequence. That is, for some filtration* $\{\mathcal{F}_n\}_{n=0}^k$, *the matrix* $\Psi_n$ *is* $\mathcal{F}_n$*-measurable, and* $\mathbb{E}\left[ \Psi_n \middle| \mathcal{F}_{n-1} \right] = 0$. *Suppose that* $\|\Psi_n\| \leq \sigma_n$, *for some fixed sequence* $\{\sigma_n\}_{n=1}^k$. *Then, with probability at least* $1 - \delta$, *we have*

$$\left\| \sum_{n=1}^k \Psi_n \right\|^2 \lesssim \left( \sum_{n=1}^k \sigma_n^2 \right) \log \frac{d_1 + d_2}{\delta}.$$

So, to study $\Phi_4$, we apply Theorem 3 to the random matrices $\Psi_n = w_n w_n^\top - \sigma_w^2 I_q$, using the trivial filtration and the high probability upper-bounds for $\|\Psi_n\| \leq \|w_n\|^2 + \sigma_w^2$;

$$\|\Psi_n\| \leq \sigma_n = \sigma_w^2 \left( 1 + q \log \frac{q\boldsymbol{\kappa}}{\delta} \right),$$

as well as the fact

$$\left\| \int_0^{\boldsymbol{\tau\kappa}^{-1}} \int_0^{\boldsymbol{\tau\kappa}^{-1}} \left( e^{\overline{A}|t-s|} - I_q \right) \boldsymbol{ds dt} \right\| \lesssim \boldsymbol{\tau}^3 \boldsymbol{\kappa}^{-3},$$

to obtain the following bound, which holds with probability at least $1 - \delta$:

$$\|\Phi_4\| \lesssim \|B_0\| \sigma_w^2 \boldsymbol{\tau}^3 \boldsymbol{\kappa}^{-2} \left( 1 + \frac{q}{\boldsymbol{\kappa}^{1/2}} \log^{3/2} \frac{\boldsymbol{\kappa} q}{\delta} \right). \tag{19}$$

On the other hand, to establish an upper-bound for $\Phi_5$, consider the random matrices

$$\Psi_n = \sum_{m=n+1}^{\boldsymbol{\kappa}-1} \left( e^{\overline{A}(m-n)\boldsymbol{\tau\kappa}^{-1}} B_0 w_n w_m^\top \right),$$

subject to the natural filtration they generate, and apply Theorem 3, using the bounds

$$\|\Psi_n\| \leq \sigma_n \lesssim \boldsymbol{\tau}^{-1} \boldsymbol{\kappa} \left( \int_0^{\boldsymbol{\tau}} \left\| e^{\overline{A}t} \right\| \boldsymbol{dt} \right) \|B_0\| \sigma_w^2 q \log \frac{\boldsymbol{\kappa} q}{\delta},$$

together with

$$\left\| \int_0^{\boldsymbol{\tau\kappa}^{-1}} \int_0^{\boldsymbol{\tau\kappa}^{-1}} e^{\overline{A}|t-s|} \boldsymbol{ds dt} \right\| \lesssim \boldsymbol{\tau}^2 \boldsymbol{\kappa}^{-2}.$$

Therefore, Theorem 3 indicates that with probability at least $1 - \delta$, it holds that

$$\Phi_5 \lesssim \frac{\boldsymbol{\tau}}{\boldsymbol{\kappa}^{1/2}} \left( \int_0^{\boldsymbol{\tau}} \left\| e^{\overline{A}t} \right\| \boldsymbol{dt} \right) \|B_0\| \sigma_w^2 q \log^{3/2} \frac{\boldsymbol{\kappa} q}{\delta}. \tag{20}$$

Finally, put (16), (17), (18), (19), and (20) together, to get the desired result.

$\blacksquare$

## A.2 Bounding cross products of state and Wiener process

**Lemma 2** *In Algorithm 1, with probability at least $1 - \delta$, we have*

$$\left\| \int_0^t \boldsymbol{x}_s d\mathbb{W}_s^\top \right\| \lesssim \left( \int_0^\tau \left\| e^{\overline{A}t} \right\| dt \right) \left( \overline{\lambda}\left(\Sigma_\mathbb{W}\right) + \sigma_w^2 \right) p \left(p+q\right)^{1/2} \boldsymbol{\tau}^{1/2} \log\left(\frac{pq}{\delta}\right).$$

Proof. First, according to (15), we can write

$$\int_0^\tau \boldsymbol{x}_t d\mathbb{W}_t^\top = \Phi_1 + \Phi_2 + \Phi_3,$$

where

$$\Phi_1 \quad = \quad \int_0^\tau e^{\overline{A}t}\boldsymbol{x}_0 d\mathbb{W}_t^\top, \tag{21}$$

$$\Phi_2 \quad = \quad \int_0^\tau \int_0^t e^{\overline{A}(t-s)} B_0 v(s) ds d\mathbb{W}_t^\top, \tag{22}$$

$$\Phi_3 \quad = \quad \int_0^\tau \int_0^t e^{\overline{A}(t-s)} d\mathbb{W}_s d\mathbb{W}_t^\top. \tag{23}$$

Now, according to Ito Isometry [31], similar to (16), we have

$$\left\| \Phi_1 \right\| \lesssim \overline{\lambda}\left(\Sigma_\mathbb{W}\right)^{1/2} \left( \int_0^\tau \left\| e^{\overline{A}t} \right\| dt \right) \|\boldsymbol{x}_0\| \sqrt{pq \log\left(\frac{pq}{\delta}\right)}, \tag{24}$$

with probability at least $1 - \delta$. Moreover, in a procedure similar to the one that lead to (17), one can show that with probability at least $1 - \delta$, it holds that

$$\left\| \Phi_2 \right\| \lesssim \left( \int_0^\tau \left\| e^{\overline{A}t} \right\| dt \right) \overline{\lambda}\left(\Sigma_\mathbb{W}\right)^{1/2} \sigma_w pq^{1/2} \boldsymbol{\tau}^{1/2} \log\left(\frac{pq}{\delta}\right). \tag{25}$$

Therefore, we need to find a similar upper-bound for $\Phi_3$. To that end, Ito formula provides

$$\boldsymbol{d}\left(e^{-\overline{A}s}\mathbb{W}_s\right) = -\overline{A}e^{-\overline{A}s}\mathbb{W}_s \boldsymbol{ds} + e^{-\overline{A}s}\boldsymbol{d}\mathbb{W}_s.$$

Therefore, integration gives

$$\int_0^t e^{-\overline{A}s}\boldsymbol{d}\mathbb{W}_s = e^{-\overline{A}t}\mathbb{W}_t + \overline{A}\int_0^t e^{-\overline{A}s}\mathbb{W}_s \boldsymbol{ds},$$

which after rearranging and letting $\Psi_t = \int_0^t e^{\overline{A}(t-s)}\boldsymbol{d}\mathbb{W}_s$, leads to

$$\Psi_t \mathbb{W}_t^\top = \left( \int_0^t e^{\overline{A}(t-s)}\boldsymbol{d}\mathbb{W}_s \right) \mathbb{W}_t^\top = \mathbb{W}_t \mathbb{W}_t^\top + \overline{A}\left( \int_0^t e^{\overline{A}(t-s)}\mathbb{W}_s \boldsymbol{ds} \right) \mathbb{W}_t^\top.$$

Now, since $d\Psi_t = d\mathbb{W}_t$, Ito Isometry [31] implies that $d\Psi_t d\mathbb{W}_t^\top = \Sigma_\mathbb{W} dt$. So, apply integration by part and use the above equation to get

$$
\begin{aligned}
\Phi_3 = \int_0^\tau \Psi_t d\mathbb{W}_t^\top &= \int_0^\tau d(\Psi_t \mathbb{W}_t^\top) - \left(\int_0^\tau \mathbb{W}_t d\Psi_t^\top\right)^\top - \int_0^\tau d\Psi_t d\mathbb{W}_t^\top \\
&= \Psi_\tau \mathbb{W}_\tau^\top - \left(\int_0^\tau \mathbb{W}_t d\mathbb{W}_t^\top\right)^\top - \Sigma_\mathbb{W}\tau \\
&= \mathbb{W}_\tau \mathbb{W}_\tau^\top + \overline{A}\left(\int_0^\tau e^{\overline{A}(\tau-s)}\mathbb{W}_s ds\right)\mathbb{W}_\tau^\top - \left(\int_0^\tau \mathbb{W}_t d\mathbb{W}_t^\top\right)^\top - \Sigma_\mathbb{W}\tau.
\end{aligned}
$$

Therefore, every entry of $\Phi_3$ is a quadratic function of the normally distributed random vectors $\mathbb{W}_\tau, \int_0^\tau e^{\overline{A}(\tau-s)}\mathbb{W}_s ds$. Note that we used the fact that

$$
\mathbb{W}_\tau \mathbb{W}_\tau^\top = \int_0^\tau d(\mathbb{W}_t \mathbb{W}_t^\top) = \left(\int_0^\tau \mathbb{W}_t d\mathbb{W}_t^\top\right)^\top + \left(\int_0^\tau \mathbb{W}_t d\mathbb{W}_t^\top\right) + \Sigma_\mathbb{W}\tau.
$$

Thus, exponential inequalities for quadratic forms of normal random vectors [39] imply that for all $i, j = 1, \cdots, p$, it holds that

$$
\left(e_i^\top \Phi_3 e_j\right)^2 \lesssim p\mathbb{E}\left[\left(e_i^\top \Phi_3 e_j\right)^2\right]\log^2\frac{1}{\delta}, \tag{26}
$$

since $\mathbb{E}\left[e_i^\top \Phi_3 e_j\right] = 0$. So, it suffices to find the expectation in (26). For that purpose, we use Ito Isometry [31] to obtain:

$$
\begin{aligned}
\mathbb{E}\left[\left(e_i^\top \Phi_3 e_j\right)^2\right] &= \mathbb{E}\left[\left(\int_0^\tau e_i^\top \Psi_t e_j^\top \Sigma_\mathbb{W}^{1/2} d\left(\Sigma_\mathbb{W}^{-1/2}\mathbb{W}_t\right)\right)^2\right] = \mathbb{E}\left[\int_0^\tau \left\|e_i^\top \Psi_t \Sigma_\mathbb{W}^{1/2} e_j\right\|^2 dt\right] \\
&\leq e_j^\top \Sigma_\mathbb{W} e_j \mathbb{E}\left[\int_0^\tau \left(e_i^\top \Psi_t\right)^2 dt\right] = e_j^\top \Sigma_\mathbb{W} e_j \mathbb{E}\left[\int_0^\tau \left(e_i^\top \int_0^t e^{\overline{A}(t-s)} d\mathbb{W}_s\right)^2 dt\right].
\end{aligned}
$$

To proceed with the above expression, apply Fubini Theorem [31] to interchange the expected value with the integral, and then use Ito Isometry again:

$$
\begin{aligned}
\mathbb{E}\left[\int_0^\tau \left(e_i^\top \int_0^t e^{\overline{A}(t-s)} d\mathbb{W}_s\right)^2 dt\right] &= \int_0^\tau \mathbb{E}\left[\left(e_i^\top \int_0^t e^{\overline{A}(t-s)}\Sigma_\mathbb{W}^{1/2} d\left(\Sigma_\mathbb{W}^{-1/2}\mathbb{W}_s\right)\right)^2\right] dt \\
&= \int_0^\tau e_i^\top \left(\int_0^t e^{\overline{A}(t-s)}\Sigma_\mathbb{W} e^{\overline{A}^\top(t-s)} ds\right) e_i dt \\
&\leq e_i^\top \left(\int_0^\tau e^{\overline{A}s}\Sigma_\mathbb{W} e^{\overline{A}^\top s} ds\right) e_i \tau.
\end{aligned}
$$

Therefore, (26) yields to

$$\|\Phi_3\|^2 \leq \sum_{i,j=1}^{p} \left(\mathrm{e}_i^\top \Phi_3 \mathrm{e}_j\right)^2 \;\lesssim\; \sum_{i,j=1}^{p}\left[\mathrm{e}_j^\top \Sigma_{\mathbb{W}} \mathrm{e}_j \mathrm{e}_i^\top \left(\int_0^{\boldsymbol{\tau}} e^{\overline{A}s}\Sigma_{\mathbb{W}}e^{\overline{A}^\top s}\boldsymbol{d}s\right)\mathrm{e}_i\right]\boldsymbol{\tau} p \log^2 \frac{p}{\delta}$$

$$= \;\; \mathbf{tr}\left(\Sigma_{\mathbb{W}}\right)\mathbf{tr}\left(\int_0^{\boldsymbol{\tau}} e^{\overline{A}s}\Sigma_{\mathbb{W}}e^{\overline{A}^\top s}\boldsymbol{d}s\right) p\boldsymbol{\tau}\log^2 \frac{p}{\delta}$$

$$\lesssim \;\; \mathbf{tr}\left(\Sigma_{\mathbb{W}}\right)^2 \left(\int_0^{\boldsymbol{\tau}}\left\|e^{\overline{A}s}\right\|\boldsymbol{d}s\right)^2 p\boldsymbol{\tau}\log^2 \frac{p}{\delta}. \tag{27}$$

Finally, putting (24), (25), and (27) together, we obtain the desired result.

∎

## A.3   Concentration of normal posterior distribution in Algorithm 1

**Lemma 3** *In Algorithm 1, letting $\overline{A} = A_0 + B_0 K$, suppose that*

$$\boldsymbol{\tau} \;\gtrsim\; \left(\int_0^{\boldsymbol{\tau}}\left\|\exp(\overline{A}s)\right\|^2 \boldsymbol{d}s\right)\left(\overline{\lambda}\left(\Sigma_{\mathbb{W}}\right)+\sigma_w^2\|B_0\|^2\right)(p+q)\log\frac{1}{\delta}, \tag{28}$$

$$\frac{\boldsymbol{\kappa}}{\boldsymbol{\tau}} \;\gtrsim\; \frac{\sigma_w^2}{\sigma_w^2 \wedge \underline{\lambda}\left(\Sigma_{\mathbb{W}}\right)}\;\|B_0\|\left(1\vee\|K\|\right)q\log\frac{\boldsymbol{\kappa}q}{\delta}. \tag{29}$$

*Then, for the matrix $\widehat{\Sigma}_{\boldsymbol{\tau}}$ in (8), with probability at least $1-\delta$ we have*

$$\underline{\lambda}\left(\widehat{\Sigma}_{\boldsymbol{\tau}}\right) \;\gtrsim\; \boldsymbol{\tau}\left(\underline{\lambda}\left(\Sigma_{\mathbb{W}}\right)\wedge\sigma_w^2\right)\left(1+\|K\|^2\right)^{-1}.$$

Proof. First, we can write the control action in (7) as $\boldsymbol{u}_t = K\boldsymbol{x}_t + v(t)$, for the piecewise-constant signal $v(t)$ in (14). Then, the dynamics in (1) provides

$$\boldsymbol{dx}_t = \left(\overline{A}\boldsymbol{x}_t + B_0 v(t)\right)\boldsymbol{dt} + \boldsymbol{d}\mathbb{W}_t.$$

Therefore, similar to (15), one can solve the above stochastic differential equation to get

$$\boldsymbol{x}_t = e^{\overline{A}t}\boldsymbol{x}_0 + \int_0^t e^{\overline{A}(t-s)}\boldsymbol{d}\mathbb{W}_s + \int_0^t e^{\overline{A}(t-s)}B_0 v(s)\boldsymbol{ds}.$$

So, using the exponential inequalities for quadratic forms [39], with probability at least $1-\delta$, it holds that

$$\left\|\boldsymbol{x}_{\boldsymbol{\tau}} - e^{\overline{A}\boldsymbol{\tau}}\boldsymbol{x}_0\right\|^2 \;\lesssim\; \overline{\lambda}\left(\int_0^{\boldsymbol{\tau}} e^{\overline{A}s}\Sigma_{\mathbb{W}}e^{\overline{A}^\top s}\boldsymbol{d}s + \sigma_w^2\sum_{n=0}^{\boldsymbol{\kappa}-1} J_n B_0 B_0^\top J_n^\top\right)\left(p+p^{1/2}\log\frac{1}{\delta}\right), \tag{30}$$

where

$$J_n = \int_{n\boldsymbol{\tau}\boldsymbol{\kappa}^{-1}}^{(n+1)\boldsymbol{\tau}\boldsymbol{\kappa}^{-1}} e^{\overline{A}s}\boldsymbol{ds}.$$

Furthermore, an application of Ito calculus [31] leads to $\boldsymbol{dx}_t \boldsymbol{dx}_t^\top = \boldsymbol{d}\mathbb{W}_t \boldsymbol{d}\mathbb{W}_t^\top = \Sigma_{\mathbb{W}}\boldsymbol{dt}$. Now, by defining the matrix valued processes

$$\Phi_t = \int_0^t \boldsymbol{x}_s \boldsymbol{x}_s^\top \boldsymbol{ds}, \qquad M_t = \int_0^t \boldsymbol{x}_s \boldsymbol{d}\mathbb{W}_s^\top + \int_0^t \boldsymbol{x}_s v(s)^\top B_0^\top \boldsymbol{ds},$$

we obtain

$$
\begin{aligned}
\boldsymbol{d}\left(\boldsymbol{x}_t \boldsymbol{x}_t^\top\right) &= \boldsymbol{x}_t \boldsymbol{dx}_t^\top + \boldsymbol{dx}_t \boldsymbol{x}_t^\top + \boldsymbol{dx}_t \boldsymbol{dx}_t^\top \\
&= \boldsymbol{x}_t \left(\left(\overline{A}\boldsymbol{x}_t + B_0 v(t)\right) \boldsymbol{dt} + \boldsymbol{d}\mathbb{W}_t\right)^\top \\
&\quad + \left(\left(\overline{A}\boldsymbol{x}_t + B_0 v(t)\right) \boldsymbol{dt} + \boldsymbol{d}\mathbb{W}_t\right) \boldsymbol{x}_t^\top + \Sigma_\mathbb{W} \boldsymbol{dt} \\
&= \boldsymbol{d}\Phi_t \overline{A}^\top + \overline{A} \boldsymbol{d}\Phi_t + \boldsymbol{d}M_t + \boldsymbol{d}M_t^\top + \Sigma_\mathbb{W} \boldsymbol{dt}.
\end{aligned}
$$

Thus, after integrating both sides of the above equality, we obtain

$$
\Phi_t \overline{A}^\top + \overline{A}\Phi_t + M_t + M_t^\top + t\Sigma_\mathbb{W} + \boldsymbol{x}_0 \boldsymbol{x}_0^\top - \boldsymbol{x}_t \boldsymbol{x}_t^\top = 0.
$$

Because all eigenvalues of $\overline{A}$ are in the open left half-plane, we can solve the above equation for $\Phi_t$, to get

$$
\Phi_t = \int_0^\infty \exp\left(\overline{A}s\right) \left[M_t + M_t^\top + t\Sigma_\mathbb{W} + \boldsymbol{x}_0 \boldsymbol{x}_0^\top - \boldsymbol{x}_t \boldsymbol{x}_t^\top\right] \exp\left(\overline{A}^\top s\right) \boldsymbol{ds}. \tag{31}
$$

Next, putting Lemma 1, Lemma 2, and (30) together, as long as (28) holds, with probability at least $1 - \delta$ we have

$$
\underline{\lambda}\left(M_\tau + M_\tau^\top + \tau\Sigma_\mathbb{W} + \boldsymbol{x}_0 \boldsymbol{x}_0^\top - \boldsymbol{x}_\tau \boldsymbol{x}_\tau^\top\right) \gtrsim \tau\underline{\lambda}\left(\Sigma_\mathbb{W}\right).
$$

Thus, (31) implies that $\underline{\lambda}\left(\Phi_\tau\right) \gtrsim \tau\underline{\lambda}\left(\Sigma_\mathbb{W}\right)$. To proceed, consider the matrix $\widehat{\Sigma}_\tau$ in (8), which comprises two signals $\boldsymbol{x}_t, v(t)$. The empirical covariance matrix of the state signal is studied above, while for the piecewise-constant randomization signal $v(t)$ in (14), we have

$$
\int_0^t v(s)v(s)^\top \boldsymbol{ds} = \sum_{n=0}^{\kappa-1} \int_{n\tau\kappa^{-1}}^{(n+1)\tau\kappa^{-1}} w_n w_n^\top \boldsymbol{ds} = \tau\kappa^{-1} \sum_{n=0}^{\kappa-1} w_n w_n^\top.
$$

Thus, according to Theorem 3, similar to (19) we have

$$
\left\|\sum_{n=0}^{\kappa-1} w_n w_n^\top - \kappa\sigma_w^2 I_q\right\| \lesssim \kappa^{1/2}\sigma_w^2 q \log^{3/2} \frac{\kappa q}{\delta},
$$

with probability at least $1 - \delta$, which for

$$
H_\tau = \int_0^\tau \begin{bmatrix} 0_p \\ v(s) \end{bmatrix} \begin{bmatrix} 0_p \\ v(s) \end{bmatrix}^\top \boldsymbol{ds} - \tau\sigma_w^2 \begin{bmatrix} 0_{p\times p} & 0_{p\times q} \\ 0_{q\times p} & I_q \end{bmatrix},
$$

leads to

$$
\|H_\tau\| \lesssim \sigma_w^2 q \log^{3/2} \frac{\kappa q}{\delta}, \tag{32}
$$

because $\kappa \gtrsim \tau^2$.

Next, using $\boldsymbol{z}_s = \left[\boldsymbol{x}_s^\top, \boldsymbol{x}_s^\top K^\top + v(s)^\top\right]^\top$, the matrix $\widehat{\Sigma}_\tau$ can be written as

$$
\widehat{\Sigma}_\tau = \begin{bmatrix} I_p \\ K \end{bmatrix} \Phi_\tau \begin{bmatrix} I_p \\ K \end{bmatrix}^\top + \tau\sigma_w^2 \begin{bmatrix} 0_{p\times p} & 0_{p\times q} \\ 0_{q\times p} & I_q \end{bmatrix} + F_\tau + H_\tau, \tag{33}
$$

where

$$
F_\tau = \int_0^\tau \left(\begin{bmatrix} I_p \\ K \end{bmatrix} \boldsymbol{x}_s \begin{bmatrix} 0_p \\ v(s) \end{bmatrix}^\top + \begin{bmatrix} 0_p \\ v(s) \end{bmatrix} \boldsymbol{x}_s^\top \begin{bmatrix} I_p \\ K \end{bmatrix}^\top\right) \boldsymbol{ds}.
$$

However, Lemma 1 and $\kappa \gtrsim \tau^2$ give a high probability upper-bound for the above matrix:

$$
\|F_\tau\| \lesssim (1 \vee \|K\|)\left(\overline{\lambda}\left(\Sigma_\mathbb{W}\right) + \sigma_w^2\right)\left(pq^{1/2}\tau^{1/2}\log\frac{pq}{\delta} + q\log^{3/2}\frac{\kappa q}{\delta}\right). \tag{34}
$$

In the sequel, we show that with probability at least $1 - \delta$, it holds that

$$\underline{\lambda}\left(\begin{bmatrix} I_p \\ K \end{bmatrix} \Phi_{\boldsymbol{\tau}} \begin{bmatrix} I_p \\ K \end{bmatrix}^{\top} + \boldsymbol{\tau}\sigma_w^2 \begin{bmatrix} 0_{p\times p} & 0_{p\times q} \\ 0_{q\times p} & I_q \end{bmatrix}\right) \gtrsim \boldsymbol{\tau}\left(\underline{\lambda}\left(\Sigma_{\mathbb{W}}\right) \wedge \sigma_w^2\right)\left(1 + \|K\|^2\right)^{-1},$$

which, according to (32), (33), and (34), implies the desired result. To show the above least eigenvalue inequality, we use $\underline{\lambda}\left(\Phi_{\boldsymbol{\tau}}\right) \gtrsim \boldsymbol{\tau}\underline{\lambda}\left(\Sigma_{\mathbb{W}}\right)$ to obtain

$$\underline{\lambda}\left(\begin{bmatrix} I_p \\ K \end{bmatrix} \Phi_{\boldsymbol{\tau}} \begin{bmatrix} I_p \\ K \end{bmatrix}^{\top} + \boldsymbol{\tau}\sigma_w^2 \begin{bmatrix} 0_{p\times p} & 0_{p\times q} \\ 0_{q\times p} & I_q \end{bmatrix}\right) \gtrsim \boldsymbol{\tau}\left(\underline{\lambda}\left(\Sigma_{\mathbb{W}}\right) \wedge \sigma_w^2\right)\underline{\lambda}\left(\begin{bmatrix} I_p & K^{\top} \\ K & KK^{\top} + I_q \end{bmatrix}\right).$$

However, block matrix inversion gives

$$\underline{\lambda}\left(\begin{bmatrix} I_p & K^{\top} \\ K & KK^{\top} + I_q \end{bmatrix}\right) = \overline{\lambda}\left(\begin{bmatrix} I_p & K^{\top} \\ K & KK^{\top} + I_q \end{bmatrix}^{-1}\right)^{-1} = \overline{\lambda}\left(\begin{bmatrix} K^{\top}K + I_p & -K^{\top} \\ -K & I_q \end{bmatrix}\right)^{-1},$$

that is clearly at least $\left(1 + \|K\|^2\right)^{-1}$, apart from a constant factor. Therefore, we get the desired result. ∎

### A.4 Approximation of true drift parameter by Algorithm 1

**Lemma 4** *Suppose that $\widehat{\boldsymbol{\theta}}$ is given by Algorithm 1. Then, with probability at least $1 - \delta$, we have*

$$\left\|\widehat{\boldsymbol{\theta}} - \boldsymbol{\theta_0}\right\| \lesssim \frac{p\left(p + q\right)^{1/2}}{\boldsymbol{\tau}^{1/2}} \frac{\overline{\lambda}\left(\Sigma_{\mathbb{W}}\right) \vee \sigma_w^2}{\underline{\lambda}\left(\Sigma_{\mathbb{W}}\right) \wedge \sigma_w^2} \left(1 + \|K\|\right)^3 \log\left(\frac{pq\boldsymbol{\kappa}}{\delta}\right). \tag{35}$$

Proof.    First, consider the mean matrix of the Gaussian posterior distribution. Using the data generation mechanism $\boldsymbol{dx}_t = \boldsymbol{\theta_0}^{\top}\boldsymbol{z}_t dt + \boldsymbol{d}\mathbb{W}_t$, we have

$$\widehat{\mu}_{\boldsymbol{\tau}} = \widehat{\Sigma}_{\boldsymbol{\tau}}^{-1}\int_0^{\boldsymbol{\tau}}\boldsymbol{z}_s\boldsymbol{dx}_s^{\top} = \widehat{\Sigma}_{\boldsymbol{\tau}}^{-1}\left(\int_0^{\boldsymbol{\tau}}\boldsymbol{z}_s\boldsymbol{z}_s^{\top}\boldsymbol{ds}\boldsymbol{\theta_0} + \int_0^{\boldsymbol{\tau}}\boldsymbol{z}_s\boldsymbol{d}\mathbb{W}_s^{\top}\right) = \boldsymbol{\theta_0} - \widehat{\Sigma}_{\boldsymbol{\tau}}^{-1}\left(\boldsymbol{\theta_0} - \int_0^{\boldsymbol{\tau}}\boldsymbol{z}_s\boldsymbol{d}\mathbb{W}_s^{\top}\right),$$

where we used the definition of $\widehat{\Sigma}_{\boldsymbol{\tau}}$ in (8). Now, the sample $\widehat{\boldsymbol{\theta}}$ from $\mathcal{D}_{\boldsymbol{\tau}}$ can be written as $\widehat{\boldsymbol{\theta}}_{\boldsymbol{\tau}} = \widehat{\mu}_{\boldsymbol{\tau}} + \widehat{\Sigma}_{\boldsymbol{\tau}}^{-1/2}\Phi$, where $\Phi \sim \boldsymbol{N}\left(0_{(p+q)\times p}, I_{p+q}\right)$ is a standard normal random matrix, as defined in the notation. So, for the error matrix, it holds that

$$\left\|\widehat{\boldsymbol{\theta}} - \boldsymbol{\theta_0}\right\| \leq \left\|\widehat{\Sigma}_{\boldsymbol{\tau}}^{-1}\right|\left(\|\boldsymbol{\theta_0}\| + \left\|\int_0^{\boldsymbol{\tau}}\boldsymbol{z}_s\boldsymbol{d}\mathbb{W}_s^{\top}\right\|\right) + \left\|\widehat{\Sigma}_{\boldsymbol{\tau}}^{-1}\right\|^{1/2}\|\Phi\|.$$

To proceed towards bounding the above error matrix, use

$$\int_0^{\boldsymbol{\tau}}\boldsymbol{z}_s\boldsymbol{d}\mathbb{W}_s^{\top} = \int_0^{\boldsymbol{\tau}}\left(\begin{bmatrix} I_p \\ K \end{bmatrix}\boldsymbol{x}_s + \begin{bmatrix} 0 \\ v(s) \end{bmatrix}\right)\boldsymbol{d}\mathbb{W}_s^{\top},$$

to obtain

$$\left\|\int_0^{\boldsymbol{\tau}}\boldsymbol{z}_s\boldsymbol{d}\mathbb{W}_s^{\top}\right\| \leq \left(1 \vee \|K\|\right)\left\|\int_0^{\boldsymbol{\tau}}\boldsymbol{x}_s\boldsymbol{d}\mathbb{W}_s^{\top}\right\| + \left\|\int_0^{\boldsymbol{\tau}}v(s)\boldsymbol{d}\mathbb{W}_s^{\top}\right\|,$$

To proceed, note that with probability at least $1 - \delta$, we have

$$\|\Phi\|^2 \lesssim p(p + q)\log\frac{p(p + q)}{\delta}.$$

Now, by putting this together with the results of Lemma 2, Lemma 3, and (25), we get the desired result.

∎

## A.5 Eigenvalue ratio bound for sum of two matrices

**Lemma 5** *Suppose that $M, E$ are $p \times p$ matrices, and let $M = \Gamma^{-1}\Lambda\Gamma$ be the Jordan diagonalization of $M$. So, for some positive integer $k$, we have $\Lambda \in \mathbb{C}^{p \times p} = \mathrm{diag}\,(\Lambda_1, \cdots, \Lambda_k)$, where the blocks $\Lambda_1, \cdots, \Lambda_k$ are Jordan matrices of the form*

$$
\Lambda_i = \begin{bmatrix}
\lambda_i & 1 & 0 & \cdots & 0 & 0 \\
0 & \lambda_i & 1 & 0 & \cdots & 0 \\
\vdots & \vdots & \vdots & \vdots & \vdots & \vdots \\
0 & 0 & \cdots & 0 & \lambda_i & 1 \\
0 & 0 & 0 & \cdots & 0 & \lambda_i
\end{bmatrix} \in \mathbb{C}^{\boldsymbol{r}_i \times \boldsymbol{r}_i}.
$$

*Then, let $\boldsymbol{r} = \max_{1 \leq i \leq k} \boldsymbol{r}_i \leq p$, and define $\Delta_M(E)$ as the difference between the largest real-part of the eigenvalues of $M + E$ and that of $M$. Then, it holds that*

$$
\Delta_M(E) \leq \left(1 \vee \boldsymbol{r}^{1/2}\|E\|\mathrm{cond}\,(\Gamma)\right)^{1/\boldsymbol{r}}, \tag{36}
$$

*where $\mathrm{cond}\,(\Gamma)$ is the condition number of $\Gamma$: $\mathrm{cond}\,(\Gamma) = \overline{\lambda}\left(\Gamma^\top\Gamma\right)^{1/2}\underline{\lambda}\left(\Gamma^\top\Gamma\right)^{-1/2}$.*

Proof. Since the expression on the right-hand-side of (36) is positive, it is enough to consider an eigenvalue $\lambda$ of $M + E$ which is not an eigenvalue of $M$, and show that $\Re\,(\lambda) - \log\overline{\lambda}\,(\exp\,(M))$ is less than the expression on the RHS of (36). So, for such $\lambda$, the matrix $M - \lambda I_p$ is non-singular, while $M + E - \lambda I_p$ is singular. Let the vector $v \neq 0$ be such that $(M + E - \lambda I_p)\,v = 0$, which by Jordan diagonalization above implies that

$$
v = -\Gamma^{-1}\left(\Lambda - \lambda I\right)^{-1}\Gamma E v. \tag{37}
$$

Then, $\Lambda = \mathrm{diag}\,(\Lambda_1, \cdots, \Lambda_k)$ indicates that $\Lambda - \lambda I$ and $(\Lambda - \lambda I)$ are block diagonal, the latter consisting of the blocks $\mathrm{diag}\left((\Lambda_1 - \lambda I_{\boldsymbol{r}_1})^{-1}, \cdots, (\Lambda_k - \lambda I_{\boldsymbol{r}_k})^{-1}\right)$.

Now, multiplications show that

$$
(\Lambda_i - \lambda I_{\boldsymbol{r}_i})^{-1} = -\begin{bmatrix}
(\lambda - \lambda_i)^{-1} & (\lambda - \lambda_i)^{-2} & \cdots & (\lambda - \lambda_i)^{-\boldsymbol{r}_i} \\
0 & (\lambda - \lambda_i)^{-1} & \cdots & (\lambda - \lambda_i)^{-\boldsymbol{r}_i+1} \\
\vdots & \vdots & \vdots & \vdots \\
0 & \cdots & 0 & (\lambda - \lambda_i)^{-1}
\end{bmatrix}.
$$

Therefore, according to the definition of matrix operator norms in Section 1, we obtain

$$
\left\|(\Lambda_i - \lambda I_{\boldsymbol{r}_i})^{-1}\right\|^2 \leq \boldsymbol{r}\left(1 \vee |\lambda - \lambda_i|^{-\boldsymbol{r}}\right)^2.
$$

Putting these bounds for the blocks of $(\Lambda - \lambda I)^{-1}$ together, (37) leads to

$$
\begin{aligned}
1 &\leq \left\|(\Lambda - \lambda I)^{-1}\right\|\|\Gamma\|\|\Gamma^{-1}\|\|E\| \\
&\leq \boldsymbol{r}^{1/2}\mathrm{cond}\,(\Gamma)\,\|E\| \max_{1 \leq i \leq k}\left(1 \wedge |\lambda - \lambda_i|^{\boldsymbol{r}}\right)^{-1} \\
&\leq \boldsymbol{r}^{1/2}\mathrm{cond}\,(\Gamma)\,\|E\|\left(1 \wedge \left(\Re(\lambda) - \log\overline{\lambda}\,(\exp\,(M))\right)^{\boldsymbol{r}}\right)^{-1}.
\end{aligned}
$$

To see the last inequality above, note that if $\Re(\lambda) - \log\overline{\lambda}\,(\exp\,(M))$ is positive, then it is larger than all the terms $|\lambda - \lambda_i|$, for $i = 1, \cdots, k$. Thus, for

$$
\Re\,(\lambda) = \log\overline{\lambda}\,(\exp\,(M + E)),
$$

we obtain (36). ∎

# B Proof of Theorem 2

To establish the rates of exploration Algorithm 2 performs, we utilize Lemma 8, which indicates that

$$\|\widehat{\mu}_{\boldsymbol{\tau}_n} - \boldsymbol{\theta_0}\| \lesssim \left\|\widehat{\Sigma}_{\boldsymbol{\tau}_n}^{-1/2}\right\| \left\|\widehat{\Sigma}_{\boldsymbol{\tau}_n}^{-1/2} \int_0^{\boldsymbol{\tau}_n} \boldsymbol{x}_t d\mathbb{W}_t^\top\right\| \lesssim \underline{\lambda}\left(\widehat{\Sigma}_{\boldsymbol{\tau}_n}\right)^{-1/2} \left(p(p+q)\overline{\lambda}\left(\Sigma_{\mathbb{W}}\right) \log \overline{\lambda}\left(\widehat{\Sigma}_{\boldsymbol{\tau}_n}\right)\right)^{1/2}.$$

Now, (51) gives $\log \overline{\lambda}\left(\widehat{\Sigma}_{\boldsymbol{\tau}_n}\right) \lesssim \log \boldsymbol{\tau}_n$, while Lemma 9 provides $\underline{\lambda}\left(\widehat{\Sigma}_{\boldsymbol{\tau}_n}\right) \gtrsim \boldsymbol{\tau}_n^{1/2}\underline{\lambda}\left(\Sigma_{\mathbb{W}}\right)$. Moreover, since $\widehat{\Sigma}_{\boldsymbol{\tau}_n}^{1/2}\left(\boldsymbol{\theta_n} - \widehat{\mu}_{\boldsymbol{\tau}_n}\right)$ is a standard normal $(p+q) \times p$ matrix, we have

$$\left\|\widehat{\boldsymbol{\theta}_n} - \widehat{\mu}_{\boldsymbol{\tau}_n}\right\| \lesssim \boldsymbol{\tau}_n^{-1/4}\underline{\lambda}\left(\Sigma_{\mathbb{W}}\right)^{-1/2} \left(p(p+q)\log(pq)\right)^{1/2}.$$

Thus, we obtain the desired result for the estimation error.

To proceed toward establishing the regret bound, Lemma 7 shows that we need to integrate $\left\|\boldsymbol{u}_t + Q_u^{-1}\widehat{B}_n^\top P\left(\widehat{\boldsymbol{\theta}_n}\right)\boldsymbol{x}_t\right\|^2$ over the stabilized period of Algorithm 2: $\boldsymbol{\tau}_0 \leq t \leq T$:

$$\mathrm{Reg}\left(T\right) \lesssim \left(\overline{\lambda}\left(\Sigma_{\mathbb{W}}\right) + \sigma_w^2\right)\boldsymbol{\tau}_0 + \int_{\boldsymbol{\tau}_0}^T \left\|\boldsymbol{u}_t + Q_u^{-1}B_0^\top P\left(\boldsymbol{\theta_0}\right)\boldsymbol{x}_t\right\|^2 dt.$$

Further, according to (51), for $\boldsymbol{\tau}_{n-1} < T \leq \boldsymbol{\tau}_n$, we have

$$\int_{\boldsymbol{\tau}_0}^T \left\|\boldsymbol{u}_t + Q_u^{-1}B_0^\top P\left(\boldsymbol{\theta_0}\right)\boldsymbol{x}_t\right\|^2 dt \lesssim \overline{\lambda}\left(\Sigma_{\mathbb{W}}\right)\sum_{i=0}^{n-1}\left(\boldsymbol{\tau}_{i+1} - \boldsymbol{\tau}_i\right)\left\|K\left(\widehat{\boldsymbol{\theta}_i}\right) - K\left(\boldsymbol{\theta_0}\right)\right\|^2.$$

On the other hand, Lemma 12 implies that

$$\left\|K\left(\widehat{\boldsymbol{\theta}_i}\right) - K\left(\boldsymbol{\theta_0}\right)\right\| \lesssim \frac{\|P\left(\boldsymbol{\theta_0}\right)\|^3}{\underline{\lambda}\left(Q_x\right)\underline{\lambda}\left(Q_u\right)^2}\left\|\widehat{\boldsymbol{\theta}_i} - \boldsymbol{\theta_0}\right\|.$$

Thus, we have

$$\mathrm{Reg}\left(T\right) \lesssim \left(\overline{\lambda}\left(\Sigma_{\mathbb{W}}\right) + \sigma_w^2\right)\boldsymbol{\tau}_0 + \frac{\overline{\lambda}\left(\Sigma_{\mathbb{W}}\right)^2}{\underline{\lambda}\left(\Sigma_{\mathbb{W}}\right)}\frac{\|P\left(\boldsymbol{\theta_0}\right)\|^6}{\underline{\lambda}\left(Q_x\right)^2\underline{\lambda}\left(Q_u\right)^4}p(p+q)\sum_{i=0}^{n-1}\left(\boldsymbol{\tau}_{i+1} - \boldsymbol{\tau}_i\right)\frac{\log\boldsymbol{\tau}_i}{\boldsymbol{\tau}_i^{1/2}}.$$

Thus, according to (12), we obtain the desired regret bound result in Theorem 2.

## B.1 Geometry of drift parameters and optimal policies

**Lemma 6** *For the drift parameter $\boldsymbol{\theta_1}$, and for $X \in \mathbb{R}^{p \times p}, Y \in \mathbb{R}^{p \times q}$, define*

$$\Delta_{\boldsymbol{\theta_1}}(X,Y) = P\left(\boldsymbol{\theta_1}\right)Y + \int_0^\infty e^{\overline{A}_1^\top t}\left[M\left(X,Y\right)^\top P\left(\boldsymbol{\theta_1}\right) + P\left(\boldsymbol{\theta_1}\right)M\left(X,Y\right)\right]e^{\overline{A}_1 t}B_1 dt,$$

*where $\overline{A}_1 = A_1 - B_1 Q_u^{-1}B_1^\top P\left(\boldsymbol{\theta_1}\right)$ and $M\left(X,Y\right) = X - YQ_u^{-1}B_1^\top P\left(\boldsymbol{\theta_1}\right)$. Then, $\Delta_{\boldsymbol{\theta_1}}(X,Y)$ is the directional derivative of $B^\top P\left(\boldsymbol{\theta}\right)$ at $\boldsymbol{\theta_1}$ in the direction $[X,Y]$. Importantly, the tangent space of the manifold of matrices $\boldsymbol{\theta} \in \mathbb{R}^{p \times (p+q)}$ that satisfy $B^\top P\left(\boldsymbol{\theta}\right) = B_1^\top P\left(\boldsymbol{\theta_1}\right)$ at $\boldsymbol{\theta_1}$ contains all matrices $X, Y$ that $\Delta_{\boldsymbol{\theta_1}}(X,Y) = 0$.*

Proof. First, note that according to the Lipschitz continuity of $P\left(\boldsymbol{\theta}\right)$ in Lemma 12, the directional derivative exists and is well-defined, as long as $\|P\left(\boldsymbol{\theta_1}\right)\| < \infty$. However, Lemma 11 provides that $P\left(\boldsymbol{\theta_1}\right)$ is finite in a neighborhood of $\boldsymbol{\theta_0}$, and so the required condition holds. Below, we start by establishing the second result to identify the tangent space, and then prove the general result on the directional derivative.

To proceed, let $\boldsymbol{\theta} = \boldsymbol{\theta_1} + \epsilon\,[X, Y]^\top$ be such that $B^\top P(\boldsymbol{\theta}) = B_1^\top P(\boldsymbol{\theta_1})$, and denote $K(\boldsymbol{\theta_1}) = -Q_u^{-1}B_1^\top P(\boldsymbol{\theta_1})$. So, the directional derivative of $P(\boldsymbol{\theta_1})$ along the matrix $[X, Y]^\top$ can be found as follows. First, denoting the closed-loop transition matrix by $\overline{A} = A - BQ_u^{-1}B^\top P(\boldsymbol{\theta})$, since

$$\overline{A}^\top P(\boldsymbol{\theta}) + P(\boldsymbol{\theta})\,\overline{A} + Q_x + K(\boldsymbol{\theta})^\top Q_u K(\boldsymbol{\theta}) = 0,$$

we have

$$\left(\overline{A}_1 + \epsilon X + \epsilon Y K(\boldsymbol{\theta_1})\right)^\top P(\boldsymbol{\theta}) + P(\boldsymbol{\theta})\left(\overline{A}_1 + \epsilon X + \epsilon Y K(\boldsymbol{\theta_1})\right)$$
$$= -Q_x - K(\boldsymbol{\theta_1})^\top Q_u K(\boldsymbol{\theta_1}) = \overline{A}_1^\top P(\boldsymbol{\theta_1}) + P(\boldsymbol{\theta_1})\,\overline{A}_1.$$

For the matrix $E = \lim_{\epsilon \to 0} \epsilon^{-1}(P(\boldsymbol{\theta}) - P(\boldsymbol{\theta_1}))$, the latter result implies that

$$\overline{A}_1^\top E + E\overline{A}_1 + (X + Y K(\boldsymbol{\theta_1}))^\top P(\boldsymbol{\theta_1}) + P(\boldsymbol{\theta_1})(X + Y K(\boldsymbol{\theta_1})) = 0.$$

Then, since all eigenvalues of $\overline{A}_1$ are in the open left half-plane, the above Lyapunov equation for $E$ leads to the integral form

$$E = \int_0^\infty e^{\overline{A}_1^\top t}\left((X + Y K(\boldsymbol{\theta_1}))^\top P(\boldsymbol{\theta_1}) + P(\boldsymbol{\theta_1})(X + Y K(\boldsymbol{\theta_1}))\right) e^{\overline{A}_1 t} dt.$$

On the other hand, $K(\boldsymbol{\theta}) = -Q_u^{-1}B^\top P(\boldsymbol{\theta})$ gives

$$0 = \lim_{\epsilon \to 0} \frac{1}{\epsilon}\left(B^\top P(\boldsymbol{\theta}) - B_1^\top P(\boldsymbol{\theta_1})\right) = \lim_{\epsilon \to 0} \frac{1}{\epsilon}\left[(B^\top - B_1^\top) P(\boldsymbol{\theta}) - B_1^\top(P(\boldsymbol{\theta_1}) - P(\boldsymbol{\theta}))\right],$$

which, according to the definitions of $E, M(X, Y)$, implies the desired result about the tangent space of the manifold under consideration.

Next, to establish the more general result on the directional derivative, we use the directional derivative of $P(\boldsymbol{\theta})$ in (65):

$$\int_0^\infty e^{\overline{A}_1^\top t}\left(P(\boldsymbol{\theta_1})[X + Y K(\boldsymbol{\theta_1})] + [X + Y K(\boldsymbol{\theta_1})]^\top P(\boldsymbol{\theta_1})\right) e^{\overline{A}_1 t} dt.$$

Finally, since the directional derivative for $B^\top$ is $Y$, for $B^\top P(\boldsymbol{\theta})$, by the product rule it is $\Delta_{\boldsymbol{\theta_1}}(X, Y)$, which finishes the proof. ∎

## B.2 Regret bounds in terms of deviations in control actions

**Lemma 7** *Let $\boldsymbol{u}_t$ be the action that Algorithm 2 takes at time t. Then, for the regret of Algorithm 2, it holds that*

$$\mathrm{Reg}(T) \lesssim \left(\overline{\lambda}(\Sigma_{\mathbb{W}}) + \sigma_w^2\right)\boldsymbol{\tau}_0\left\|K + Q_u^{-1}B_0^\top P(\boldsymbol{\theta_0})\right\|^2$$
$$+ \int_{\boldsymbol{\tau}_0}^T \left\|\boldsymbol{u}_t + Q_u^{-1}B_0^\top P(\boldsymbol{\theta_0})\,\boldsymbol{x}_t\right\|^2 dt + x_T^{*\top} P(\boldsymbol{\theta_0})\,x_T^*,$$

*where $x_T^*$ is the terminal state under the optimal trajectory $\boldsymbol{\pi}_{\mathbf{opt}}$ in (6).*

Proof.     First, denote the optimal linear feedback of $\boldsymbol{\pi}_{\mathbf{opt}}$ in (6) by $\boldsymbol{u}_t = K(\boldsymbol{\theta_0})\,\boldsymbol{x}_t$, where $K(\boldsymbol{\theta_0}) = -Q_u^{-1}B_0 P(\boldsymbol{\theta_0})$. According to the episodic structure of Algorithm 2, for $\boldsymbol{\tau}_n \leq t < \boldsymbol{\tau}_{n+1}$, denote

$$K_t = -Q_u^{-1}\widehat{B}_n^\top P\left(\widehat{\boldsymbol{\theta}_n}\right).$$

We first consider the regret of Algorithm 2 after finishing stabilization by running Algorithm 1; i.e., for $\boldsymbol{\tau}_0 \leq t \leq T$. Fix some small $\epsilon > 0$, that we will let decay later. We proceed by finding an approximation of the regret through sampling at times $\boldsymbol{\tau}_0 + k\epsilon$, for non-negative integers $k$. To do that, denote $N = \lceil(T - \boldsymbol{\tau}_0)/\epsilon\rceil$, and define the sequence of policies $\{\widehat{\boldsymbol{\pi}}_i\}_{i=0}^N$ according to

$$\widehat{\boldsymbol{\pi}}_i = \begin{cases} \boldsymbol{u}_t = K_t \boldsymbol{x}_t & t < \boldsymbol{\tau}_0 + i\epsilon \\ \boldsymbol{u}_t = K(\boldsymbol{\theta_0})\,\boldsymbol{x}_t & t \geq \boldsymbol{\tau}_0 + i\epsilon \end{cases}.$$

That is, the policy $\widehat{\boldsymbol{\pi}}_i$ switches to the optimal feedback at time $\tau_0 + i\epsilon$. So, the zeroth policy $\widehat{\boldsymbol{\pi}}_0$ corresponds to applying the optimal policy $\boldsymbol{\pi_{\mathrm{opt}}}$ after stabilization at time $\tau_0$, while the last one $\widehat{\boldsymbol{\pi}}_N$ is nothing but the one in Algorithm 2, that we denote by $\widehat{\boldsymbol{\pi}}$, for the sake of brevity. As such, we have $\mathrm{Reg}_{\widehat{\boldsymbol{\pi}}_0}(T) = 0$, and the telescopic summation below holds true:

$$\mathrm{Reg}_{\widehat{\boldsymbol{\pi}}}(T) = \sum_{i=0}^{N-1} \left( \mathrm{Reg}_{\widehat{\boldsymbol{\pi}}_{i+1}}(T) - \mathrm{Reg}_{\widehat{\boldsymbol{\pi}}_i}(T) \right). \tag{38}$$

Now, to consider the difference $\mathrm{Reg}_{\widehat{\boldsymbol{\pi}}_{i+1}}(T) - \mathrm{Reg}_{\widehat{\boldsymbol{\pi}}_i}(T)$, for a fixed $i$ in the range $0 \le i < N$, denote $t_1 = \tau_0 + i\epsilon$ and let $\boldsymbol{x}_t^{\widehat{\boldsymbol{\pi}}_i}, \boldsymbol{x}_t^{\widehat{\boldsymbol{\pi}}_{i+1}}$ be the state trajectories under $\widehat{\boldsymbol{\pi}}_i, \widehat{\boldsymbol{\pi}}_{i+1}$, respectively. By definition, we have $\boldsymbol{x}_t^{\widehat{\boldsymbol{\pi}}_i} = \boldsymbol{x}_t^{\widehat{\boldsymbol{\pi}}_{i+1}}$, for all $t \le t_1$. So, we drop the policy superscript and use $\boldsymbol{x}_{t_1}$ to refer to the states of both of them at time $t_1$. Therefore, as long as $t_1 \le t < t_1 + \epsilon$, similar to (15), the solutions of the stochastic differential equation are

$$\boldsymbol{x}_t^{\widehat{\boldsymbol{\pi}}_i} = e^{\overline{A}_0(t-t_1)}\boldsymbol{x}_{t_1} + \int_{t_1}^{t} e^{\overline{A}_0(t-s)}d\mathbb{W}_s,$$

$$\boldsymbol{x}_t^{\widehat{\boldsymbol{\pi}}_{i+1}} = e^{\overline{A}(t-t_1)}\boldsymbol{x}_{t_1} + \int_{t_1}^{t} e^{\overline{A}(t-s)}d\mathbb{W}_s,$$

where $\overline{A}_0 = A_0 + B_0 K(\boldsymbol{\theta_0})$ and $\overline{A} = A_0 + B_0 K_{t_1}$ are the closed-loop transition matrices under $\widehat{\boldsymbol{\pi}}_i$ and $\widehat{\boldsymbol{\pi}}_{i+1}$, respectively. To work with the above two state trajectories, we define some notations for convenience:

$$
\begin{aligned}
M_0 &= Q_x + K(\boldsymbol{\theta_0})^\top Q_u K(\boldsymbol{\theta_0}), \\
M_1 &= Q_x + K_{t_1} Q_u K_{t_1}, \\
y_t &= \boldsymbol{x}_t^{\widehat{\boldsymbol{\pi}}_{i+1}} - \boldsymbol{x}_t^{\widehat{\boldsymbol{\pi}}_i}, \\
E_t &= e^{\overline{A}(t-t_1)} - e^{\overline{A}_0(t-t_1)}.
\end{aligned}
$$

Thus, letting

$$Z_t = \int_{t_1}^{t} \left[ e^{\overline{A}(t-s)} - e^{\overline{A}_0(t-s)} \right] d\mathbb{W}_s,$$

it holds that $y_t = E_t \boldsymbol{x}_{t_1} + Z_t + O(\epsilon^2)$. Further, for the observation signal $\boldsymbol{z}_t$ and the cost matrix $Q$ defined in Section 2, we have

$$
\begin{aligned}
& \int_{t_1}^{t_1+\epsilon} \left( \left\| Q^{1/2} \boldsymbol{z}_t(\widehat{\boldsymbol{\pi}}_{i+1}) \right\|^2 - \left\| Q^{1/2} \boldsymbol{z}_t(\widehat{\boldsymbol{\pi}}_i) \right\|^2 \right) dt \\
= & \int_{t_1}^{t_1+\epsilon} \left[ \left( \boldsymbol{x}_t^{\widehat{\boldsymbol{\pi}}_i} + y_t \right)^\top M_1 \left( \boldsymbol{x}_t^{\widehat{\boldsymbol{\pi}}_i} + y_t \right) - \boldsymbol{x}_t^{\widehat{\boldsymbol{\pi}}_i \top} M_0 \boldsymbol{x}_t^{\widehat{\boldsymbol{\pi}}_i} \right] dt \\
= & \int_{t_1}^{t_1+\epsilon} \left[ \boldsymbol{x}_t^{\widehat{\boldsymbol{\pi}}_i \top} S \boldsymbol{x}_t^{\widehat{\boldsymbol{\pi}}_i} + 2 y_t^\top M_1 \boldsymbol{x}_t^{\widehat{\boldsymbol{\pi}}_i} + y_t^\top M_1 y_t \right] dt, \tag{39}
\end{aligned}
$$

where $S = M_1 - M_0 = K_{t_1}^\top Q_u K_{t_1} - K(\boldsymbol{\theta_0})^\top Q_u K(\boldsymbol{\theta_0})$.

On the other hand, for $t \ge t_1 + \epsilon$, the evolutions of the state vectors are the same for the two policies and we have

$$\boldsymbol{x}_t^{\widehat{\boldsymbol{\pi}}_i} = e^{\overline{A}_0(t-t_1-\epsilon)} \boldsymbol{x}_{t_1+\epsilon}^{\widehat{\boldsymbol{\pi}}_i} + \int_{t_1+\epsilon}^{t} e^{\overline{A}_0(t-s)} d\mathbb{W}_s.$$

739  Therefore, the difference signal becomes

$$y_t = e^{\overline{A}_0(t-t_1+\epsilon)}\left[x_{t_1+\epsilon}^{\widehat{\pi}_{i+1}} - x_{t_1+\epsilon}^{\widehat{\pi}_i}\right] = e^{\overline{A}_0(t-t_1+\epsilon)}y_{t_1+\epsilon} = e^{\overline{A}_0(t-t_1+\epsilon)}\left[E_{t_1+\epsilon}x_{t_1} + Z_{t_1+\epsilon}\right],$$

740  and we obtain

$$\int_{t_1+\epsilon}^{T}\left(\left\|Q^{1/2}z_t\left(\widehat{\pi}_{i+1}\right)\right\|^2 - \left\|Q^{1/2}z_t\left(\widehat{\pi}_i\right)\right\|^2\right)dt$$

$$= \int_{t_1+\epsilon}^{T}\left[\left(x_t^{\widehat{\pi}_i} + y_t\right)^{\top}M_0\left(x_t^{\widehat{\pi}_i} + y_t\right) - x_t^{\widehat{\pi}_i\top}M_0x_t^{\widehat{\pi}_i}\right]dt$$

$$= \int_{t_1+\epsilon}^{T}\left[2y_t^{\top}M_0x_t^{\widehat{\pi}_i} + y_t^{\top}M_0y_t\right]dt. \tag{40}$$

741  Now, after doing some algebra, the expressions in (39) and (40) lead ro the following for small $\epsilon$:

$$\operatorname{Reg}_{\widehat{\pi}_{i+1}}(T) - \operatorname{Reg}_{\widehat{\pi}_i}(T) = \left(x_{t_1}^{\top}F_{t_1}x_{t_1} + 2x_{t_1}^{\top}g_{t_1}\right)\epsilon + O\left(\epsilon^2\right),$$

742  where

$$F_{t_1} = S_t + \int_{t_1}^{T}\left(2H_{t_1}^{\top}e^{\overline{A}_0^{\top}(s-t_1)}\left(Q_x + K\left(\theta_0\right)^{\top}Q_uK\left(\theta_0\right)\right)e^{\overline{A}_0(s-t_1)}\right)ds + O\left(\epsilon\right),$$

$$g_{t_1} = \int_{t_1}^{T}\left(H_{t_1}^{\top}e^{\overline{A}_0^{\top}(s-t_1)}\left(Q_x + K\left(\theta_0\right)^{\top}Q_uK\left(\theta_0\right)\right)\int_{t_1}^{s}e^{\overline{A}_0(s-u)}d\mathbb{W}_u\right)ds + O\left(\epsilon\right),$$

$$S_{t_1} = K_{t_1}^{\top}Q_uK_{t_1} - K\left(\theta_0\right)^{\top}Q_uK\left(\theta_0\right),$$

$$H_{t_1} = B_0\left(K_{t_1} - K\left(\theta_0\right)\right).$$

743  Thus, as $\epsilon$ tends to zero, by (38), we have

$$\operatorname{Reg}_{\widehat{\pi}}(T) - \operatorname{Reg}_{\widehat{\pi}}(\tau_0) = \int_{\tau_0}^{T}\left(x_t^{\top}F_tx_t + 2x_t^{\top}g_t\right)dt, \tag{41}$$

744  where $F_t, g_t$ are the above expressions, without the $O\left(\epsilon\right)$ terms.

745  Next, by (61), the quadratic expression in terms of the matrix $F_t$ can be equivalently written with

$$F_t = S_t + H_t^{\top}P\left(\theta_0\right) + P\left(\theta_0\right)H_t - H_t^{\top}E_t - E_tH_t,$$

746  where

$$E_t = \int_{T}^{\infty}e^{\overline{A}_0^{\top}(s-t)}M_0e^{\overline{A}_0(s-t)}ds = e^{\overline{A}_0^{\top}(T-t)}P\left(\theta_0\right)e^{\overline{A}_0(T-t)}.$$

747  Note that in the last equality above, we again used (61). Now, after doing some algebra similar to the
748  expression in (59), we have

$$S_t + H_t^{\top}P\left(\theta_0\right) + P\left(\theta_0\right)H_t = \left(K_t - K\left(\theta_0\right)\right)^{\top}Q_u\left(K_t - K\left(\theta_0\right)\right),$$

749  which in turn implies that

$$\int_{\tau_0}^{T}x_t^{\top}F_tx_tdt = \int_{\tau_0}^{T}\left\|Q_u^{1/2}\left(K_t - K\left(\theta_0\right)\right)x_t\right\|^2dt - 2\int_{\tau_0}^{T}x_t^{\top}E_tH_tx_tdt. \tag{42}$$

750  To study the latter integral, suppose that $x_t^*$ is the state trajectory under the optimal policy $\pi_{\mathbf{opt}}$
751  in (6), and define $\xi_t = x_t - x_t^*$.Note that (1) gives $dx_t = \left(\overline{A}_0 + H_t\right)x_tdt + d\mathbb{W}_t$, as well as

752    $dx_t^* = \overline{A}_0 x_t^* dt + d\mathbb{W}_t$. Thus, we get $d\xi_t = H_t x_t dt + \overline{A}_0 \xi_t dt$, using which, we have the following

753    for $\varphi_t = e^{-\overline{A}_0 t} \xi_t$:

$$d\varphi_t = d\left(e^{-\overline{A}_0 t}\xi_t\right) = e^{-\overline{A}_0 t}d\xi_t - \overline{A}_0 e^{-\overline{A}_0 t}\xi_t dt = e^{-\overline{A}_0 t}H_t x_t dt.$$

754    Above, we used the fact that the matrices $e^{-\overline{A}_0 t}, \overline{A}_0$ commute. So, it holds that

$$\begin{aligned}
x_t^\top E_t H_t x_t dt &= x_t^\top e^{\overline{A}_0^\top (T-t)} P(\boldsymbol{\theta_0}) e^{\overline{A}_0 T} d\varphi_t \\
&= x_t^{*\top} e^{\overline{A}_0^\top (T-t)} P(\boldsymbol{\theta_0}) e^{\overline{A}_0 T} d\varphi_t + \varphi_t^\top e^{\overline{A}_0^\top T} P(\boldsymbol{\theta_0}) e^{\overline{A}_0 T} d\varphi_t \\
&= x_t^{*\top} e^{\overline{A}_0^\top (T-t)} P(\boldsymbol{\theta_0}) e^{\overline{A}_0 T} d\varphi_t + \frac{1}{2} d\left[\varphi_t^\top e^{\overline{A}_0^\top T} P(\boldsymbol{\theta_0}) e^{\overline{A}_0 T} \varphi_t\right].
\end{aligned}$$

755    In the above expression, writing the solution of the stochastic differential equation as in (15), we have

$$e^{\overline{A}_0(T-t)}x_t^* = x_T^* - \int_t^T e^{\overline{A}_0(T-s)}d\mathbb{W}_s,$$

756    which gives

$$\begin{aligned}
2x_t^\top E_t H_t x_t dt &= 2x_t^{*\top} e^{\overline{A}_0^\top (T-t)} P(\boldsymbol{\theta_0}) e^{\overline{A}_0 T} d\varphi_t + d\left[\varphi_t^\top e^{\overline{A}_0^\top T} P(\boldsymbol{\theta_0}) e^{\overline{A}_0 T} \varphi_t\right] \\
&= -2\left(\int_t^T e^{\overline{A}_0(T-s)}d\mathbb{W}_s\right)^\top P(\boldsymbol{\theta_0}) e^{\overline{A}_0 T} d\varphi_t \\
&\quad + 2x_T^{*\top} P(\boldsymbol{\theta_0}) e^{\overline{A}_0 T} d\varphi_t + d\left[\varphi_t^\top e^{\overline{A}_0^\top T} P(\boldsymbol{\theta_0}) e^{\overline{A}_0 T} \varphi_t\right] \\
&= -2\left(\int_t^T e^{\overline{A}_0(T-s)}d\mathbb{W}_s\right)^\top P(\boldsymbol{\theta_0}) e^{\overline{A}_0 T} d\varphi_t \\
&\quad + d\left[\left(x_T^* + e^{\overline{A}_0 T}\varphi_t\right)^\top P(\boldsymbol{\theta_0})\left(x_T^* + e^{\overline{A}_0 T}\varphi_t\right)\right],
\end{aligned}$$

757    where the latest equality holds since the differential of the constant term $x_T^* P(\boldsymbol{\theta_0}) x_T^*$ is zero. Next,

758    integration by part yields to

$$\begin{aligned}
\int_{\boldsymbol{\tau}_0}^T \left(\int_t^T e^{\overline{A}_0(T-s)}d\mathbb{W}_s\right)^\top P(\boldsymbol{\theta_0}) e^{\overline{A}_0 T} d\varphi_t &= -\left(\int_{\boldsymbol{\tau}_0}^T e^{\overline{A}_0(T-s)}d\mathbb{W}_s\right)^\top P(\boldsymbol{\theta_0}) e^{\overline{A}_0 T} \varphi_{\boldsymbol{\tau}_0} \\
&\quad + \int_{\boldsymbol{\tau}_0}^T \varphi_t^\top e^{\overline{A}_0^\top T} P(\boldsymbol{\theta_0}) e^{\overline{A}_0(T-t)} d\mathbb{W}_t
\end{aligned}$$

759    Now, note the following simplifying expressions: First, by definition, we have $x_T^* + e^{\overline{A}_0 T}\varphi_T = $

760    $x_T^* + \xi_T = x_T$ and

$$x_T^* + e^{\overline{A}_0 T}\varphi_{\boldsymbol{\tau}_0} = x_T^* + e^{\overline{A}_0(T-t)}(x_{\boldsymbol{\tau}_0} - x_{\boldsymbol{\tau}_0}^*) = e^{\overline{A}_0(T-t)}x_{\boldsymbol{\tau}_0} + \int_{\boldsymbol{\tau}_0}^T e^{\overline{A}_0(T-s)}d\mathbb{W}_s,$$

761    is the terminal state vector under the policy $\widehat{\boldsymbol{\pi}}_0$ that switches to the optimal policy $\boldsymbol{\pi_{opt}}$ after the time

762    $\boldsymbol{\tau}_0$, because $\int_{\boldsymbol{\tau}_0}^T e^{\overline{A}_0(T-s)}d\mathbb{W}_s = x_T^* - e^{\overline{A}_0(T-\boldsymbol{\tau}_0)}x_{\boldsymbol{\tau}_0}^*$. Finally, according to Lemma 8, we have

$$\left\|\int_{\boldsymbol{\tau}_0}^T \varphi_t^\top e^{\overline{A}_0^\top T} P(\boldsymbol{\theta_0}) e^{\overline{A}_0(T-t)} d\mathbb{W}_t\right\| \lesssim \left(\int_{\boldsymbol{\tau}_0}^T \left\|e^{\overline{A}_0(T-t)}\xi_t\right\|^2 dt\right)^{1/2} \log\int_{\boldsymbol{\tau}_0}^T \left\|e^{\overline{A}_0(T-t)}\xi_t\right\|^2 dt.$$

Putting the above bounds together, we obtain

$$-2 \int_{\tau_0}^{T} \boldsymbol{x}_t^{\top} E_t H_t \boldsymbol{x}_t \boldsymbol{dt} - x_T^{* \top} P\left(\boldsymbol{\theta_0}\right) x_T^{*} \lesssim \int_{\tau_0}^{T} \left\| e^{\overline{A}_0 (T-t)} \left(\boldsymbol{x}_t\right) \right\|^2 \boldsymbol{dt}. \tag{43}$$

To proceed toward working with the integration of $\boldsymbol{x}_t^{\top} g_t$, employ Fubini Theorem [31] to obtain

$$
\begin{aligned}
\int_{\tau_0}^{T} \boldsymbol{x}_t^{\top} \widetilde{g}_t \boldsymbol{dt} &= \int_{\tau_0}^{T} \int_{t}^{T} \int_{t}^{s} \left( \boldsymbol{x}_t^{\top} H_t^{\top} e^{\overline{A}_0^{\top} (s-t)} M e^{\overline{A}_0 (s-u)} \right) \boldsymbol{d} \mathbb{W}_u \boldsymbol{dsdt} \\
&= \int_{\tau_0}^{T} \int_{\tau_0}^{u} \int_{u}^{T} \left( \boldsymbol{x}_t^{\top} H_t^{\top} e^{\overline{A}_0^{\top} (s-t)} M e^{\overline{A}_0 (s-u)} \right) \boldsymbol{dsdt} \boldsymbol{d} \mathbb{W}_u.
\end{aligned}
$$

Now, denote the inner double integral by $y_u^{\top}$:

$$y_u^{\top} = \int_{\tau_0}^{u} \int_{u}^{T} \left( \boldsymbol{x}_t^{\top} H_t^{\top} e^{\overline{A}_0^{\top} (s-t)} M_0 e^{\overline{A}_0 (s-u)} \right) \boldsymbol{dsdt} = \int_{0}^{u} \left( \boldsymbol{x}_t^{\top} \left(K_t - K\left(\boldsymbol{\theta_0}\right)\right)^{\top} P_{t,u}^{\top} \right) \boldsymbol{dt},$$

where

$$P_{t,u}^{\top} = B_0^{\top} \int_{u}^{T} e^{\overline{A}_0^{\top} (s-t)} M_0 e^{\overline{A}_0 (s-u)} \boldsymbol{ds}.$$

Now, let $\beta_T = \int_{\tau_0}^{T} \|y_u\|^2 \boldsymbol{du}$, and employ Lemma 8 to get

$$\int_{\tau_0}^{T} \boldsymbol{x}_t^{\top} \widetilde{g}_t \boldsymbol{dt} = \int_{\tau_0}^{T} y_u^{\top} \boldsymbol{d} \mathbb{W}_u = O\left( \beta_T^{1/2} \log^{1/2} \beta_T \right). \tag{44}$$

Thus, we can work with $\beta_T$ to bound the portion of the regret the integral of $\boldsymbol{x}_t^{\top} g_t$ captures. For that purpose, the triangle inequality and Fubini Theorem [31] lead to

$$
\begin{aligned}
\beta_T &\leq \int_{0}^{T} \int_{0}^{u} \left\| P_{t,u} \left(K_t - K\left(\boldsymbol{\theta_0}\right)\right) \boldsymbol{x}_t \right\|^2 \boldsymbol{dtdu} \\
&= \int_{0}^{T} \left( \boldsymbol{x}_t^{\top} \left(K_t - K\left(\boldsymbol{\theta_0}\right)\right)^{\top} \left[ \int_{t}^{T} P_{t,u}^{\top} P_{t,u} \boldsymbol{du} \right] \left(K_t - K\left(\boldsymbol{\theta_0}\right)\right) \boldsymbol{x}_t \right) \boldsymbol{dt} \\
&\leq \overline{\lambda} \left( \int_{t}^{T} P_{t,u}^{\top} P_{t,u} \boldsymbol{du} \right) \int_{0}^{T} \left\| \left(K_t - K\left(\boldsymbol{\theta_0}\right)\right) \boldsymbol{x}_t \right\|^2 \boldsymbol{dt}.
\end{aligned}
$$

770 We can show that $\overline{\lambda}\left(\int_t^T P_{t,u}^\top P_{t,u}\boldsymbol{d}u\right) \lesssim 1$:

$$
\begin{aligned}
\overline{\lambda}\left(\int_t^T P_{t,u}^\top P_{t,u}\boldsymbol{d}u\right) &\leq \int_t^T \left\|P_{t,u}^\top\right\|^2 \boldsymbol{d}u \\
&\lesssim \int_t^T \left\|\int_u^T e^{\overline{A}_0^\top (s-t)} M_0 e^{\overline{A}_0(s-u)}\boldsymbol{d}s\right\|^2 \boldsymbol{d}u \\
&\leq \int_t^T \left\|e^{\overline{A}_0^\top (u-t)}\right\|^2 \left\|\int_u^T e^{\overline{A}_0^\top (s-u)} M_0 e^{\overline{A}_0(s-u)}\boldsymbol{d}s\right\|^2 \boldsymbol{d}u \\
&\leq \|P(\boldsymbol{\theta_0})\|^2 \int_t^T \left\|e^{\overline{A}_0^\top (u-t)}\right\|^2 \boldsymbol{d}u \lesssim 1.
\end{aligned}
$$

771 Above, in the last inequality we use (61). Note that the last expression is a bounded constant, since
772 all eigenvalues of $\overline{A}_0$ are in the open left half-plane.

773 Thus, according to (44), it is enough to consider

$$
\beta_T \lesssim \int_0^T \|(K_t - K(\boldsymbol{\theta_0}))\,\boldsymbol{x}_t\|^2 dt, \tag{45}
$$

774 in order to bound the portion of the regret that the integration of $\boldsymbol{x}_t^\top g_t$ contributes.

775 While the above discussions apply to the regret during the time interval $\tau_0 \leq t \leq T$, we can similarly
776 bound the regret during the stabilization period $0 \leq t \leq \tau_0$. The difference is in the randomization
777 sequence $w_n, n = 0, 1, \cdots$, which is reflected through the piece-wise constant signal $v(t)$ in (14).
778 Therefore, it suffices to add the effect of $v(t)$ to the one of the Wiener process $\mathbb{W}_t$, and so $\Sigma_\mathbb{W}$ will be
779 replaced with $\left(\Sigma_\mathbb{W} + \sigma_w^2\right)$:

$$
\text{Reg}_{\widehat{\boldsymbol{\pi}}}(\boldsymbol{\tau}_0) \leq \left(\Sigma_\mathbb{W} + \sigma_w^2\right)\boldsymbol{\tau}_0\|K - K(\boldsymbol{\theta_0})\|^2. \tag{46}
$$

780 Finally, (41), (42), (43), (44), (45), and (46) together, we get the desired result. ∎

### B.3 Stochastic inequality for continuous-time self-normalized martingales

782 **Lemma 8** *Let $\boldsymbol{z}_t = \left[\boldsymbol{x}_t^\top, \boldsymbol{u}_t^\top\right]^\top$ be the observation signal and $\widehat{\Sigma}_t$ be as in (8). Then, for the*
783 *stochastic integral $\Phi_t = \int_0^t \boldsymbol{z}_s\boldsymbol{d}\mathbb{W}_s^\top$, we have*

$$
\overline{\lambda}\left(\Phi_t^\top \widehat{\Sigma}_t^{-1} \Phi_t\right) \lesssim p\overline{\lambda}(\Sigma_\mathbb{W})\left[\log \det \widehat{\Sigma}_t - \log \det \widehat{\Sigma}_0\right]. \tag{47}
$$

784 Proof. We approximate the integrals over the interval $[0, t]$ through $n$ equally distanced points in the
785 interval, and then let $n \to \infty$. So, let $\epsilon = \lfloor t/n \rfloor$, and for $k = 0, 1, \cdots, n-1$, consider the matrix

$$
M_k = \frac{1}{\epsilon}\widehat{\Sigma}_0 + \sum_{i=0}^k \boldsymbol{z}_{i\epsilon}\boldsymbol{z}_{i\epsilon}^\top.
$$

786 Using the above matrices, for $k = 1, \cdots, n$, define $\alpha_k = \boldsymbol{z}_{k\epsilon}^\top M_{k-1}^{-1}\boldsymbol{z}_{k\epsilon}$. Thus, we have

$$
\det M_k = \det\left[M_{k-1}\left(I + M_{k-1}^{-1}\boldsymbol{z}_{k\epsilon}\boldsymbol{z}_{k\epsilon}^\top\right)\right] = \det(M_{k-1})\det\left(I + M_{k-1}^{-1}\boldsymbol{z}_{k\epsilon}\boldsymbol{z}_{k\epsilon}^\top\right).
$$

787 Now, $M_{k-1}^{-1}\boldsymbol{z}_{k\epsilon}\boldsymbol{z}_{k\epsilon}^\top$ is a rank-one matrix, and so $p + q - 1$ eigenvalues of $I + M_{k-1}^{-1}\boldsymbol{z}_{k\epsilon}\boldsymbol{z}_{k\epsilon}^\top$ except
788 one are 1, and one eigenvalue is $1 + \alpha_k$. So, it holds that

$$
\frac{\det M_k}{\det M_{k-1}} = 1 + \alpha_k.
$$

Next, it is straightforward to show that

$$M_k^{-1} = \left(M_{k-1} + \boldsymbol{z}_{k\epsilon}\boldsymbol{z}_{k\epsilon}^\top\right)^{-1} = M_{k-1}^{-1} - \frac{M_{k-1}^{-1}\boldsymbol{z}_{k\epsilon}\boldsymbol{z}_{k\epsilon}^\top M_{k-1}^{-1}}{1 + \boldsymbol{z}_{k\epsilon}^\top M_{k-1}^{-1}\boldsymbol{z}_{k\epsilon}}.$$

Therefore, we have

$$\boldsymbol{z}_{k\epsilon}^\top M_k^{-1}\boldsymbol{z}_{k\epsilon} = \boldsymbol{z}_{k\epsilon}^\top \left(M_{k-1} + \boldsymbol{z}_{k\epsilon}\boldsymbol{z}_{k\epsilon}^\top\right)^{-1} \boldsymbol{z}_{k\epsilon} = \alpha_k - \frac{\alpha_k^2}{1 + \alpha_k} = \frac{\det M_k - \det M_{k-1}}{\det M_k}.$$

However, since for all $\alpha \in \mathbb{R}$ we have $1 + \alpha \le e^\alpha$, we obtain

$$\boldsymbol{z}_{k\epsilon}^\top M_k^{-1}\boldsymbol{z}_{k\epsilon} \le \log\det M_k - \log\det M_{k-1}. \tag{48}$$

To proceed, let $\mathcal{F}_k$ be the sigma-field generated by the Wiener process up to time $k\epsilon$:

$$\mathcal{F}_k = \mathcal{F}\left(\mathbb{W}_s, 0 \le s \le k\epsilon\right).$$

Further, define $L_k = \sum_{i=0}^{k} \boldsymbol{z}_{i\epsilon}\left(\mathbb{W}_{(i+1)\epsilon} - \mathbb{W}_{i\epsilon}\right)^\top$.

So, we have

$$\mathbb{E}\left[L_k^\top M_k^{-1} L_k\right] = \mathbb{E}\left[\mathbb{E}\left[L_k^\top M_k^{-1} L_k \Big| \mathcal{F}_k\right]\right] = \mathbb{E}\left[\mathbb{E}\left[\Psi_k^\top M_k^{-1} \Psi_k \Big| \mathcal{F}_k\right]\right],$$

where $\Psi_k = L_{k-1} + \boldsymbol{z}_{k\epsilon}\left(\mathbb{W}_{(k+1)\epsilon} - \mathbb{W}_{k\epsilon}\right)^\top$. Since $L_{k-1}$ is $\mathcal{F}_k$–measurable, we get

$$\begin{aligned}
&\mathbb{E}\left[L_k^\top M_k^{-1} L_k\right] \\
=\ & \mathbb{E}\left[L_{k-1}^\top M_k^{-1} L_{k-1} + \mathbb{E}\left[\left(\mathbb{W}_{(k+1)\epsilon} - \mathbb{W}_{k\epsilon}\right)\boldsymbol{z}_{k\epsilon}^\top M_k^{-1}\boldsymbol{z}_{k\epsilon}\left(\mathbb{W}_{(k+1)\epsilon} - \mathbb{W}_{k\epsilon}\right)^\top \Big| \mathcal{F}_k\right]\right] \\
=\ & \mathbb{E}\left[L_{k-1}^\top M_k^{-1} L_{k-1} + \left(\boldsymbol{z}_{k\epsilon}^\top M_k^{-1}\boldsymbol{z}_{k\epsilon}\right)\epsilon\Sigma_{\mathbb{W}}\right],
\end{aligned}$$

where in the last line above we used $\mathcal{F}_k$–measurability of $\boldsymbol{z}_{k\epsilon}, M_k$, as well as the independent increments property and the covariance matrix of the Wiener process. So, (48) implies that

$$\overline{\lambda}\left(\mathbb{E}\left[L_k^\top M_k^{-1} L_k\right]\right) - \overline{\lambda}\left(\mathbb{E}\left[L_{k-1}^\top M_{k-1}^{-1} L_{k-1}\right]\right) \le \epsilon\overline{\lambda}\left(\Sigma_{\mathbb{W}}\right)\left(\log\det\left(\epsilon M_k\right) - \log\det\left(\epsilon M_{k-1}\right)\right).$$

Thus, summing over $k = 1, \cdots, n$, we get

$$\overline{\lambda}\left(\mathbb{E}\left[L_n^\top \left(\epsilon M_n\right)^{-1} L_n\right]\right) \le \overline{\lambda}\left(\Sigma_{\mathbb{W}}\right)\left(\log\det\left(\epsilon M_n\right) - \log\det\left(\epsilon M_0\right)\right).$$

Now, consider $\overline{\lambda}\left(L_n^\top \left(\epsilon M_n\right)^{-1} L_n\right)$. Since $L_n^\top \left(\epsilon M_n\right)^{-1} L_n$ is positive semidefinite, its largest eigenvalue can be upper-bounded by its trace, which implies that

$$\begin{aligned}
\mathbb{E}\left[\overline{\lambda}\left(L_n^\top \left(\epsilon M_n\right)^{-1} L_n\right)\right] &\le \mathbb{E}\left[\mathbf{tr}\left(L_n^\top \left(\epsilon M_n\right)^{-1} L_n\right)\right] \\
&= \mathbf{tr}\left(\mathbb{E}\left[L_n^\top \left(\epsilon M_n\right)^{-1} L_n\right]\right) \\
&\le p\overline{\lambda}\left(\mathbb{E}\left[L_n^\top \left(\epsilon M_n\right)^{-1} L_n\right]\right) \\
&\le p\overline{\lambda}\left(\Sigma_{\mathbb{W}}\right)\left(\log\det\left(\epsilon M_n\right) - \log\det\left(\epsilon M_0\right)\right),
\end{aligned}$$

where we used the fact that the linear operators of trace and expected value interchange.

Thus, Martingale Convergence Theorem [31] implies that

$$\overline{\lambda}\left(L_n^\top \left(\epsilon M_n\right)^{-1} L_n\right) \lesssim p\overline{\lambda}\left(\Sigma_{\mathbb{W}}\right)\left(\log\det\left(\epsilon M_n\right) - \log\det\left(\epsilon M_0\right)\right)$$

Finally, as $n$ tends to infinity, $\epsilon$ shrinks and we obtain the desired result. ∎

## B.4  Anti-concentration of the posterior precision matrix in Algorithm 2

**Lemma 9** *In Algorithm 2, we have the following for the matrix $\widehat{\Sigma}_{\boldsymbol{\tau}_n}$ that is defined in (8):*

$$\liminf_{n \to \infty} \boldsymbol{\tau}_n^{-1/2} \underline{\lambda}\left(\widehat{\Sigma}_{\boldsymbol{\tau}_n}\right) \gtrsim \underline{\lambda}\left(\Sigma_{\mathbb{W}}\right).$$

Proof.  First, we define some notation. Recall that during the time interval $\boldsymbol{\tau}_i \leq t < \boldsymbol{\tau}_{i+1}$ corresponding to episode $i$, Algorithm 2 uses a single parameter estimate $\widehat{\boldsymbol{\theta}_i}$. So, for $i = 0, 1, \cdots$, we use $\Phi_i, K_i, \overline{A}_i$ to denote the sample covariance matrix of the state vectors of episode $i$, and the feedback and closed-loop matrices during episode $i$:

$$\Phi_i \;\; = \;\; \int_{\boldsymbol{\tau}_i}^{\boldsymbol{\tau}_{i+1}} \boldsymbol{x}_t \boldsymbol{x}_t^\top \, dt,$$

$$K_i \;\; = \;\; -Q_u^{-1} \widehat{B}_i^\top P\left(\widehat{\boldsymbol{\theta}_i}\right),$$

$$\overline{A}_i \;\; = \;\; A_0 + B_0 K_i.$$

So, it holds that

$$\widehat{\Sigma}_{\boldsymbol{\tau}_n} = \widehat{\Sigma}_{\boldsymbol{\tau}_0} + \sum_{i=0}^{n-1} L_i \Phi_i L_i^\top, \tag{49}$$

where $L_i = \begin{bmatrix} I_p \\ K_i \end{bmatrix}$.

Now, consider the matrix $\Phi_i$. Note that according to the bounded grows rates of the episode (from both above and below) in (12), both $\boldsymbol{\tau}_{i+1} - \boldsymbol{\tau}_i$ and $\boldsymbol{\tau}_i$ tend to infinity as $i$ grows. Thus, in the sequel, we suppose that the indices $n, i, j, k$ that are used for denoting the episodes, are large enough. Similar to (31), we have

$$\Phi_i = \int_0^\infty e^{\overline{A}_i s} \left[ (\boldsymbol{\tau}_{i+1} - \boldsymbol{\tau}_i) \Sigma_{\mathbb{W}} + M_i + M_i^\top + \boldsymbol{x}_{\boldsymbol{\tau}_i} \boldsymbol{x}_{\boldsymbol{\tau}_i}^\top - \boldsymbol{x}_{\boldsymbol{\tau}_{i+1}} \boldsymbol{x}_{\boldsymbol{\tau}_{i+1}}^\top \right] e^{\overline{A}_i^\top s} \, ds,$$

where

$$M_i = \int_{\boldsymbol{\tau}_i}^{\boldsymbol{\tau}_{i+1}} \boldsymbol{x}_t d\mathbb{W}_t^\top.$$

So, using the fact that the real-parts of all eigenvalues of $\overline{A}_i$ are negative and so $\boldsymbol{x}_{\boldsymbol{\tau}_{i+1}}$ can be bounded with $\exp\left(\overline{A}_i(\boldsymbol{\tau}_{i+1} - \boldsymbol{\tau}_i)\right) \boldsymbol{x}_{\boldsymbol{\tau}_i}$ similar to (30), as well as Lemma 2, we obtain the following bounds for the largest and smallest eigenvalues of $\Phi_i$

$$\underline{\lambda}\left(\Phi_i\right) \;\; \gtrsim \;\; (\boldsymbol{\tau}_{i+1} - \boldsymbol{\tau}_i) \underline{\lambda}\left(\Sigma_{\mathbb{W}}\right) \underline{\lambda}\left(\int_0^\infty e^{\overline{A}_i s} e^{\overline{A}_i^\top s} ds\right), \tag{50}$$

$$\overline{\lambda}\left(\Phi_i\right) \;\; \lesssim \;\; (\boldsymbol{\tau}_{i+1} - \boldsymbol{\tau}_i) \overline{\lambda}\left(\Sigma_{\mathbb{W}}\right) \int_0^\infty \left\| e^{\overline{A}_i s} \right\|^2 ds. \tag{51}$$

On the other hand, for the parameter estimates at the end of episodes, similar to (35), we have

$$\widehat{\Sigma}_{\boldsymbol{\tau}_i}^{1/2} \left(\widehat{\boldsymbol{\theta}_i} - \boldsymbol{\theta_0}\right) = \widehat{\Sigma}_{\boldsymbol{\tau}_i}^{1/2} \left(\widehat{\boldsymbol{\theta}_i} - \widehat{\mu}_{\boldsymbol{\tau}_i}\right) + \widehat{\Sigma}_{\boldsymbol{\tau}_i}^{-1/2} \left(-\boldsymbol{\theta_0} + \int_0^{\boldsymbol{\tau}_i} \boldsymbol{z}_s d\mathbb{W}_s^\top\right).$$

Note that by the construction of the posterior $\mathcal{D}_{\boldsymbol{\tau}_i}$ in (9), for the first term we have $\widehat{\Sigma}_{\boldsymbol{\tau}_i}^{1/2} \left(\widehat{\boldsymbol{\theta}_i} - \widehat{\mu}_{\boldsymbol{\tau}_i}\right) \sim \boldsymbol{N}\left(0, I_{p+q}\right)$. Further, for the second term, Lemma 8 together with (51) lead to

$$\left\| \widehat{\Sigma}_{\boldsymbol{\tau}_i}^{-1/2} \left(-\boldsymbol{\theta_0} + \int_0^{\boldsymbol{\tau}_i} \boldsymbol{z}_s d\mathbb{W}_s^\top\right) \right\| \lesssim (p+q) \log^{1/2} \boldsymbol{\tau}_i.$$

Therefore, we have

$$\left\|\widehat{\Sigma}_{\boldsymbol{\tau}_i}^{1/2}\left(\widehat{\boldsymbol{\theta}}_{\boldsymbol{i}} - \boldsymbol{\theta_0}\right)\right\| \lesssim (p+q)\log^{1/2}\boldsymbol{\tau}_i.$$

However, using the relationship between $\widehat{\Sigma}_{\boldsymbol{\tau}_i}$ and $\Phi_0, \cdots, \Phi_{i-1}$ in (49), we can write

$$(p+q)^2\log\boldsymbol{\tau}_i \gtrsim \left(\widehat{\boldsymbol{\theta}}_{\boldsymbol{i}} - \boldsymbol{\theta_0}\right)^{\top} \widehat{\Sigma}_{\boldsymbol{\tau}_i}\left(\widehat{\boldsymbol{\theta}}_{\boldsymbol{i}} - \boldsymbol{\theta_0}\right) \geq \left(\widehat{\boldsymbol{\theta}}_{\boldsymbol{i}} - \boldsymbol{\theta_0}\right)^{\top}\left[\sum_{j=0}^{i-1} L_j\Phi_j L_j^{\top}\right]\left(\widehat{\boldsymbol{\theta}}_{\boldsymbol{i}} - \boldsymbol{\theta_0}\right),$$

which according to the bound in (50) implies that

$$\underline{\lambda}\left(\Sigma_{\mathbb{W}}\right)\sum_{j=0}^{i-1}(\boldsymbol{\tau}_{j+1} - \boldsymbol{\tau}_j)\left\|L_j^{\top}\left(\widehat{\boldsymbol{\theta}}_{\boldsymbol{i}} - \boldsymbol{\theta_0}\right)\right\|^2 \lesssim (p+q)^2\log\boldsymbol{\tau}_i.$$

Clearly, the above result indicates that for $j < i$, it holds that

$$\left\|\left(\widehat{\boldsymbol{\theta}}_{\boldsymbol{i}} - \boldsymbol{\theta_0}\right)^{\top} L_j\right\|^2 \lesssim \frac{(p+q)^2\log\boldsymbol{\tau}_i}{\underline{\lambda}\left(\Sigma_{\mathbb{W}}\right)\left(\boldsymbol{\tau}_{j+1} - \boldsymbol{\tau}_j\right)}. \tag{52}$$

Next, we employ Lemma 6 to study hoe Algorithm 2 utilizes Thompson sampling to diversify the matrices $L_1, L_2, \cdots$. To do so, we consider the randomization the posterior $\mathcal{D}_{\boldsymbol{\tau}_i}$ applies to the sub-matrix of the parameter estimate corresponding to the input matrix $\widehat{B}_i$. That is, we aim to find the distribution of the random $p \times q$ matrix $\left(\widehat{\boldsymbol{\theta}}_{\boldsymbol{i}} - \widehat{\mu}_{\boldsymbol{\tau}_i}\right)^{\top}\begin{bmatrix}0_{p \times q}\\ I_q\end{bmatrix}$. Since $\widehat{\boldsymbol{\theta}}_{\boldsymbol{i}} - \widehat{\mu}_{\boldsymbol{\tau}_i} \sim \boldsymbol{N}\left(0, \widehat{\Sigma}_{\boldsymbol{\tau}_i}^{-1}\right)$, we have

$$E_i = \begin{bmatrix}0_{p \times q}\\ I_q\end{bmatrix}^{\top}\left(\widehat{\boldsymbol{\theta}}_{\boldsymbol{i}} - \widehat{\mu}_{\boldsymbol{\tau}_i}\right) \sim \boldsymbol{N}\left(0, \begin{bmatrix}0_{p \times q}\\ I_q\end{bmatrix}^{\top}\widehat{\Sigma}_{\boldsymbol{\tau}_i}^{-1}\begin{bmatrix}0_{p \times q}\\ I_q\end{bmatrix}\right) = \boldsymbol{N}\left(0, \left[\widehat{\Sigma}_{\boldsymbol{\tau}_i}^{-1}\right]_{22}\right), \tag{53}$$

where $\left[\widehat{\Sigma}_{\boldsymbol{\tau}_i}^{-1}\right]_{22}$ is the $q \times q$ lower-left block in $\widehat{\Sigma}_{\boldsymbol{\tau}_i}^{-1}$:

$$\widehat{\Sigma}_{\boldsymbol{\tau}_i}^{-1} = \begin{bmatrix}\left[\widehat{\Sigma}_{\boldsymbol{\tau}_i}^{-1}\right]_{11} & \left[\widehat{\Sigma}_{\boldsymbol{\tau}_i}^{-1}\right]_{12}\\ \left[\widehat{\Sigma}_{\boldsymbol{\tau}_i}^{-1}\right]_{21} & \left[\widehat{\Sigma}_{\boldsymbol{\tau}_i}^{-1}\right]_{22}\end{bmatrix}.$$

Note that $\widehat{\Sigma}_{\boldsymbol{\tau}_0}$ is a positive semi-definite matrix. Therefore, it suffices to show the desired result for $\widehat{\Sigma}_{\boldsymbol{\tau}_n} - \widehat{\Sigma}_{\boldsymbol{\tau}_0}$, and so in the sequel we remove the effect of $\widehat{\Sigma}_{\boldsymbol{\tau}_0}$ by treating $\boldsymbol{\tau}_0$ as $0$. So, to calculate the inverse $\widehat{\Sigma}_{\boldsymbol{\tau}_i}^{-1}$, we apply block matrix inversion to

$$\widehat{\Sigma}_{\boldsymbol{\tau}_i} = \begin{bmatrix}\left[\widehat{\Sigma}_{\boldsymbol{\tau}_i}\right]_{11} & \left[\widehat{\Sigma}_{\boldsymbol{\tau}_i}\right]_{12}\\ \left[\widehat{\Sigma}_{\boldsymbol{\tau}_i}\right]_{21} & \left[\widehat{\Sigma}_{\boldsymbol{\tau}_i}\right]_{22}\end{bmatrix} = \begin{bmatrix}\sum_{j=0}^{i-1}\Phi_j & \sum_{j=0}^{i-1}\Phi_j K_j^{\top}\\ \sum_{j=0}^{i-1} K_j\Phi_j & \sum_{j=0}^{i-1} K_j\Phi_j K_j^{\top}\end{bmatrix},$$

to obtain

$$\begin{aligned}\left[\widehat{\Sigma}_{\boldsymbol{\tau}_i}^{-1}\right]_{11} &= \left[\widehat{\Sigma}_{\boldsymbol{\tau}_i}\right]_{11}^{-1} + \left[\widehat{\Sigma}_{\boldsymbol{\tau}_i}\right]_{11}^{-1}\left[\widehat{\Sigma}_{\boldsymbol{\tau}_i}\right]_{12}\Omega_i^{-1}\left[\widehat{\Sigma}_{\boldsymbol{\tau}_i}\right]_{21}\left[\widehat{\Sigma}_{\boldsymbol{\tau}_i}\right]_{11}^{-1},\\ \left[\widehat{\Sigma}_{\boldsymbol{\tau}_i}^{-1}\right]_{12} &= -\left[\widehat{\Sigma}_{\boldsymbol{\tau}_i}\right]_{11}^{-1}\left[\widehat{\Sigma}_{\boldsymbol{\tau}_i}\right]_{12}\Omega_i^{-1},\\ \left[\widehat{\Sigma}_{\boldsymbol{\tau}_i}^{-1}\right]_{22} &= \Omega_i^{-1},\\ \Omega_i &= \left[\widehat{\Sigma}_{\boldsymbol{\tau}_i}\right]_{22} - \left[\widehat{\Sigma}_{\boldsymbol{\tau}_i}\right]_{21}\left[\widehat{\Sigma}_{\boldsymbol{\tau}_i}\right]_{11}^{-1}\left[\widehat{\Sigma}_{\boldsymbol{\tau}_i}\right]_{12}.\end{aligned}$$

The smallest eigenvalue of $\widehat{\Sigma}_{\boldsymbol{\tau}_i}$ is related to that of $\Omega_i$. On one hand, since $\Omega_i^{-1}$ is a sub-matrix of $\widehat{\Sigma}_{\boldsymbol{\tau}_i}^{-1}$; i.e., $\overline{\lambda}\left(\Omega_i^{-1}\right) \leq \overline{\lambda}\left(\widehat{\Sigma}_{\boldsymbol{\tau}_i}^{-1}\right)$, which implies that $\underline{\lambda}\left(\Omega_i\right) \geq \underline{\lambda}\left(\widehat{\Sigma}_{\boldsymbol{\tau}_i}\right)$. Now, we show that the

inequality holds in the opposite direction as well, modulo a constant factor. Suppose that $\nu \in \mathbb{R}^{p+q}$ is a unit vector, $\nu = [\nu_1^\top, \nu_2^\top]^\top$, $\nu_1 \in \mathbb{R}^p$, and $\nu_2 \in \mathbb{R}^q$. So, after doing some algebra as follows, we have

$$
\begin{aligned}
\nu^\top \widehat{\Sigma}_{\boldsymbol{\tau}_i} \nu &= \nu_1^\top \left[\widehat{\Sigma}_{\boldsymbol{\tau}_i}\right]_{11} \nu_1 + 2\nu_1^\top \left[\widehat{\Sigma}_{\boldsymbol{\tau}_i}\right]_{12} \nu_2 + \nu_2^\top \left[\widehat{\Sigma}_{\boldsymbol{\tau}_i}\right]_{22} \nu_2 \\
&= \nu_1^\top \left[\widehat{\Sigma}_{\boldsymbol{\tau}_i}\right]_{11} \nu_1 + 2\nu_1^\top \left[\widehat{\Sigma}_{\boldsymbol{\tau}_i}\right]_{11} \left[\widehat{\Sigma}_{\boldsymbol{\tau}_i}\right]_{11}^{-1} \left[\widehat{\Sigma}_{\boldsymbol{\tau}_i}\right]_{12} \nu_2 \\
&\quad + \nu_2^\top \left[\widehat{\Sigma}_{\boldsymbol{\tau}_i}\right]_{21} \left[\widehat{\Sigma}_{\boldsymbol{\tau}_i}\right]_{11}^{-1} \left[\widehat{\Sigma}_{\boldsymbol{\tau}_i}\right]_{11} \left[\widehat{\Sigma}_{\boldsymbol{\tau}_i}\right]_{11}^{-1} \left[\widehat{\Sigma}_{\boldsymbol{\tau}_i}\right]_{12} \nu_2 \\
&\quad + \nu_2^\top \left[\widehat{\Sigma}_{\boldsymbol{\tau}_i}\right]_{22} \nu_2 - \nu_2^\top \left[\widehat{\Sigma}_{\boldsymbol{\tau}_i}\right]_{21} \left[\widehat{\Sigma}_{\boldsymbol{\tau}_i}\right]_{11}^{-1} \left[\widehat{\Sigma}_{\boldsymbol{\tau}_i}\right]_{12} \nu_2 \\
&= \left(\nu_1 + \left[\widehat{\Sigma}_{\boldsymbol{\tau}_i}\right]_{11}^{-1} \left[\widehat{\Sigma}_{\boldsymbol{\tau}_i}\right]_{12} \nu_2\right)^\top \left[\widehat{\Sigma}_{\boldsymbol{\tau}_i}\right]_{11} \left(\nu_1 + \left[\widehat{\Sigma}_{\boldsymbol{\tau}_i}\right]_{11}^{-1} \left[\widehat{\Sigma}_{\boldsymbol{\tau}_i}\right]_{12} \nu_2\right) \\
&\quad + \nu_2 \Omega_i \nu_2.
\end{aligned}
$$

For the matrix $\left[\widehat{\Sigma}_{\boldsymbol{\tau}_i}\right]_{11} = \sum_{j=0}^{i-1} \Phi_j$, the smallest eigenvalue lower bounds in (50) lead to $\underline{\lambda}\left(\left[\widehat{\Sigma}_{\boldsymbol{\tau}_i}\right]_{11}\right) \gtrsim \boldsymbol{\tau}_i \underline{\lambda}(\Sigma_{\mathbb{W}})$. Thus, in order to show the desired smallest eigenvalue result for $\widehat{\Sigma}_{\boldsymbol{\tau}_n}$, it suffices to consider unit vectors $\nu$ for which $\left\| \nu_1 + \left[\widehat{\Sigma}_{\boldsymbol{\tau}_i}\right]_{11}^{-1} \left[\widehat{\Sigma}_{\boldsymbol{\tau}_i}\right]_{12} \nu_2 \right\| \lesssim \tau_i^{-1/4}$ holds. For such unit vectors $\nu$, the expressions $\left[\widehat{\Sigma}_{\boldsymbol{\tau}_i}\right]_{11} = \sum_{j=0}^{i-1} \Phi_j$ and $\left[\widehat{\Sigma}_{\boldsymbol{\tau}_i}\right]_{12} = \sum_{j=0}^{i-1} \Phi_j K_j^\top$, as well as Lemma 11 that indicates that the matrices $K_j$ are bounded, $\|\nu_2\|$ needs to be bounded away from zero since $\|\nu_1\|^2 + \|\nu_2\|^2 = \|\nu\|^2 = 1$. Thus, we have

$$
\underline{\lambda}(\Omega_i) \geq \underline{\lambda}\left(\widehat{\Sigma}_{\boldsymbol{\tau}_i}\right) \gtrsim \underline{\lambda}(\Omega_i). \tag{54}
$$

Otherwise, the desired result about the eigenvalue of $\widehat{\Sigma}_{\boldsymbol{\tau}_n}$ holds true.

By simplifying the following expression, we get

$$
\begin{aligned}
&\sum_{j=0}^{i-1} \left(K_j^\top - \left[\widehat{\Sigma}_{\boldsymbol{\tau}_i}\right]_{11}^{-1} \left[\widehat{\Sigma}_{\boldsymbol{\tau}_i}\right]_{12}\right)^\top \Phi_j \left(K_j^\top - \left[\widehat{\Sigma}_{\boldsymbol{\tau}_i}\right]_{11}^{-1} \left[\widehat{\Sigma}_{\boldsymbol{\tau}_i}\right]_{12}\right) \\
&= \sum_{j=0}^{i-1} K_j \Phi_j K_j^\top - \sum_{j=0}^{i-1} K_j \Phi_j \left[\widehat{\Sigma}_{\boldsymbol{\tau}_i}\right]_{11}^{-1} \left[\widehat{\Sigma}_{\boldsymbol{\tau}_i}\right]_{12} \\
&\quad - \sum_{j=0}^{i-1} \left(\left[\widehat{\Sigma}_{\boldsymbol{\tau}_i}\right]_{11}^{-1} \left[\widehat{\Sigma}_{\boldsymbol{\tau}_i}\right]_{12}\right)^\top \Phi_j K_j^\top + \sum_{j=0}^{i-1} \left(\left[\widehat{\Sigma}_{\boldsymbol{\tau}_i}\right]_{11}^{-1} \left[\widehat{\Sigma}_{\boldsymbol{\tau}_i}\right]_{12}\right)^\top \Phi_j \left[\widehat{\Sigma}_{\boldsymbol{\tau}_i}\right]_{11}^{-1} \left[\widehat{\Sigma}_{\boldsymbol{\tau}_i}\right]_{12} \\
&= \left[\widehat{\Sigma}_{\boldsymbol{\tau}_i}\right]_{22} - \left[\widehat{\Sigma}_{\boldsymbol{\tau}_i}\right]_{21} \left[\widehat{\Sigma}_{\boldsymbol{\tau}_i}\right]_{11}^{-1} \left[\widehat{\Sigma}_{\boldsymbol{\tau}_i}\right]_{12} - \left(\left[\widehat{\Sigma}_{\boldsymbol{\tau}_i}\right]_{11}^{-1} \left[\widehat{\Sigma}_{\boldsymbol{\tau}_i}\right]_{12}\right)^\top \left[\widehat{\Sigma}_{\boldsymbol{\tau}_i}\right]_{21} \\
&\quad + \left(\left[\widehat{\Sigma}_{\boldsymbol{\tau}_i}\right]_{11}^{-1} \left[\widehat{\Sigma}_{\boldsymbol{\tau}_i}\right]_{12}\right)^\top \left[\widehat{\Sigma}_{\boldsymbol{\tau}_i}\right]_{11} \left[\widehat{\Sigma}_{\boldsymbol{\tau}_i}\right]_{11}^{-1} \left[\widehat{\Sigma}_{\boldsymbol{\tau}_i}\right]_{12} \\
&= \Omega_i.
\end{aligned}
$$

However, we have

$$
K_j^\top - \left[\widehat{\Sigma}_{\boldsymbol{\tau}_i}\right]_{11}^{-1} \left[\widehat{\Sigma}_{\boldsymbol{\tau}_i}\right]_{12} = \left[\widehat{\Sigma}_{\boldsymbol{\tau}_i}\right]_{11}^{-1} \left(\left[\widehat{\Sigma}_{\boldsymbol{\tau}_i}\right]_{11} K_j^\top - \sum_{k=0}^{i-1} \Phi_k K_k^\top\right) = \left[\widehat{\Sigma}_{\boldsymbol{\tau}_i}\right]_{11}^{-1} \sum_{k=0}^{i-1} \Phi_k (K_j - K_k)^\top,
$$

i.e.,

$$
\Omega_i = \sum_{j=0}^{i-1} \left(\left[\widehat{\Sigma}_{\boldsymbol{\tau}_i}\right]_{11}^{-1} \sum_{k=0}^{i-1} \Phi_k (K_j - K_k)^\top\right)^\top \Phi_j \left(\left[\widehat{\Sigma}_{\boldsymbol{\tau}_i}\right]_{11}^{-1} \sum_{k=0}^{i-1} \Phi_k (K_j - K_k)^\top\right). \tag{55}
$$

We use the above expression to relate the matrices $\Omega_0, \Omega_1, \cdots$ to each others. First, let $\Psi_0, \Psi_1, \cdots$ be a sequence of independent random $q \times p$ matrices with standard normal distribution

$$\Psi_i \sim \boldsymbol{N}\left(0_{q \times p}, I_q\right). \tag{56}$$

Then, since $\left[\widehat{\Sigma}_{\boldsymbol{\tau}_i}^{-1}\right]_{22} = \Omega_i^{-1}$ and (53), we can let $E_i = \Omega_i^{-1/2}\Psi_i$. Further, for $j, k = 0, 1, \cdots$, denote the $B$-part of the differences $\widehat{\mu}_k - \widehat{\mu}_j$ by

$$H_{kj} = [0_{q \times p}, I_q]\left(\widehat{\mu}_k - \widehat{\mu}_j\right).$$

Note that the above result together with (53) give

$$[0_{q \times p}, I_q]\left(\widehat{\boldsymbol{\theta}}_{\boldsymbol{k}} - \widehat{\boldsymbol{\theta}}_{\boldsymbol{j}}\right) = H_{kj} + \Omega_k^{-1/2}\Psi_k - \Omega_j^{-1/2}\Psi_j.$$

We will show in the sequel that the above normally distributed random matrices are the effective randomizations that Thompson sampling Algorithm 2 applies for exploration. For that purpose, using the directional derivatives and the optimality manifolds in Lemma 6, we calculate $K_k - K_j$ according to $H_{kj} + \Omega_k^{1/2}\Psi_k - \Omega_j^{1/2}\Psi_j$. Plugging (52) in the expression for $\Delta_{\boldsymbol{\theta}_1}(X, Y)$ in Lemma 6 for

$$[X, Y] = \widehat{\boldsymbol{\theta}}_{\boldsymbol{k}}^{\top} - \widehat{\boldsymbol{\theta}}_{\boldsymbol{j}}^{\top},$$

we have

$$\left\|\int_0^\infty e^{\overline{A}_j^\top t}\left[L_j^\top\left(\widehat{\boldsymbol{\theta}}_{\boldsymbol{k}} - \widehat{\boldsymbol{\theta}}_{\boldsymbol{j}}\right)P\left(\widehat{\boldsymbol{\theta}}_{\boldsymbol{j}}\right) + P\left(\widehat{\boldsymbol{\theta}}_{\boldsymbol{j}}\right)\left(\widehat{\boldsymbol{\theta}}_{\boldsymbol{k}} - \widehat{\boldsymbol{\theta}}_{\boldsymbol{j}}\right)^\top L_j\right]e^{\overline{A}_j t}dt\right\|^2 \lesssim \frac{(p+q)^2 \log \boldsymbol{\tau}_k}{\underline{\lambda}\left(\Sigma_{\mathbb{W}}\right)\left(\boldsymbol{\tau}_{j+1} - \boldsymbol{\tau}_j\right)},$$

and

$$P\left(\widehat{\boldsymbol{\theta}}_{\boldsymbol{j}}\right)Y = P\left(\widehat{\boldsymbol{\theta}}_{\boldsymbol{j}}\right)\left(\widehat{\boldsymbol{\theta}}_{\boldsymbol{k}} - \widehat{\boldsymbol{\theta}}_{\boldsymbol{j}}\right)^\top\begin{bmatrix}0_{p \times q}\\ I_q\end{bmatrix} = P\left(\widehat{\boldsymbol{\theta}}_{\boldsymbol{j}}\right)\left(H_{kj} + \Omega_k^{-1/2}\Psi_k - \Omega_j^{-1/2}\Psi_j\right)^\top.$$

Putting the above two portions of $\Delta_{\boldsymbol{\theta}_1}(X, Y)$ together, since $\Psi_k, \Psi_j$ are independent and standard normal random matrices, (54) implies that the latter portion of $\Delta_{\boldsymbol{\theta}_1}(X, Y)$ is the dominant one. Thus, according to Lemma 6 and the expression for the optimal feedbacks in (6), we can approximate $K_k - K_j$ in (55) by

$$-Q_u^{-1}\left(H_{kj} + \Omega_k^{-1/2}\Psi_k - \Omega_j^{-1/2}\Psi_j\right)P\left(\widehat{\boldsymbol{\theta}}_{\boldsymbol{j}}\right).$$

We use the above approximation for the matrix $\left[\widehat{\Sigma}_{\boldsymbol{\tau}_i}\right]_{11}^{-1} \sum_{k=0}^{i-1} \Phi_k\left(K_j - K_k\right)^\top$ in (55), letting the episode number $i$ grow. So, the following expression captures the limit behavior of the least eigenvalue of $\widehat{\Sigma}_{\boldsymbol{\tau}_n}$ in Algorithm 2:

$$\lim_{n \to \infty} \frac{Q_u \Omega_n Q_u}{\boldsymbol{\tau}_n^{1/2} \underline{\lambda}\left(\Sigma_{\mathbb{W}}\right)} = \lim_{n \to \infty} \sum_{j=0}^{n-1}\left(\sum_{k=0}^{n-1}\widetilde{\Phi}_k P\left(\widehat{\boldsymbol{\theta}}_{\boldsymbol{j}}\right)\left(\frac{H_{kj} + \Omega_k^{-1/2}\Psi_k - \Omega_j^{-1/2}\Psi_j}{\boldsymbol{\tau}_n^{-1/4}}\right)^\top\right)^\top$$

$$\frac{\boldsymbol{\tau}_{j+1} - \boldsymbol{\tau}_j}{\boldsymbol{\tau}_n \underline{\lambda}\left(\Sigma_{\mathbb{W}}\right)}\frac{\Phi_j}{\boldsymbol{\tau}_{j+1} - \boldsymbol{\tau}_j}\left(\sum_{k=0}^{n-1}\widetilde{\Phi}_k P\left(\widehat{\boldsymbol{\theta}}_{\boldsymbol{j}}\right)\left(\frac{H_{kj} + \Omega_k^{-1/2}\Psi_k - \Omega_j^{-1/2}\Psi_j}{\boldsymbol{\tau}_n^{-1/4}}\right)^\top\right), \tag{57}$$

where

$$\widetilde{\Phi}_k = \left[\sum_{i=0}^{n-1}\Phi_i\right]^{-1}\Phi_k.$$

The equation in (57) provides the limit behavior of the randomized exploration Algorithm 2 performs for learning to control the diffusion process. More precisely, it shows the roles of the random samples from the posteriors through the random matrices $\Omega_k^{-1/2}\Psi_k$, for $k = 0, \cdots, n-1$, which render the limit matrix in (57) a positive definite one, as describe below.

Note that since $\sum_{i=0}^{n-1} \widetilde{\Phi_i} = I_p$, the expression

$$\sum_{k=0}^{n-1} \widetilde{\Phi_k} P\left(\widehat{\boldsymbol{\theta_j}}\right) \left(\frac{H_{kj} + \Omega_k^{-1/2}\Psi_k - \Omega_j^{-1/2}\Psi_j}{\boldsymbol{\tau}_n^{-1/4}}\right)^\top$$

is a weighted average of the random matrices $\boldsymbol{\tau}_n^{1/4}\left(H_{kj} + \Omega_k^{-1/2}\Psi_k - \Omega_j^{-1/2}\Psi_j\right)^\top$. Moreover, according to the discussions leading to (50) and (51), the matrix $(\boldsymbol{\tau}_{j+1} - \boldsymbol{\tau}_j)^{-1}\Phi_j$ converge as $j$ grows to a positive definite matrix, for which all eigenvalues are larger than $\underline{\lambda}(\Sigma_{\mathbb{W}})$, modulo a constant factor. On the other hand, because the lengths of the episodes satisfies the bounded growth rates in (12), the ratios $\boldsymbol{\tau}_n^{-1}(\boldsymbol{\tau}_{j+1} - \boldsymbol{\tau}_j)$ are bounded from above and below by $\underline{\alpha}^{n-j}$ and $\overline{\alpha}(\overline{\alpha}+1)^{n-j-1}$, and their sum over $j = 0, \cdots, n-1$ is 1. A similar property of boundedness from above and below applies to $\boldsymbol{\tau}_j^{-1/4}\boldsymbol{\tau}_n^{-1/4}$. So, the expression on the right-hand-side of (57) is in fact a weighted average of

$$\sum_{k=0}^{n-1} \widetilde{\Phi_k} P\left(\widehat{\boldsymbol{\theta_j}}\right) \left(\frac{H_{kj} + \Omega_k^{-1/2}\Psi_k - \Omega_j^{-1/2}\Psi_j}{\boldsymbol{\tau}_n^{-1/4}}\right)^\top,$$

for $j = 0, \cdots, n-1$.

Note that by the distribution of the random matrices in (56), all rows of $\boldsymbol{\tau}_n^{1/4}\left(H_{kj} + \Omega_k^{-1/2}\Psi_k - \Omega_j^{-1/2}\Psi_j\right)^\top$ are independent normal random vectors, implying that these random matrices are almost surely full-rank. Therefore, $\boldsymbol{\tau}_n^{-1/2}\Omega_n$ converges to a positive definite random matrix, which according to (54) implies the desired result.

■

## C Auxiliary Lemmas

### C.1 Behaviors of diffusion processes under non-optimal feedback

**Lemma 10** *Let $\widehat{A}, \widehat{B}$ be an arbitrary pair of stabilizable system matrices. Suppose that for the closed-loop matrix $\overline{A} = \widehat{A} + \widehat{B}K$, we have $\overline{\lambda}\left(\exp\left(\overline{A}\right)\right) < 1$, and $P$ satisfies*

$$\overline{A}^\top P + P\overline{A} + Q_x + K^\top Q_u K = 0.$$

*Then, it holds that*

$$P = P\left(\widehat{\boldsymbol{\theta}}\right) + \int_0^\infty e^{\overline{A}^\top t}\left(K + Q_u^{-1}\widehat{B}^\top P\left(\widehat{\boldsymbol{\theta}}\right)\right)^\top \left(Q_u K + \widehat{B}^\top P\left(\widehat{\boldsymbol{\theta}}\right)\right)e^{\overline{A}t}dt.$$

Proof. Denote $K\left(\widehat{\boldsymbol{\theta}}\right) = -Q_u^{-1}\widehat{B}^\top P\left(\widehat{\boldsymbol{\theta}}\right)$ and $\widehat{\overline{A}} = \widehat{A} + \widehat{B}K\left(\widehat{\boldsymbol{\theta}}\right)$. So, after doing some algebra, it is easy to show that the algebraic Riccati equation in (5) gives

$$\widehat{\overline{A}}^\top P\left(\widehat{\boldsymbol{\theta}}\right) + P\left(\widehat{\boldsymbol{\theta}}\right)\widehat{\overline{A}} + Q_x + K\left(\widehat{\boldsymbol{\theta}}\right)^\top Q_u K\left(\widehat{\boldsymbol{\theta}}\right).$$

Now, let $\Phi = K^\top Q_u K - K\left(\widehat{\boldsymbol{\theta}}\right)^\top Q_u K\left(\widehat{\boldsymbol{\theta}}\right)$, and subtract the above equation that $P\left(\widehat{\boldsymbol{\theta}}\right)$ solves, from the similar one in the statement of the lemma that $P$ satisfies, to get

$$\left(\overline{A} - \widehat{\overline{A}}_1\right)^\top P\left(\widehat{\boldsymbol{\theta}}\right) + P\left(\widehat{\boldsymbol{\theta}}\right)\left(\overline{A} - \widehat{\overline{A}}_1\right) + \overline{A}^\top\left(P - P\left(\widehat{\boldsymbol{\theta}}\right)\right) + \left(P - P\left(\widehat{\boldsymbol{\theta}}\right)\right)\overline{A} + \Phi = 0. \quad (58)$$

Because $\overline{\lambda}\left(\exp\left(\overline{A}\right)\right) < 1$, by solving (58) for $P - P\left(\widehat{\boldsymbol{\theta}}\right)$, we have

$$P - P\left(\widehat{\boldsymbol{\theta}}\right) = \int_0^\infty e^{\overline{A}^\top t}\left(\Phi + \left[K - K\left(\widehat{\boldsymbol{\theta}}\right)\right]^\top \widehat{B}^\top P\left(\widehat{\boldsymbol{\theta}}\right) + P\left(\widehat{\boldsymbol{\theta}}\right)\widehat{B}\left[K - K\left(\widehat{\boldsymbol{\theta}}\right)\right]\right)e^{\overline{A}t}dt,$$

where the fact $\overline{A} - \widehat{\overline{A}} = \widehat{B}\left[K - K\left(\widehat{\boldsymbol{\theta}}\right)\right]$ is used above. Then, using $\widehat{B}^\top P\left(\widehat{\boldsymbol{\theta}}\right) = -Q_u K\left(\widehat{\boldsymbol{\theta}}\right)$, it is straightforward to see

$$\left(K - K\left(\widehat{\boldsymbol{\theta}}\right)\right)^\top Q_u\left(K - K\left(\widehat{\boldsymbol{\theta}}\right)\right)$$
$$= \Phi + \left[K - K\left(\widehat{\boldsymbol{\theta}}\right)\right]^\top \widehat{B}^\top P\left(\widehat{\boldsymbol{\theta}}\right) + P\left(\widehat{\boldsymbol{\theta}}\right)\widehat{B}\left[K - K\left(\widehat{\boldsymbol{\theta}}\right)\right], \quad (59)$$

which leads to the desired result. ∎

### C.2 Behaviors of diffusion processes in a neighborhood of the truth

**Lemma 11** *Letting $\boldsymbol{\zeta_0}$ be as defined in (10), assume that*

$$\left\|\widehat{\boldsymbol{\theta}} - \boldsymbol{\theta_0}\right\| \lesssim \frac{\lambda\left(Q_u\right)}{\|B_0\|\|P\left(\boldsymbol{\theta_0}\right)\|}\left(\frac{[\boldsymbol{\zeta_0} \wedge 1]^p}{p^{1/2}} \wedge \frac{\lambda\left(Q_x\right)}{\|P\left(\boldsymbol{\theta_0}\right)\|}\right). \quad (60)$$

*Then, for the Riccati equation in (5) which is denoted by $P\left(\boldsymbol{\theta}\right)$, we have $\left\|P\left(\widehat{\boldsymbol{\theta}}\right)\right\| \lesssim \|P\left(\boldsymbol{\theta_0}\right)\|$. Furthermore, for any eigenvalue $\lambda$ of $\widehat{A} - \widehat{B}Q_u^{-1}\widehat{B}^\top P\left(\widehat{\boldsymbol{\theta}}\right)$, it holds that $\Re\left(\lambda\right) \lesssim -\underline{\lambda}\left(Q_x\right)\|P\left(\boldsymbol{\theta_0}\right)\|^{-1}$.*

Proof. First, let us write $\overline{A}_0 = A_0 - B_0 Q_u^{-1} B_0^\top P\left(\boldsymbol{\theta_0}\right)$ and

$$\overline{A}_1 = \widehat{A} - \widehat{B}Q_u^{-1}B_0^\top P\left(\boldsymbol{\theta_0}\right) = \overline{A}_0 + E_1,$$

where $E_1 = \widehat{A} - A_0 - \left(\widehat{B} - B_0\right)Q_u^{-1}B_0^\top P\left(\boldsymbol{\theta_0}\right)$. Since $r \le p$, (60) implies that $E_1$ satisfies

$$\|E_1\| \lesssim r^{-1/2}[\boldsymbol{\zeta_0} \wedge 1]^r.$$

So, letting $M = \overline{A}_0$ in (36), Lemma 5 leads to the fact that all eigenvalues of $\exp\left(\overline{A}_1\right)$ are inside the unit-circle. Therefore, all eigenvalues of $\overline{A}_1$ are on the open left half-plane of the complex plane. Now, in Lemma 10, let $K = -Q_u^{-1} B_0^\top P\left(\boldsymbol{\theta_0}\right)$ and $\overline{A} = \overline{A}_1$, to obtain the matrix denoted by $P$ in the lemma. Since $P$ satisfies

$$
Q_x + K^\top Q_u K = -\overline{A}_1^\top P - P\overline{A}_1 = -\overline{A}_0^\top P - P\overline{A}_0 - E_1^\top P - PE_1,
$$

writing Lemma 10 for $\overline{A} = \overline{A}_0$, but replacing $Q_x$ with $Q_x + E_1^\top P + PE_1$, we have

$$
P = \int_0^\infty e^{\overline{A}_0^\top t} \left[Q_x + K^\top Q_u K + E_1^\top P + PE_1\right] e^{\overline{A}_0 t} dt.
$$

However, according to (5), we have

$$
P\left(\boldsymbol{\theta_0}\right) = \int_0^\infty e^{\overline{A}_0^\top t} \left[Q_x + K^\top Q_u K\right] e^{\overline{A}_0 t} dt. \tag{61}
$$

Thus, it holds that

$$
P = P\left(\boldsymbol{\theta_0}\right) + \int_0^\infty e^{\overline{A}_0^\top t} \left[E_1^\top P + PE_1\right] e^{\overline{A}_0 t} dt,
$$

which leads to

$$
\|P\| \leq \|P\left(\boldsymbol{\theta_0}\right)\| + 2\|E_1\|\|P\| \int_0^\infty \left\|e^{\overline{A}_0 t}\right\|^2 dt.
$$

We will shortly show that $2\|E_1\| \int_0^\infty \left\|e^{\overline{A}_0 t}\right\|^2 dt < 1$. So, by Lemma 10, we have $\left\|P\left(\widehat{\boldsymbol{\theta}}\right)\right\| \leq \|P\| \lesssim \|P\left(\boldsymbol{\theta_0}\right)\|$, which is the desired result.

To proceed, denote the closed-loop matrix by $\widehat{\overline{A}} = \widehat{A} - \widehat{B}Q_u^{-1}\widehat{B}^\top P\left(\widehat{\boldsymbol{\theta}}\right)$, and let the $p$ dimensional unit vector $\nu$ attain the maximum of $\left\|\exp(\widehat{\overline{A}})\nu\right\|$, i.e., $\left\|\exp(\widehat{\overline{A}})\nu\right\| = \left\|\exp(\widehat{\overline{A}})\right\|$. Then, (61) for $\widehat{\boldsymbol{\theta}}$ (instead of $\boldsymbol{\theta_0}$) implies that

$$
\left\|P\left(\widehat{\boldsymbol{\theta}}\right)\right\| \geq \nu^\top P\left(\widehat{\boldsymbol{\theta}}\right)\nu = \int_0^\infty \nu^\top e^{\widehat{\overline{A}}^\top t} \left[Q_x + P\left(\widehat{\boldsymbol{\theta}}\right)^\top \widehat{B}Q_u^{-1}\widehat{B}^\top P\left(\widehat{\boldsymbol{\theta}}\right)\right] e^{\widehat{\overline{A}} t} \nu dt.
$$

Therefore, $\underline{\lambda}\left(Q_x + P\left(\widehat{\boldsymbol{\theta}}\right)^\top \widehat{B}Q_u^{-1}\widehat{B}^\top P\left(\widehat{\boldsymbol{\theta}}\right)\right) \geq \underline{\lambda}\left(Q_x\right)$, together with the fact that the magnitudes of all eigenvalues are smaller than the operator norm, imply that for an arbitrary eigenvalue $\lambda$ of $\widehat{\overline{A}}$, we have

$$
\|P\left(\boldsymbol{\theta_0}\right)\| \gtrsim \left\|P\left(\widehat{\boldsymbol{\theta}}\right)\right\| \geq \underline{\lambda}\left(Q_x\right) \int_0^\infty e^{2\Re(\lambda)t} dt, \tag{62}
$$

which leads to the second desired result of the lemma. To complete the proof, we need to establish that $\|E_1\| \int_0^\infty \left\|e^{\overline{A}_0 t}\right\|^2 dt < 1/2$. For that purpose, if we write (62) for $\boldsymbol{\theta_0}$ instead of $\widehat{\boldsymbol{\theta}}$, the condition in (60) implies the above bound. $\blacksquare$

### C.3 Perturbation analysis for algebraic Riccati equation in (5)

**Lemma 12** *Assume that* (60) *holds. Then, we have*

$$
\left\|P\left(\widehat{\boldsymbol{\theta}}\right) - P\left(\boldsymbol{\theta_0}\right)\right\| \lesssim \frac{\|P\left(\boldsymbol{\theta_0}\right)\|^2}{\underline{\lambda}\left(Q_x\right)} \left(1 \vee \|Q_u^{-1}B_0^\top P\left(\boldsymbol{\theta_0}\right)\|\right) \left\|\widehat{\boldsymbol{\theta}} - \boldsymbol{\theta_0}\right\|.
$$

Proof. First, fix the dynamics matrix $\widehat{\boldsymbol{\theta}}$, and let $\mathcal{C}$ be a linear segment connecting $\boldsymbol{\theta_0}$ and $\widehat{\boldsymbol{\theta}}$:

$$\mathcal{C} = \left\{ (1-\alpha)\boldsymbol{\theta_0} + \alpha\widehat{\boldsymbol{\theta}} \right\}_{0 \leq \alpha \leq 1}.$$

Let $\boldsymbol{\theta_1} \in \mathcal{C}$ be arbitrary. Then, the derivative of $P(\boldsymbol{\theta})$ at $\boldsymbol{\theta_1}$ in the direction of $\mathcal{C}$ can be found by using the difference matrices $E_A = \widehat{A} - A_0$, $E_B = \widehat{B} - B_0$. Denote $E = [E_A, E_B]^\top$. Then, we find $P(\boldsymbol{\theta_2})$, where $\boldsymbol{\theta_2} = \boldsymbol{\theta_1} + \epsilon E$, for an infinitesimal value of $\epsilon$. So, we have

$$
\begin{aligned}
&P(\boldsymbol{\theta_1}) B_2 Q_u^{-1} B_2^\top P(\boldsymbol{\theta_1}) \\
&= \epsilon P(\boldsymbol{\theta_1}) E_B Q_u^{-1} B_2^\top P(\boldsymbol{\theta_1}) + P(\boldsymbol{\theta_1}) B_1 Q_u^{-1} B_2^\top P(\boldsymbol{\theta_1}) \\
&= O(\epsilon^2) + \epsilon P(\boldsymbol{\theta_1}) E_B Q_u^{-1} B_1^\top P(\boldsymbol{\theta_1}) + \epsilon P(\boldsymbol{\theta_1}) B_1 Q_u^{-1} E_B^\top P(\boldsymbol{\theta_1}) + P(\boldsymbol{\theta_1}) B_1 Q_u^{-1} B_1^\top P(\boldsymbol{\theta_1}).
\end{aligned}
$$

Therefore, we can calculate $P(\boldsymbol{\theta_2}) B_2 Q_u^{-1} B_2^\top P(\boldsymbol{\theta_2})$. To that end, let $P = P(\boldsymbol{\theta_2}) - P(\boldsymbol{\theta_1})$, write $P(\boldsymbol{\theta_2})$ in terms of $P, P(\boldsymbol{\theta_1})$, and use the above result to get

$$
\begin{aligned}
&P(\boldsymbol{\theta_2}) B_2 Q_u^{-1} B_2^\top P(\boldsymbol{\theta_2}) \\
&= P(\boldsymbol{\theta_2}) B_2 Q_u^{-1} B_2^\top P + P(\boldsymbol{\theta_2}) B_2 Q_u^{-1} B_2^\top P(\boldsymbol{\theta_1}) \\
&= O\left(\|P\|^2\right) + P(\boldsymbol{\theta_1}) B_2 Q_u^{-1} B_2^\top P + P B_2 Q_u^{-1} B_2^\top P(\boldsymbol{\theta_1}) + P(\boldsymbol{\theta_1}) B_2 Q_u^{-1} B_2^\top P(\boldsymbol{\theta_1}) \\
&= O\left(\|P\|^2\right) + P(\boldsymbol{\theta_1}) B_2 Q_u^{-1} B_2^\top P + P B_2 Q_u^{-1} B_2^\top P(\boldsymbol{\theta_1}) \\
&\quad + O(\epsilon^2) + \epsilon P(\boldsymbol{\theta_1}) E_B Q_u^{-1} B_1^\top P(\boldsymbol{\theta_1}) + \epsilon P(\boldsymbol{\theta_1}) B_1 Q_u^{-1} E_B^\top P(\boldsymbol{\theta_1}) \\
&\quad + P(\boldsymbol{\theta_1}) B_1 Q_u^{-1} B_1^\top P(\boldsymbol{\theta_1}).
\end{aligned}
\tag{63}
$$

Again, expanding $A_2 = A_1 + E_A$ and $P(\boldsymbol{\theta_2}) = P(\boldsymbol{\theta_1}) + P$, it yields to

$$
\begin{aligned}
&A_2^\top P(\boldsymbol{\theta_2}) + P(\boldsymbol{\theta_2}) A_2 \\
&= A_2^\top P(\boldsymbol{\theta_1}) + A_2^\top P + P(\boldsymbol{\theta_1}) A_2 + P A_2 \\
&= A_1^\top P(\boldsymbol{\theta_1}) + \epsilon E_A^\top P(\boldsymbol{\theta_1}) + A_2^\top P \\
&\quad + P(\boldsymbol{\theta_1}) A_1 + \epsilon P(\boldsymbol{\theta_1}) E_A + P A_2.
\end{aligned}
$$

To proceed, plug in the continuous-time algebraic Riccati equation in (5) for $\boldsymbol{\theta_1}, \boldsymbol{\theta_2}$ below in the above expression:

$$
\begin{aligned}
A_2^\top P(\boldsymbol{\theta_2}) + P(\boldsymbol{\theta_2}) A_2 &= P(\boldsymbol{\theta_2}) B_2 Q_u^{-1} B_2^\top P(\boldsymbol{\theta_2}) + Q_x, \\
A_1^\top P(\boldsymbol{\theta_1}) + P(\boldsymbol{\theta_1}) A_1 &= P(\boldsymbol{\theta_1}) B_1 Q_u^{-1} B_1^\top P(\boldsymbol{\theta_1}) + Q_x.
\end{aligned}
$$

So, we obtain

$$
\begin{aligned}
&A_2^\top P(\boldsymbol{\theta_2}) + P(\boldsymbol{\theta_2}) A_2 - A_1^\top P(\boldsymbol{\theta_1}) - P(\boldsymbol{\theta_1}) A_1 \\
&= \epsilon E_A^\top P(\boldsymbol{\theta_1}) + \epsilon P(\boldsymbol{\theta_1}) E_A + P A_2 + A_2^\top P \\
&= P(\boldsymbol{\theta_2}) B_2 Q_u^{-1} B_2^\top P(\boldsymbol{\theta_2}) - P(\boldsymbol{\theta_1}) B_1 Q_u^{-1} B_1^\top P(\boldsymbol{\theta_1}) \\
&= O\left(\|P\|^2\right) + P(\boldsymbol{\theta_1}) B_2 Q_u^{-1} B_2^\top P + P B_2 Q_u^{-1} B_2^\top P(\boldsymbol{\theta_1}) \\
&\quad + O(\epsilon^2) + \epsilon P(\boldsymbol{\theta_1}) E_B Q_u^{-1} B_1^\top P(\boldsymbol{\theta_1}) + \epsilon P(\boldsymbol{\theta_1}) B_1 Q_u^{-1} E_B^\top P(\boldsymbol{\theta_1}),
\end{aligned}
$$

where in the last equality above, we used (63). Now, rearrange the terms in the above statement to get an equation that does not contain any expression in term of $\boldsymbol{\theta_2}$. So, it becomes

$$
\begin{aligned}
0 &= \left[ A_2^\top - P(\boldsymbol{\theta_1}) B_2 Q_u^{-1} B_2^\top \right] P + P \left[ A_2 - B_2 Q_u^{-1} B_2^\top P(\boldsymbol{\theta_1}) \right] - O\left(\|P\|^2\right) - O(\epsilon^2) \\
&\quad + \epsilon E_A^\top P(\boldsymbol{\theta_1}) + \epsilon P(\boldsymbol{\theta_1}) E_A - \epsilon P(\boldsymbol{\theta_1}) E_B Q_u^{-1} B_1^\top P(\boldsymbol{\theta_1}) - \epsilon P(\boldsymbol{\theta_1}) B_1 Q_u^{-1} E_B^\top P(\boldsymbol{\theta_1}).
\end{aligned}
$$

Next, to simplify the above equality, define the followings:

$$
\begin{aligned}
D &= A_2 - B_2 Q_u^{-1} B_2^\top P(\boldsymbol{\theta_1}), \\
K(\boldsymbol{\theta_1}) &= -Q_u^{-1} B_1^\top P(\boldsymbol{\theta_1}), \\
R &= \epsilon P(\boldsymbol{\theta_1}) \left[ E_A + E_B K(\boldsymbol{\theta_1}) \right] + \epsilon \left[ K(\boldsymbol{\theta_1})^\top E_B^\top + E_A^\top \right] P(\boldsymbol{\theta_1}) - O(\epsilon^2).
\end{aligned}
$$

So, writing our equation in terms of $D, K\left(\boldsymbol{\theta_1}\right), R$, it gives

$$0 = D^\top P + PD - O\left(\|P\|^2\right) + R. \tag{64}$$

The discussion after (6) states that all eigenvalues of $\overline{A}_1 = A_1 - B_1 Q_u^{-1} B_1^\top P\left(\boldsymbol{\theta_1}\right)$ lie in the open left half-plane. Therefore, if $\epsilon$ is small enough, real-parts of all eigenvalues of $D$ are negative, according to Lemma 5. Therefore, (64) implies that

$$\overline{\lambda}\left(P\right) \leq \overline{\lambda}\left(\int_0^\infty e^{D^\top t} R e^{Dt}\, \boldsymbol{dt}\right) \leq \|R\| \int_0^\infty \left\|e^{Dt}\right\|^2 \boldsymbol{dt}.$$

So, as $\epsilon$ decays, $R$ vanishes, which by the above inequality shows that $P$ shrinks as $\epsilon$ tends to zero. Further, as $\epsilon$ decays, $D$ converges to $\overline{A}_1$. Thus, by (64), we have

$$\lim_{\epsilon \to 0} \epsilon^{-1} P = \int_0^\infty e^{\overline{A}_1^\top t} \left(P\left(\boldsymbol{\theta_1}\right)\left[E_A + E_B K\left(\boldsymbol{\theta_1}\right)\right] + \left[E_A + E_B K\left(\boldsymbol{\theta_1}\right)\right]^\top P\left(\boldsymbol{\theta_1}\right)\right) e^{\overline{A}_1 t}\boldsymbol{dt}. \tag{65}$$

Recall that the above expression is the derivative of $P\left(\boldsymbol{\theta}\right)$ at $\boldsymbol{\theta_1}$, along the linear segment $\mathcal{C}$. Thus, integrating along $\mathcal{C}$, (65) and Cauchy-Schwarz Inequality imply that

$$\left\|P\left(\widehat{\boldsymbol{\theta}}\right) - P\left(\boldsymbol{\theta_0}\right)\right\| \;\lesssim\; \left\|\widehat{\boldsymbol{\theta}} - \boldsymbol{\theta_0}\right\| \sup_{\boldsymbol{\theta_1} \in \mathcal{C}} \|P\left(\boldsymbol{\theta_1}\right)\| \left(1 \vee \|K\left(\boldsymbol{\theta_1}\right)\|\right) \int_0^\infty \left\|e^{\overline{A}_1 t}\right\|^2 \boldsymbol{dt}.$$

Finally, using Lemma 11, (61), and (62), we obtain the desired result. ∎

# D   Numerical Results

In this section, we provide further empirical results illustrating the performance of Algorithm 2 in the settings of flight control, as well and blood glucose control. First, we provide box plots depicting the distribution of the normalized squared estimation error and the normalized regret of Algorithm 2 for X-29A airplane. Note that the corresponding worst- and average-case curves are presented in Figure 1. Then, Figures 3 and 4 provide the corresponding curves of estimation and regret versus time as well as the box-plots, for Boeing 747. Finally, we present similar empirical result for learning to control blood glucose level. As shown in the presented figures, Thompson sampling Algorithm 2 clearly outperforms the competing reinforcement learning policy.

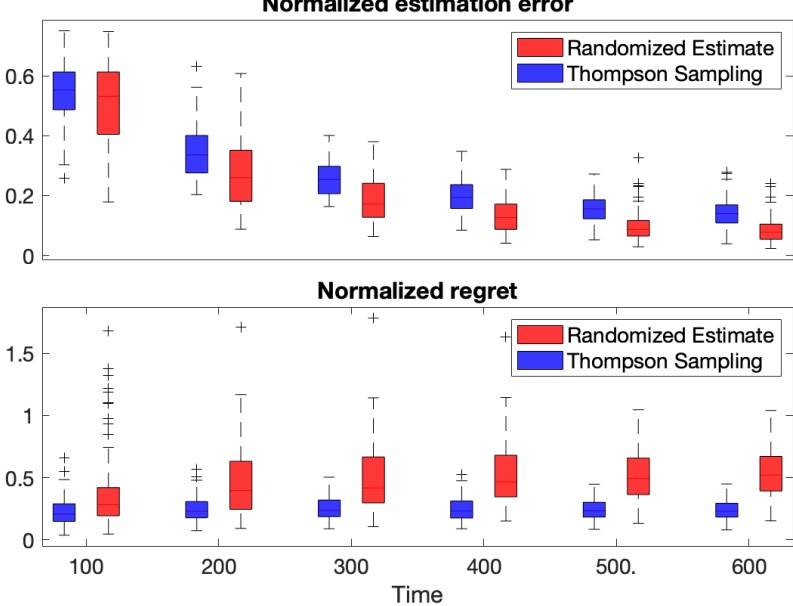

Figure 2: The performance of Algorithm 2 (blue) compared to Randomized Estimate policy (red) [2] for flight control of X-29A airplane. The top box-plots are for the normalized squared estimation error, $\left\|\widehat{\boldsymbol{\theta}_n} - \boldsymbol{\theta_0}\right\|^2$ divided by $p(p + q)\boldsymbol{\tau}_n^{-1/2} \log \boldsymbol{\tau}_n$, at times 100, 200, 300, 400, 500, and 600 for 100 replications. Similarly, the lower graph showcases the distribution of the regret $\mathrm{Reg}\,(T)$, normalized by $p(p + q)T^{1/2} \log T$.

Figure 2 depicts the box plot corresponding to Figure 1 that is for the flight control of X-29A airplane at 2000 ft. In the following experiments, we keep the setting given in Section 6 for the cost and noise covariance matrices, and compare Algorithm 2 to Randomized Estimate policy [2].

Next, the empirical results of the flight control problem in Boeing 747 airplane at 20000 ft altitude are provided [37]. The true drift matrices of the Boeing 747 are

$$A_0 = \begin{bmatrix} -0.199 & 0.003 & -0.980 & 0.038 \\ -3.868 & -0.929 & 0.471 & -0.008 \\ 1.591 & -0.015 & -0.309 & 0.003 \\ -0.198 & 0.958 & 0.021 & 0.000 \end{bmatrix}, \quad B_0 = \begin{bmatrix} -0.001 & 0.058 \\ 0.296 & 0.153 \\ 0.012 & -0.908 \\ 0.015 & 0.008 \end{bmatrix}.$$

Then, the blood glucose control problem is studied [41, 47]. The true drift matrices are

$$A_0 = \begin{bmatrix} 1.91 & -2.82 & 0.91 \\ 1.00 & -1.00 & 0.00 \\ 0.00 & 1.00 & -1.00 \end{bmatrix}, \quad B_0 = \begin{bmatrix} -0.0992 \\ 0.0000 \\ 0.0000 \end{bmatrix}.$$

Note that from a practical point of view, worst-case behavior are of crucial importance in this problem.

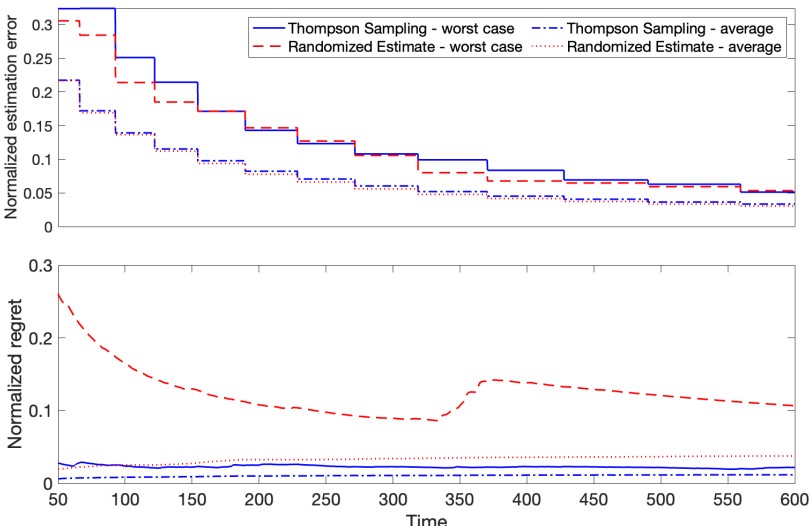

Figure 3: The performance of Algorithm 2 (blue) compared to Randomized Estimate policy (red) [2] for the flight control of Boeing 747 airplane. The top graph plots the normalized squared estimation error, $\left\|\widehat{\boldsymbol{\theta}}_{\boldsymbol{n}} - \boldsymbol{\theta}_{\boldsymbol{0}}\right\|^2$ divided by $p(p+q)\boldsymbol{\tau}_n^{-1/2}\log\boldsymbol{\tau}_n$, for 100 replications. Similarly, the lower graph showcases the regret $\operatorname{Reg}(T)$, normalized by $p(p+q)T^{1/2}\log T$.

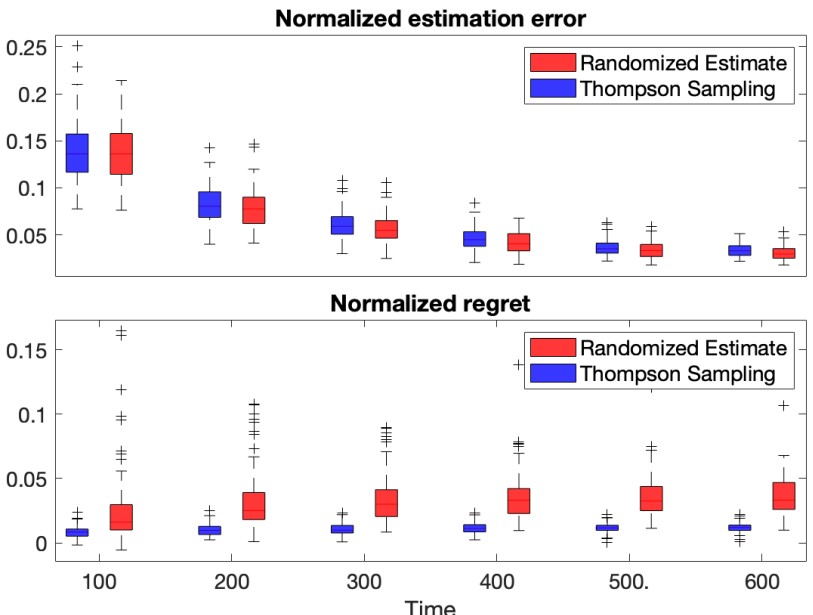

Figure 4: The performance of Algorithm 2 (blue) compared to Randomized Estimate policy (red) [2] for the flight control of Boeing 747 airplane. The top graph plots the normalized squared estimation error, $\left\|\widehat{\boldsymbol{\theta}}_{\boldsymbol{n}} - \boldsymbol{\theta}_{\boldsymbol{0}}\right\|^2$ divided by $p(p+q)\boldsymbol{\tau}_n^{-1/2}\log\boldsymbol{\tau}_n$, at times 100, 200, 300, 400, 500, and 600 for 100 replications. Similarly, the lower graph showcases the regret $\operatorname{Reg}(T)$, normalized by $p(p+q)T^{1/2}\log T$.

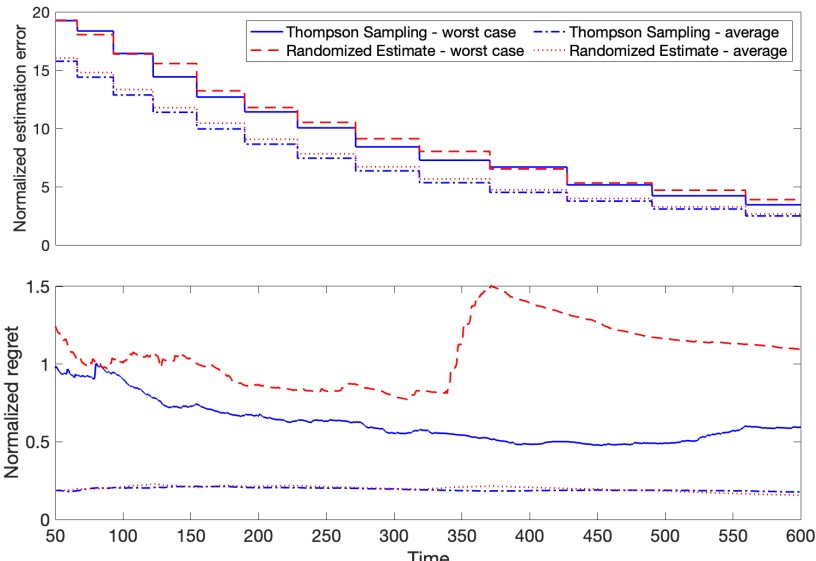

Figure 5: The performance of Algorithm 2 (blue) compared to Randomized Estimate policy (red) [2] for the the blood glucose control. The top graph plots the normalized squared estimation error, $\left\|\widehat{\boldsymbol{\theta}}_{\boldsymbol{n}} - \boldsymbol{\theta}_{\boldsymbol{0}}\right\|^2$ divided by $p(p+q)\boldsymbol{\tau}_n^{-1/2}\log\boldsymbol{\tau}_n$, for 100 replications. Similarly, the lower graph showcases the regret $\mathrm{Reg}\,(T)$, normalized by $p(p+q)T^{1/2}\log T$.

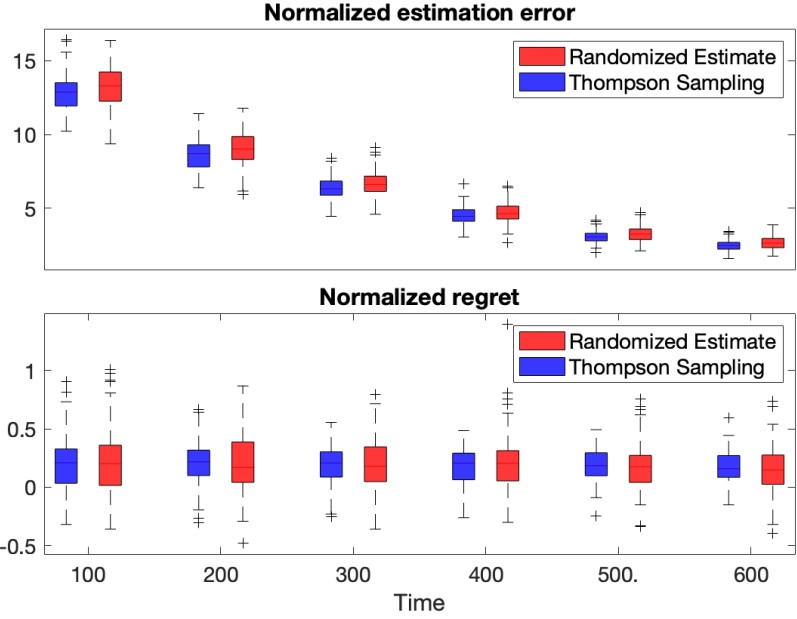

Figure 6: The performance of Algorithm 2 (blue) compared to Randomized Estimate policy (red) [2] for the the blood glucose control. The top graph plots the normalized squared estimation error, $\left\|\widehat{\boldsymbol{\theta}}_{\boldsymbol{n}} - \boldsymbol{\theta}_{\boldsymbol{0}}\right\|^2$ divided by $p(p+q)\boldsymbol{\tau}_n^{-1/2}\log\boldsymbol{\tau}_n$, at times 100, 200, 300, 400, 500, and 600 for 100 replications. Similarly, the lower graph showcases the regret $\mathrm{Reg}\,(T)$, normalized by $p(p+q)T^{1/2}\log T$.