# OpenReview forum: "Thompson Sampling Efficiently Learns to Control Diffusion Processes"
_NeurIPS.cc/2022/Conference — NeurIPS 2022 Accept_

### Official Review · Reviewer_u7y2 · 2022-06-29

**Rating:** 7
**Confidence:** 4
**Soundness:** 4 excellent
**Presentation:** 3 good
**Contribution:** 3 good

**Summary:**

The paper discusses Thompson sampling for linear-quadratic continuous-time stochastic optimal control problems.

The analysis and numerical algorithms are developed using a conjugate Gaussian prior distribution.
The paper discusses thoroughly theoretical implications, including (i) a bound on the probability of not stabilizing the system under a linear exploration strategy, (ii) a bound on the squared estimation error, and (iii) a finite-time regret bound.

The developed algorithm is numerically investigated using some synthetic examples of stochastic optimal control problems.

**Questions:**

Something I did not understand was in the paragraph in line 131:
"We assume that the process (1) with the drift parameter [...] is stabilizable. Therefore, [...] exists, is unique, and can be computed using continuous-time Riccati differential equations similar to (5), except that the zero matrix on the right-hand side will be replaced by the derivative of [...] [32–35]."
What is meant by this? I know of the Riccati equation for finite-time optimal control problems, which is an ODE. Though, here an average reward criterion is discussed. Maybe the authors can elaborate a bit on this.

In line 192: "Nonetheless, if there is no such prior, we simply let [...] and [...]." This formulation confuses me. I think it shadows a bit that the hyper-parameters are set in this way because in the proceeding bounds the hyper-parameters $\hat{\mu}_0$ and  $\hat{\Sigma}_0$ are not appearing anymore. Some explanation for this would be appreciated.

The last question I have is: Would it be possible to also derive these bounds under different posterior distributions than a Gaussian? For example when using a sparsity-prior.

**Limitations:**

Yes, the authors have adequately addressed the limitations and potential negative societal impact of their work.

**Strengths And Weaknesses:**

It was very surprising to me that nobody until now did a theoretical analysis for this linear-quadratic setting. In my opinion, this paper solidly develops theoretical guarantees for the popular Thompson sampling algorithm. Since model-based RL algorithms in continuous-time are still lacking within machine learning, I very much appreciate this work.
It is nicely written, and I enjoyed reading it. I appreciate that the authors discuss the intuitive implications of the bandit analysis, which is normally filled with hard-to-understand mathematical intricacies.

A minor point I have to critique is, in my opinion, the claim in line 70:  "Additionally, through extensive simulations we illustrate that TS enjoys smaller average regret and substantially lower worst-case regret than the existing RL policies, thanks to its informed exploration.", which is a bit far stretched, as this work only compares one other algorithm to this Thompson sampling strategy.

This brings me to my main critique point, which is that the topics adaptive control and dual control (Bayesian reinforcement learning within machine learning) are not sufficiently mentioned in the paper, see, e.g., [1]. Theory discussing simultaneous optimal estimation and control is unsurprisingly very old. This goes back to the more than 50-year-old works of Feldbaum [2] and is since then been discussed within the control community. This topic, which in principle optimally solves the problem has to be discussed within the related work.
This would also give a nice numerical example. For example, there are very simple dual control problems, see, e.g., [3], where one could try to find the solution to the HJB equation numerically using the finite difference method. This would result in a control strategy that Bayes-optimally selects the actions and hence, balances exploration and exploitation optimally.

However, all critique aside, I am of the opinion that this paper is a solid contribution to the community.

- [1] Stengel, Robert F. Optimal control and estimation. Courier Corporation, 1994.
- [2] Feldbaum, Aleksandr Aronovich. "Dual control theory. I." Avtomatika i Telemekhanika 21.9 (1960): 1240-1249.
- [3] Florentin, J. J. "Optimal, probing, adaptive control of a simple Bayesian system." International Journal of Electronics 13.2 (1962): 165-177.

---

> ### Author Response · Authors · 2022-08-02
> **Re Official Review by Reviewer u7y2**
>
> Thanks for the deep conceptual and technical comments the reviewer correctly provided. The authors appreciate the comprehensive review and the constructive comments, are grateful that the reviewer found the paper interesting, and will incorporate the edits in the final version. It is also satisfactory to hear that the reviewer found the intuitive explanations helpful.
>
> - ""Additionally, through … informed exploration.", is a bit far stretched, ... strategy."
> Thanks for the comment. We will rewrite this according to the comment, in the final version.
>
> - “the topics … are not sufficiently mentioned in the paper”
> We thank the reviewer for the relevant references. We will add them, as well as the following references, to the final version.
> This work is essentially similar to [2], [3], and it studies one of the closest policies to [2], [3] that admits both a fast implementation for multidimensional systems, as well as theoretical tractability. Technically, approaches based on the augmented dynamics method (for both the state and the unknown parameter) suggested in [2] are analyzed for a few special settings, e.g., [3], [7]. Furthermore, as reviewed in [4], since design and analysis of dual control policies utilize additional ideas such as the linearization of the dynamics and the quadratic expansions of the cost functions as in [5], [6], the authors expect the technical framework developed in this work (e.g., Lemma 9) to pave the road toward theoretical analysis of the performance of dual and adaptive policies for balancing exploration and exploitation.
>
> - [4] Wittenmark, Björn. "Adaptive dual control methods: An overview." Adaptive Systems in Control and Signal Processing 1995 (1995): 67-72.
> - [5] Klenske, Edgar D., and Philipp Hennig. "Dual control for approximate Bayesian reinforcement learning." The Journal of Machine Learning Research 17.1 (2016): 4354-4383.
> - [6] Tse, Edison, and Yaakov Bar-Shalom. "An actively adaptive control for linear systems with random parameters via the dual control approach." IEEE Transactions on Automatic Control 18.2 (1973): 109-117.
> - [7] Sternby, Jan. "A simple dual control problem with an analytical solution." IEEE Transactions on Automatic Control 21.6 (1976): 840-844.
>
> - ""We assume … [32–35]." What is meant by this? ... a bit on this."
> Thank you for the deep technical question. The aforementioned lines aim to explain that to compute the solution of the Riccati equation, it suffices to solve the ODE of the finite-horizon Riccati equation, and then let the horizon grow. Practically, it is based on the fact that for minimizing the average cost of this work, it suffices to minimize the cumulative cost for a ‘large’ horizon. Accordingly, if the finite horizon tends to infinity, the solutions of the Riccati equations converge to those of equation (5) in the paper. In the final version, we will edit according to the explanations of the reviewer to ensure that this computation method is clear.
>
> - "In line 192: "Nonetheless, ... let [...] and [...]." This formulation confuses ... would be appreciated."
> Thanks for the interesting question. The above-mentioned lines try to explain the case that a prior distribution of the unknown dynamics matrices is available, which will be used by the algorithm to have a better ‘initial’ exploration. Otherwise, standard multivariate Gaussian distribution can be adopted as a prior (although it is not actually). Note that the performance metric here is worst-case, which differs from an averaged one wrt the prior distribution (i.e., the regret is not the so-called Bayesian regret). Therefore, $\hat \mu_0, \hat \Sigma_0$ contribute as a constant and do not affect the rates of the performance of the algorithms in Theorem 2.
> In the final version, we rewrite the corresponding lines to further clarify the roles of $\hat \mu_0, \hat \Sigma_0$.
>
> - "Would it ... sparsity-prior."
> Thanks for the interesting question. The approach can be extended to non-Gaussian posterior distributions, but further technical details are required in some cases. More precisely, as long as the drift matrices do not possess any structure (such as sparsity), every posterior that concentrates appropriately can be employed. That is, the presented analysis extends to cases where the posterior does not concentrate faster or slower than Gaussian, and has a sub-Gaussian tail as well as a bounded probability density function (e.g., mixtures of Gaussians).
> On the other hand, under structured dynamics parameters, some of the technical results can be directly used for non-Gaussian priors/posteriors (e.g., Lemmas 1, 2, 3, 5, 6, 7, 10, 11, and 12, assuming sparsity of the dynamics matrices). For some others, accordingly appropriate counterparts are required (e.g., Lemmas 4, 8, and 9, for sparse drift parameters). The authors believe these extensions constitute interesting directions for future work that this paper paves the road toward.

---

### Official Review · Reviewer_Vgkm · 2022-07-10

**Rating:** 5
**Confidence:** 4
**Soundness:** 3 good
**Presentation:** 3 good
**Contribution:** 2 fair

**Summary:**

The paper studies policies for systems that evolve according to Ito stochastic differential equation (1) with unknown drift parameters. The cost function is quadratic. The goals are: first, to minimize the regret; and second, to accurately estimate the drift parameters. The authors first propose Alg. 1 and show that with high probability it stabilize the process. The they propose Alg 2 using Thompson Sampling. Regret bound and estimation rates are given in Theorem 2. Experiments are conducted comparing the proposed method to reference [2].

**Questions:**

The first part of the paper focuses on stabilization. But Definition 1 is given before proper introduction. Moreover, the definition directly refers to the eigenvalues of the closed-loop matrix, which uses a linear feedback. Perhaps it would be better to give some intuitions about stability of the system and what the failure event implies. Also, does Algorithm 2 have similar stability result?

The numerical results in Figure 1 suggest that the randomized estimation performs better in terms of the normalized estimation error. How would you interpret the results?

The problem setting is very similar to reference [2]. It would be better to elaborate the introduction about [2], especially because it is also used as the baseline.

**Strengths And Weaknesses:**

Strengths:
There is a gap in the literature on applying TS to controlling the continuous-time diffusion process. The paper first provide algorithms and analysis for such method. Detailed proofs are given in the appendices.

Weaknesses:
The problem setting is very similar to reference [2]. The experimental results in section 6 and appendix D do not  reflect a significant improvement. In fact, the normalized estimation error using randomized estimation seems to be better than the proposed method on average.

---

> ### Author Response · Authors · 2022-08-02
> **Re Official Review by Reviewer Vgkm**
>
> We appreciate the helpful comments of the reviewer. According to them, clarifying explanations and relevant edits to the final version of the paper are provided in the sequel, and the authors will perform further edits the reviewer may recommend.
>
> Below, we elaborate differences compared to [2], including that Algorithm 2 significantly outperforms the RL policy in [2], and Algorithm 1 and its performance guarantee, both are novel. In addition, the reviewers and the authors believe that performance guarantees for the popular TS policy in the canonical systems of linear diffusion processes is a theoretically interesting strong contribution to RL theory.
>
> - “Definition 1 … implies.”
> Thanks for the comment. Following this comment, in the final version of the paper, we will bring the explanations in Section 3 (Stabilizing the Diffusion Process), in lines 149-158, before Definition 1, and edit further to ensure clarity. We will also discuss the failure event and how the diffusion process grows unboundedly in case of failure, as well as its remedies.
>
> - “Also, does Algorithm 2 have similar stability result?”
> Thanks for the interesting technical question. Algorithm 2 utilizes Algorithm 1 for learning to stabilize the diffusion process under consideration. Therefore, by Theorem 1, with high probability it stabilizes the system. As the same logic applies to the subsequent episodes of Algorithm 2, in case of no ‘resampling’ (as further explained below), it stabilizes the process with high probability.
> In addition, thanks to the randomness provided by the posterior distribution, the failure probability can shrink further by ‘resampling’ from the posterior (lines 175-177). More precisely, in case of failure of Algorithm 1 (which can be detected since the magnitude of the state vector drastically grows with time if the system is not stable), once can simply ‘resample’ from the posterior. Similarly, by employing a resampling strategy at the end of an episode of Algorithm 2 in case of observing any instability, the state evolution under the algorithm almost surely remains stable. Note that the failure probability decays exponentially as time proceeds, by Theorem 1. These explanations will be added to those in lines 224-230 in the paper.
>
> - “The numerical results in Figure 1 suggest that the randomized estimation performs better in terms of the normalized estimation error. How would you interpret the results?”
> Thanks for the interesting technical question. The interpretation of the better estimation error of the randomized estimation policy compared to Thompson sampling, while the regret comparison holds in the opposite direction, is as follows. The former policy ‘over-explores’ for learning the parameters and unduly deviates from exploiting efficiently based on the parameter estimates at the time. It can be seen in the empirical experiments that infrequently the randomized estimation policy has a better learning accuracy, while its regret is always inferior to that of Thompson sampling. Importantly, since the end goal is to have policies with small regret, numerical results showcase superiority of Thompson sampling. Note that this superiority is significant since in the graphs, ‘magnified’ squared estimation errors are reported (multiplied by $\sqrt T$, approximately), while regret curves contain the actual regret values ‘divided’ by $\sqrt T$. So, the smaller regret of Thompson sampling is consequential and important in practice. We will add these explanations to the final version of the paper.
>
> - “It would be better to elaborate the introduction about [2].”
> Thanks for the helpful suggestion. As per the above discussions, we will further elaborate and discuss comparisons with [2] in the introduction of the final version.

---

### Official Review · Reviewer_59Mc · 2022-07-11

**Rating:** 6
**Confidence:** 4
**Soundness:** 3 good
**Presentation:** 3 good
**Contribution:** 2 fair

**Summary:**

This paper proposes a Thompson-Sampling-based method to learn how to make decisions in a class of continuous-time linear-quadratic (LQ) problems with unknown coefficients in the linear dynamics.


**Questions:**

Please address the comments listed above.

**Limitations:**

Yes.

**Strengths And Weaknesses:**

Pros: Reinforcement Learning (RL) for LQ problems is a tropical topic in recent years since it is the building to understand how machine learning methods perform for general decision-making problems. Thompson Sampling, on the other hand, is a practically-popular and theoretically-plausible algorithm for bandits and reinforcement learning method for Markov Decision Process (MDP) due to its efficiency in exploration. Therefore, it is interesting to see what additional benefit Thompson Sampling could bring to the LQ problem compared to the existing methods.

Cons: The contribution seems to be marginal compared to two existing papers [1] and [2]. [1] showed that the continuous-time least-squares algorithm leads to logarithmic regret and hence greedy algorithm (with no exploration) is sufficient for finite-horizon LQ problem. [2] proved an $O(\sqrt{T})$ regret for infinite-horizon ergodic LQ problem in discrete time.

It is not clear, from both theoretical and empirical perspectives, what is the benefit of using Thompson Sampling (or Guassian exploration on the estimated model parameters). This is because it has been shown in the literature that:

(1) LG problems have self-exploration properties (due to the Gaussian noise in the linear dynamics) and the least-square estimate (exploration-free method) leads to log regret bound [1]

(2) Using linear regression to estimate parameters in the dynamics is sample efficient [3]

[1] Basei, Matteo, et al. "Logarithmic regret for episodic continuous-time linear-quadratic reinforcement learning over a finite-time horizon." arXiv preprint arXiv:2006.15316 (2020).

[2] Abeille, Marc, and Alessandro Lazaric. "Improved regret bounds for thompson sampling in linear quadratic control problems." International Conference on Machine Learning. PMLR, 2018.

[3] Dean, Sarah, et al. "On the sample complexity of the linear quadratic regulator." Foundations of Computational Mathematics 20.4 (2020): 633-679.

==========
I have read the reviews from other reviewers and the responses from the authors. The authors have addressed my concerns and I raised my rating to 6.

---

> ### Author Response · Authors · 2022-08-02
> **Re Official Review by Reviewer 59Mc**
>
>
> We thank the reviewer for the helpful feedback and are happy that they found the setting interesting for studying the popular Thompson sampling policy for the continuous-time LQ problem. Below, we address the comments and hope that the reviewer finds them satisfactory.
>
> The references [1] and [2] (cited in the paper as [29] and [16]) are important motivating papers for the authors to establish and present the results of this paper. In addition to the theoretical benefit of answering a natural question about Thompson sampling in continuous-time LQ systems, this work aims to fill some gaps in the existing literature (as expressed by Reviewers Vgkm and u7y2), for which different technical novelties are established. More details are provided below.
>
> On [1], the self-exploration property of finite (in practice, short) horizon problem does not hold in the infinite (or practically long) horizon problem considered in this work. Technically, Assumption H.1 (2)  in [1] is crucial in the sense that it renders exploration unnecessary (as discussed in Remark 2.1 therein). However, since in the setting of this work, Assumption H.1 (2) ‘cannot’ hold, the exploration-exploitation trade-off necessitates exploring the environment and precludes logarithmic regret.
>
> Further technical differences indicating that the analysis in [1] is inapplicable, are as follows. The short horizon in [1] makes stabilization unnecessary, while it is crucially required for single trajectory online RL policies of this work that cannot reset the state of the system. Accordingly, we proposed Algorithm 1 and established its performance guarantee by developing novel technical results, as discussed in Section 5. Finally, the definition of regret in [1] does not fully include the stochasticity induced by the Wiener process, while here a comprehensive worst-case analysis is performed (as presented in Lemmas 7 and 8).
>
> On the other hand, the analysis in [2] for Thompson sampling in discrete-time, focuses on one-dimensional systems. In contrast, Theorem 2 specifies dependence of the regret and estimation error on ambient dimension, as well as the other problem instances. Moreover, the conventional approach for discrete-time systems, as in [2] and [3], relies on concentration inequalities, which does not extend to continuous-time settings. In fact, because of technical difficulties including the fact that “continuous-time martingales have sub-exponential distributions, unlike sub-Gaussianity of discrete-time counterparts” (line 215), the theoretical analyses that establish rates for continuous-time settings that are similar to the corresponding rates in discrete-time systems, are considered as strong theoretical contributions.

---

### Official Review · Reviewer_qy7p · 2022-07-11

**Rating:** 6
**Confidence:** 1
**Soundness:** 3 good
**Presentation:** 3 good
**Contribution:** 3 good

**Summary:**

I do not have the background to give a reasonable assessment. So, please ignore this review.

**Questions:**

No

**Ethics Review Area:**

["I don’t know"]

**Limitations:**

I can't see any limitations and potential negative societal impact from their work.

**Strengths And Weaknesses:**

I do not have the background to give a reasonable assessment. So, please ignore this review.

---

> ### Author Response · Authors · 2022-08-02
> **Re Official Review by Reviewer qy7p**
>
> Thanks for the feedback. The authors will be happy to provide point-by-point explanations to all questions of the reviewer.

---

### Meta-Review · Area_Chair_dE9T · 2022-08-25

**Recommendation:** Accept
**Confidence:** Certain

**Metareview:**

This paper proposes and analyzes a Thompson-Sampling based method to learn to control continuous-time linear systems when the costs are quadratic. The authors first propose an algorithm that guarantees stabilization of the diffusion process and then give a second, Thompson-Sampling-based method with regret bounds and estimation rates for the parameters of the linear system.

The reviews for this paper were generally positive and found this work to positively contribute to our understanding of linear control--- though several reviews noted the similarities with with reference [2] and a general lack of contextualization of the work in the general adaptive and Bayesian control literatures. Nevertheless the results were sound---and extended our understanding of learning and control in the LQ setting, and the paper was well written and easy to follow.

**Award:**

No

---

### Decision · Program_Chairs · 2022-09-14

Accept